# Annexin A5 controls VDAC1-dependent mitochondrial Ca²⁺ homeostasis and determines cellular susceptibility to apoptosis

Furkan E Oflaz [1], Alexander I Bondarenko [1], Michael Trenker [1,7], Markus Waldeck-Weiermair[1,8], Benjamin Gottschalk [1], Eva Bernhart[1], Zhanat Koshenov[1,2], Snježana Radulović[3], Rene Rost[1], Martin Hirtl [1], Johannes Pilic[1], Aditya Karunanithi Nivedita [4], Adlet Sagintayev[1], Gerd Leitinger[3], Bent Brachvogel[5], Susanne Summerauer[3], Varda Shoshan-Barmatz [4], Roland Malli [1,6] & Wolfgang F Graier [1,6]✉

## Abstract

**Annexin A5 (AnxA5) is a Ca²⁺-dependent phospholipid-binding protein associated with the regulation of intracellular Ca²⁺ homeostasis. However, the precise role of AnxA5 in controlling mitochondrial Ca²⁺ signaling remains elusive. Here, we introduce a novel function of AnxA5 in regulating mitochondrial Ca²⁺ signaling. Our investigation revealed that AnxA5 localizes at and in the mitochondria and orchestrates intermembrane space Ca²⁺ signaling upon high Ca²⁺ elevations induced by ER Ca²⁺ release. Proximity ligation assays and co-immunoprecipitation revealed a close association but no direct contact of AnxA5 with the voltage-dependent anion channel (VDAC1) in the outer mitochondrial membrane (OMM). In single-cell mitochondrial Ca²⁺ measurements and electrophysiological recordings, AnxA5 was found to enhance Ca²⁺ flux through the OMM by promoting the Ca²⁺-permeable state of VDAC1. By modulating intermembrane space Ca²⁺ signaling, AnxA5 shapes mitochondrial ultrastructure and influences the dynamicity of the mitochondrial Ca²⁺ uniporter. Furthermore, by controlling VDAC1's oligomeric state, AnxA5 is protective against cisplatin and selenite-induced apoptotic cell death. Our study uncovers AnxA5 as an integral regulator of VDAC1 in physiological and pathological conditions.**

**Keywords** Annexin-A5; VDAC1 Ca²⁺ Permeability; Intermembrane Space Ca²⁺ Signaling; Apoptotic Cell Death
**Subject Categories** Autophagy & Cell Death; Metabolism; Signal Transduction

## Introduction

Intracellular calcium (Ca²⁺) homeostasis is a fundamental process governing various cellular functions, including survival, growth, and apoptosis (Brini et al, 2013). Annexin 5 (AnxA5) is a soluble, mainly cytosolic protein that binds to negatively charged phospholipids in a Ca²⁺-dependent manner (Carmeille et al, 2015; Lin et al, 2020; Rosenbaum et al, 2011) while the conditions for dissociation are unknown. Furthermore, AnxA5 is involved in the regulation of intracellular Ca²⁺ signaling (Kubista et al, 1999) and apoptosis (Hawkins et al, 2002; Jeong et al, 2014). AnxA5 has also been shown to facilitate Ca²⁺ influx in large unilamellar vesicles (Berendes et al, 1993) and liposomes via Ca²⁺-dependent membrane insertion, with its functionality depending on the lipid composition (Kirsch et al, 1997). Likewise, AnxA5 establishes Ca²⁺ influx across the plasma membrane upon stimulation with high hydrogen peroxide (H₂O₂) levels in chicken DT40 cells (Kubista et al, 1999). This phenomenon has been attributed to the insertion of AnxA5 into the cell membrane in the presence of peroxide (Kubista et al, 1999).

In mitochondria, AnxA5 binds to cardiolipin in the presence of physiological Ca²⁺ concentrations (Megli et al, 1995) and immobilizes cardiolipin microdomains in mitochondrial membranes (Megli et al, 2000a). In this context, AnxA5 significantly impacts mitochondria-mediated apoptosis. Notably, AnxA5 depletion enhances resistance to Ca²⁺-induced apoptosis triggered by staurosporine, H₂O₂ (Hawkins et al, 2002), and cisplatin, the latter of which prompts AnxA5 translocation to mitochondria (Jeong et al, 2014). Moreover, these stimuli have been shown to trigger apoptosis by inducing oligomerization of Voltage-Dependent Anion Channel 1 (VDAC1) (Shoshan-Barmatz et al, 2010). Interestingly, the depletion of AnxA5 has been demonstrated to

¹Gottfried Schatz Research Center, Molecular Biology and Biochemistry, Medical University of Graz, Graz, Austria. ²Department of Biochemistry, Weill Cornell Medicine, New York, USA. ³Gottfried Schatz Research Center: Cell Biology, Histology and Embryology, Medical University of Graz, Graz, Austria. ⁴Department of Life Sciences, and the National Institute for Biotechnology in the Negev, Ben-Gurion University of the Negev, 84105 Beer Sheva, Israel. ⁵Department of Pediatrics and Adolescent Medicine, Experimental Neonatology, Faculty of Medicine and University Hospital Cologne, University of Cologne, Cologne, Germany. ⁶BioTechMed Graz, Graz, Austria. ⁷Present address: MM Frohnleiten GmbH, Frohnleiten, Austria. ⁸Present address: Division of Cardiovascular Medicine, Department of Medicine, Brigham and Women's Hospital, Harvard Medical School, 75 Francis Street, Boston, MA 02115, USA. ✉E-mail: wolfgang.graier@medunigraz.at

confer resistance to apoptotic signals by regulating the expression level of VDAC1 (Jeong et al, 2014).

In addition to its importance in apoptosis-inducing pathways, VDAC1 serves as a $Ca^{2+}$-permeable channel (Báthori et al, 2006). Upon cell stimulation by an inositol 1,4,5-trisphosphate ($IP_3$)-generating agonist, $IP_3$ induces $Ca^{2+}$ release from the endoplasmic reticulum (ER) (Giacomello et al, 2010), and $Ca^{2+}$ ions pass the outer mitochondrial membrane (OMM) via VDAC1. The permeability of VDAC1 to $Ca^{2+}$ ions is influenced by several proteins such as α-synuclein and B-cell lymphoma-extra-large (Bcl-xL) (Rosencrans et al, 2021; Monaco et al, 2015; Huang et al, 2013). Within the intermembrane space (IMS), $Ca^{2+}$ needs to exceed a concentration threshold to cross the inner mitochondrial membrane (IMM) via the mitochondrial $Ca^{2+}$ uniporter complex (Baughman et al, 2011; De Stefani et al, 2011; Madreiter-Sokolowski et al, 2016; Waldeck-Weiermair et al, 2015) that is counteracted by the mitochondrial $Ca^{2+}$ efflux machinery located in the IMM, known as mitochondrial $Na^+/Ca^{2+}$ exchanger, that facilitates $Ca^{2+}$ efflux (Palty et al, 2010).

These observations led us to hypothesize that AnxA5 plays a role in modulating mitochondrial functions. Therefore, we investigated the role of AnxA5 in intracellular $Ca^{2+}$ signaling across various cellular and mitochondrial compartments using genetically encoded $Ca^{2+}$ sensors and electrophysiology. We found that AnxA5 regulates trans-OMM $Ca^{2+}$ fluxes and IMS $Ca^{2+}$ signaling upon ER $Ca^{2+}$ release. By localizing in the vicinity of VDAC1, AnxA5 modulates its $Ca^{2+}$ permeability, and safeguards against cisplatin- and selenite-induced apoptosis by controlling VDAC1's oligomeric status.

# Results

## AnxA5 controls mitochondrial matrix $Ca^{2+}$ signaling

To study the potential involvement of AnxA5 in mitochondrial $Ca^{2+}$ homeostasis, we applied the CRISPR/Cas approach to knockout AnxA5 (AnxA5-KO) in HeLa cells and the human umbilical vein-derived endothelial cell line EA.hy926 (Fig. EV1A,B). Mitochondrial matrix $Ca^{2+}$ levels ($[Ca^{2+}]_{Matrix}$) were measured with the genetically encoded matrix-targeted ratiometric $Ca^{2+}$ sensor, 4mtD3cpv (Palmer et al, 2006). Under resting conditions, the levels of mitochondrial $[Ca^{2+}]_{Matrix}$ were comparable between WT and AnxA5-KO in HeLa (Fig. 1A,B) and AnxA5-KO EA.hy926 cells (Fig. EV1E,F). Intriguingly, the mitochondrial $Ca^{2+}$ elevation upon $IP_3$-induced ER $Ca^{2+}$ release was strongly reduced in AnxA5-KO HeLa (Fig. 1A,C) and EA.hy926 cells (Fig. EV1E,G). A transient expression of AnxA5 in the AnxA5-KO HeLa cells successfully restored the mitochondrial $Ca^{2+}$ signal induced by the $IP_3$-generating agonist ATP (Fig. 1A,C).

To verify whether the lack of AnxA5 might impact the quantity of agonist-induced ER $Ca^{2+}$ release, thus resulting in a decreased mitochondrial $Ca^2$ signal, cytosolic $Ca^{2+}$ levels ($[Ca^{2+}]_{Cyto}$) were measured using Fura2-AM. Under basal conditions and upon ATP stimulation, cytosolic $Ca^{2+}$ levels in AnxA5-KO cells were comparable to WT (Fig. 1D–F) in HeLa cells. In the endothelial cell line, basal cytosolic $Ca^{2+}$ levels were comparable between AnxA5-KO and WT cells (Fig. EV1H,I). However, AnxA5-KO endothelial cells showed an enhanced cytosolic $Ca^{2+}$ elevation upon ER $Ca^{2+}$ release with histamine compared to WT cells

(Fig. EV1H,J), possibly pointing to a reduced mitochondrial $Ca^{2+}$ buffering capacity in the Anx5-KO cells.

We further monitored the $[Ca^{2+}]_{ER}$ levels with a genetically encoded ER-targeted $Ca^{2+}$ sensor (Palmer et al, 2004). To track $[Ca^{2+}]_{ER}$ dynamics, cells were stimulated with histamine in nominal $Ca^{2+}$-free buffer to induce $Ca^{2+}$ depletion, followed by the perfusion in $Ca^{2+}$-containing buffer to monitor the refilling kinetics of the ER (Fig. 1G). Notably, ER-$Ca^{2+}$ signals in response to $IP_3$-induced release and the subsequent refilling were comparable in WT and AnxA5-KO cells (Fig. 1G,H). The inhibition of the sarco/endoplasmic reticulum $Ca^{2+}$ ATPase (SERCA) with tert-butylhydroquinone (BHQ) led to a comparable depletion of $[Ca^{2+}]_{ER}$ in both WT and AnxA5-KO cells (Fig. 1G,I). To eliminate the potential variability in the $Ca^{2+}$ extrusion rate of the mitochondrial $Na^+/Ca^{2+}$ exchanger between WT and AnxA5-depleted cells, we stimulated the cells with histamine in the presence of the mitochondrial $Na^+/Ca^{2+}$ exchanger inhibitor CGP37157. CGP37157 blocked mitochondrial $Ca^{2+}$ extrusion without affecting the maximum $[Ca^{2+}]_{Matrix}$ elevation in AnxA5-KO cells (Fig. EV1K–M), thus indicating that reduced $[Ca^{2+}]_{Matrix}$ signaling in AnxA5-KO cells cannot be due to enhanced activity of the mitochondrial $Na^+/Ca^{2+}$ exchanger. Collectively, these results indicate that AnxA5 regulates mitochondrial $[Ca^{2+}]_{Matrix}$ homeostasis without interfering with mitochondrial $Ca^{2+}$ extrusion rate, and the cytosolic and ER $Ca^{2+}$ signaling upon $IP_3$-generating agonists.

To strengthen our findings on the involvement of AnxA5 in the process(es) of mitochondrial $Ca^{2+}$ signaling, we further investigated the mitochondrial $[Ca^{2+}]_{Matrix}$ levels in perivascular cells isolated from WT and AnxA5-KO mice (Brachvogel et al, 2007, 2003, 2005). In line with our data shown in HeLa and endothelial cells, basal $[Ca^{2+}]_{Matrix}$ levels were not changed (Fig. 1J,K), whereas compared to WT, mitochondrial $[Ca^{2+}]_{Matrix}$ signaling in response to ATP-induced ER-$Ca^{2+}$ release was diminished in AnxA5-KO perivascular cells (Fig. 1J–L). Basal $Ca^{2+}$ levels and histamine-induced $[Ca^{2+}]_{Cyto}$ rises were not changed (Fig. 1M–O). These results strongly point to an indispensable role of AnxA5 in mitochondrial $[Ca^{2+}]_{Matrix}$ signaling.

## AnxA5 depletion has no impact on the mitochondrial membrane potential and the expression of core components of the $Ca^{2+}$ uniporter machinery

As the mitochondrial membrane potential ($\Psi_{mito}$) facilitates mitochondrial $Ca^{2+}$ uptake by providing the driving force for ion influx (Zorova et al, 2018), the impact of AnxA5 removal on $\Psi_{mito}$ was tested. The depletion of AnxA5 did not impact on the $\Psi_{mito}$ (Fig. 2A,B).

We assessed the expression levels of key proteins involved in mitochondrial $Ca^{2+}$ uptake, including VDAC1, mitochondrial calcium uptake 1 and 2 (MICU1 and MICU2), uncoupling protein 2 (UCP2), mitochondrial $Ca^{2+}$ uniporter (MCU) and essential MCU regulator (EMRE) in both WT and AnxA5-KO HeLa cells (Fig. 2C). Quantification of the immunoblots demonstrated comparable expression of respective proteins in WT and AnxA5-KO cells (Fig. 2D).

Next, we analyzed the mRNA expression levels of the annexin family in AnxA-5KO cells. Most annexins (AnxA1, AnxA2, AnxA3,

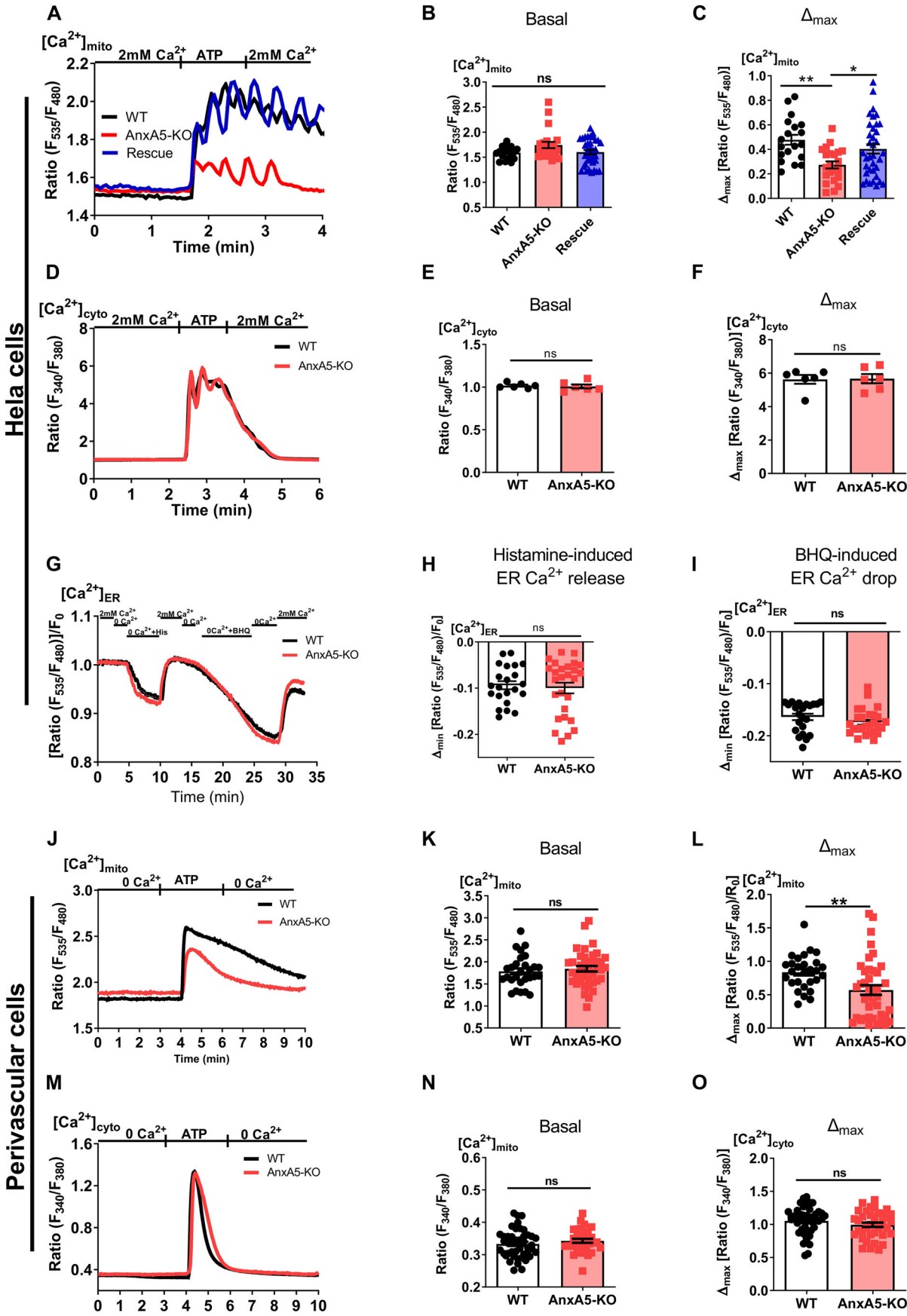

**Figure 1.** **AnxA5 regulates mitochondrial matrix Ca²⁺ homeostasis in different cell lines.**

(A) Representative time courses of the 100 µM ATP-induced $[Ca^{2+}]_{Matrix}$ responses in WT (black), AnxA5-KO (red), and rescue (blue) in HeLa cells measured in Ca²⁺ (2 mM) containing buffer. (B) Bar graphs show the basal $[Ca^{2+}]_{Matrix}$ and (C) ATP-induced maximum $[Ca^{2+}]_{Matrix}$ level in WT (black), AnxA5-KO (red), and Rescue (blue). Data points represent the mean ± SEM ($n_{WT} = 19/5$; $n_{AnxA5-KO} = 22/5$; $n_{Rescue} = 36/9$). The p-values were calculated using one-way ANOVA with Tukey's multiple comparison test. From left to right: $p = 0.0034$ (**$p < 0.01$) and $p = 0.0404$ (*$p < 0.05$). (D) Mean time courses of the ATP-induced $[Ca^{2+}]_{Cyto}$ responses in WT (black) and AnxA5-KO (red) in HeLa cells measured in Ca²⁺ (2 mM) containing buffer. (E) Bar graphs show the basal $[Ca^{2+}]_{Cyto}$ and (F) ATP-induced maximum $[Ca^{2+}]_{Cyto}$ levels in WT (black) and AnxA5-KO (red). Data points represent the mean ± SEM ($n_{WT} = 103/6$; $n_{AnxA5-KO} = 107/6$). (G) Average time courses of the 100 µM histamine and 15 µM BHQ induced $[Ca^{2+}]_{ER}$ responses in WT (black) and, AnxA5-KO (red) Hela cells. (H) Bar graphs show the 100 µM Histamine-induced $[Ca^{2+}]_{ER}$ release and (I) 15 µM BHQ-induced $[Ca^{2+}]_{ER}$ drop in WT (black) and AnxA5-KO (red) cells. Data points represent the mean ± SEM ($n_{WT} = 22/4$; $n_{AnxA5-KO} = 27/5$). (J) Mean time courses of the 100 µM ATP-induced $[Ca^{2+}]_{Matrix}$ responses in WT (black) and AnxA5-KO (red) perivascular cells measured in Ca²⁺-free buffer (containing 100 µM of EGTA). (K) Bar graphs show the basal $[Ca^{2+}]_{Matrix}$ and (L) ATP-induced maximum $[Ca^{2+}]_{Matrix}$ levels in WT (black) and AnxA5-KO (red) cells. Data points represent the mean ± SEM ($n_{WT} = 29/6$; $n_{AnxA5-KO} = 40/6$). The p-value was calculated using a two-tailed unpaired Student's t-test, $p = 0.0062$ (**$p < 0.01$). (M) Mean time courses of the 100 µM ATP-induced $[Ca^{2+}]_{Cyto}$ responses in WT (black) and AnxA5-KO (red) in perivascular cells measured in Ca²⁺-free buffer (containing 100 µM of EGTA). (N) Bar graphs show the basal $[Ca^{2+}]_{Cyto}$ and (O) ATP-induced maximum $[Ca^{2+}]_{Cyto}$ levels in WT (black) and AnxA5-KO (red). Data points represent the mean ± SEM ($n_{WT} = 46/6$; $n_{AnxA5-KO} = 33/6$). $n$ represents the number of cells/biological replicates (minimum of 3 independent experiments). Significant differences were assessed using either one-way ANOVA with Tukey's multiple comparison tests or Kruskal–Wallis test (*$p < 0.05$, **$p < 0.01$ and ns: not significant) and with the unpaired Student's t-test or Kolmogorov–Smirnov test (**$p < 0.001$ and ns: not significant). Source data are available online for this figure.

AnxA4, AnxA6, AnxA7, and AnxA11) exhibited similar expression levels in AnxA5-KO cells compared to WT, except for AnxA8 and AnxA9, which showed a 60% and 50% reduction, respectively (Appendix Fig. S1A). To test whether the reduction of AnxA8 or AnxA9 contributes to the observed reduction of mitochondrial Ca²⁺ signaling in AnxA5-KO cells, mitochondrial Ca²⁺ signaling was tested in AnxA8 and AnxA9 knockdown HeLa cells. These experiments did not reveal any impact of the knockdown of AnxA8 or that of AnxA9 on mitochondrial Ca²⁺ signaling upon IP₃-mediated ER Ca²⁺ release (Appendix Fig. S1B–D).

Collectively, these results indicate that the reduced $[Ca^{2+}]_{Matrix}$ elevation upon cell stimulation with IP₃-generating agonists in AnxA5-KO cells cannot be attributed to alterations in $\Psi_{mito}$, changes in the expression levels of proteins associated with mitochondrial Ca²⁺ uptake or reduced expression of AnxA8 and AnxA9 in KO cells.

## AnxA5 depletion has no impact on MERCs but alters the structure of mitochondria

Mitochondria and the ER build the so-called "mitochondria endoplasmic reticulum contact sites" (MERCs), enabling the mitochondria to benefit from localized Ca²⁺ hotspots fueled by ER Ca²⁺ release (Giacomello et al, 2010; Szabadkai et al, 2006). Because disruption of the contact sites harms the efficiency of the Ca²⁺ transfer from the ER to mitochondria (D'Eletto et al, 2018), we compared the integrity of MERCs in WT and AnxA5-KO cells using 3D-confocal microscopy (Fig. 2E). As indicated by the Pearson correlation coefficient, the MERCs regions remained intact in AnxA5-KO cells (Fig. 2F). To confirm this finding with an alternative approach, we employed SPLICS technology, a split-GFP-based ER-mitochondria contact site sensor (Cieri et al, 2018). This sensor is engineered to fluoresce when the ER and mitochondria are in close proximity (Appendix Fig. S1E). Using this technology, we confirmed similar levels of MERCs between WT and KO cells (Appendix Fig. S1F,G). For testing the reliability of this technique we fixed mitochondria to the sub-plasma membrane region with the AKAP-RFP-CAAX construct (Csordás et al, 2006). This procedure significantly reduced the signal pointing to MERCs (Appendix Fig. S1G). We also investigated whether there were any structural differences in mitochondrial

morphology between WT and AnxA5-KO cells (Fig. 2E). Interestingly, we found that AnxA5-KO cells showed an increase in mitochondrial volume (Fig. 2G), surface area (Appendix Fig. S1H), and branching (Fig. 2H). The mitochondrial elongation factor remained unaffected in AnxA5-KO cells (Appendix Fig. S1I).

Next, we asked whether elevated mitochondrial volume and branching induced changes in the cristae structure, which may impact mitochondrial Ca²⁺ uptake. Therefore, we explored the mitochondrial ultrastructure in both WT and AnxA5-KO cells by evaluating the density and distribution of the cristae membrane within individual mitochondria using transmission electron microscopy (TEM) (Fig. 2I). The density of cristae membrane, as assessed by the ratio of cristae membrane length to mitochondrial length and area, remained unchanged in AnxA5-KO cells (Fig. 2J,K). Subsequently, we determined the distribution of cristae membrane within individual mitochondria by analyzing consecutive circular segments of the mitochondrial structure. (shown in Fig. 2L). Depletion of AnxA5 led to a distinct increase in cristae membrane density (ρCM), concentrated explicitly at the center of the mitochondria (Fig. 2M). These findings highlight AnxA5's pivotal role in regulating mitochondrial architecture likely via suppressing mitochondrial Ca²⁺ signaling.

## AnxA5 localizes at the OMM and within mitochondria

The association of AnxA5 with mitochondria (Sun et al, 1993) and its localization within cardiolipin-rich regions have been previously reported (Megli et al, 1995, 2000b). Given that AnxA5-KO cells exhibit reduced mitochondrial $[Ca^{2+}]_{Matrix}$ homeostasis, we investigated whether AnxA5 regulates mitochondrial Ca²⁺ signaling at the OMM or the IMM. To address this question, we initially revealed the localization of AnxA5 by a subcellular fractionation assay. Cytosolic and crude mitochondrial fractions were isolated, and the crude mitochondria fraction was purified through percoll gradient centrifugation (Wieckowski et al, 2009). The immunoblot results revealed a considerable AnxA5 localization in pure mitochondrial fractions relative to the cytosol (Fig. EV2A).

We further subjected the pure mitochondrial fractions to proteinase K (PK) treatment to selectively degrade the proteins in the OMM's cytosolic leaflet. Both TOM20 and VDAC1 are localized in the OMM. PK digests TOM20's cytosolic protrusion,

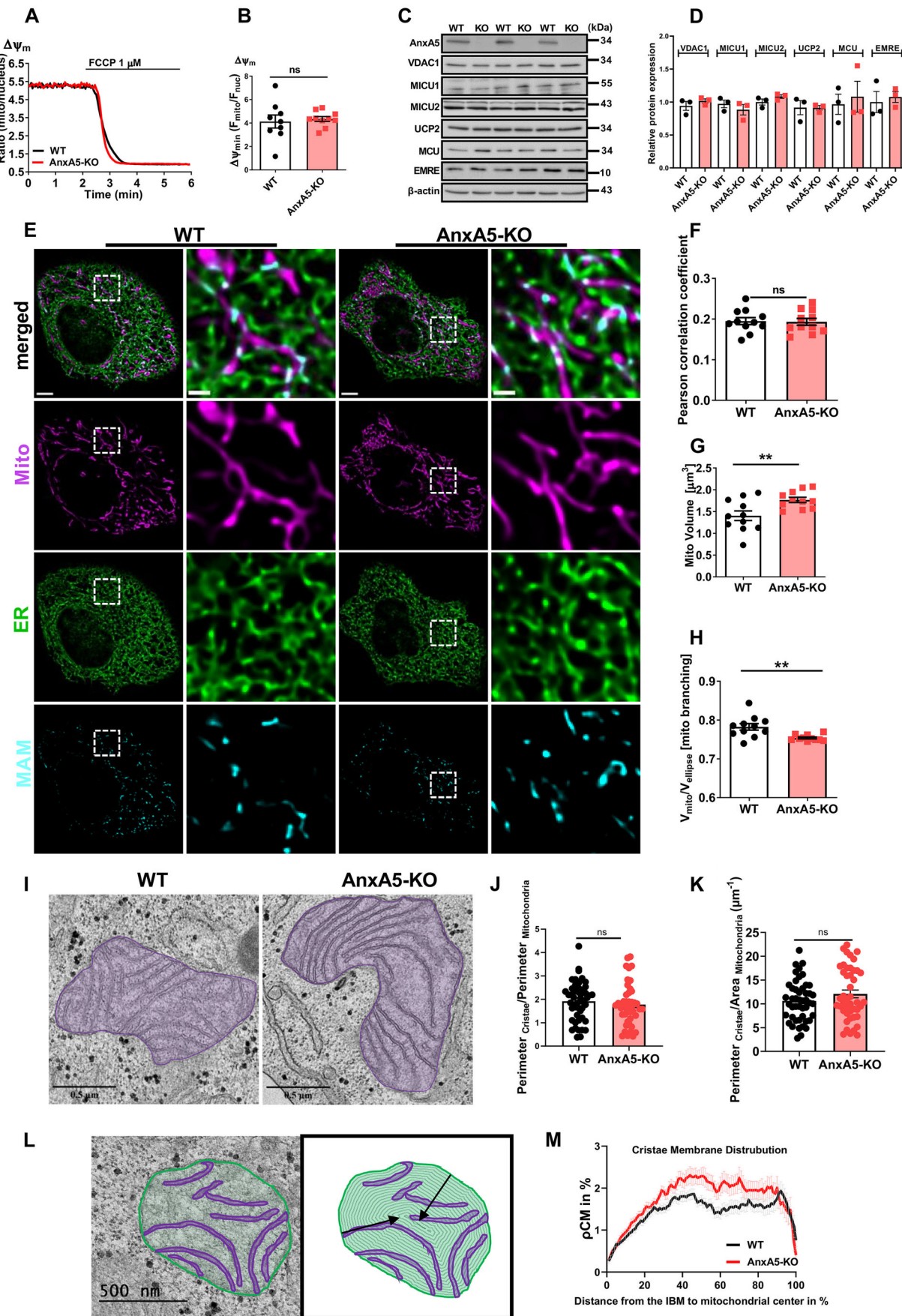

◄ **Figure 2. AnxA5 regulates mitochondrial ultrastructure.**

(A) Representative time courses of Ψmito before and after 1 µM FCCP treatment in WT (black) and AnxA5-KO (red) HeLa cells stained with TMRM. (B) Bar graph shows the relative mitochondrial membrane potential in WT (black) and AnxA5-KO (red) HeLa cells. Data points represent the mean ± SEM ($n_{WT} = 87/9$; $n_{AnxA5-KO} = 106/10$). (C) Immunoblots show the expression level of AnxA5, VDAC1, MICU1, MICU2, UCP2, MCU, and EMRE in WT and AnxA5-KO HeLa cell lysates. Uncropped blots are provided in the Source data. (D) Bar graph shows the expression level analysis of the respective proteins. Data points represent the mean ± SEM ($n_{WT} = 3$; $n_{AnxA5-KO} = 3$). (E) Representative confocal images of WT and AnxA5-KO HeLa cells were captured, stained with MTR-CMX (magenta), and expressed ERAT4.03 NA (green). Multiplying the fluorescence signals of mitochondria and ER at the pixel level, and subsequently amplifying the resulting signal, led to the visualization of MAM. Images provide an overview of the cells (scale bar = 10 µm), and magnified views of selected regions indicated by dashed squares (scale bar = 1 µm). (F) Bar graphs show the ER-mitochondrial co-localization represented by Pearson's R-value, (G) mitochondrial volume, and (H) mitochondrial branching (a lower value indicates more branched mitochondria). Data points represent the mean ± SEM ($n_{WT} = 91/11$; $n_{AnxA5-KO} = 88/11$). The p-values were calculated using a two-tailed unpaired Student's t-test for (F), $p = 0.8780$ (ns); for (G), $p = 0.0092$ (**$p < 0.01$); and for (H), $p = 0.0036$ (**$p < 0.01$). (I) Representative TEM images of mitochondria in WT and AnxA5-KO HeLa cells, where mitochondria are highlighted with magenta (scale bar = 500 nm). (J) Bar graphs show the cristae membrane-amount, measured by calculating the perimeter of cristae and normalizing it to the corresponding perimeter of mitochondria, and (K) cristae density was calculated by normalizing the perimeter of cristae to the area of mitochondria. Data points represent the mean ± SEM ($n_{WT} = 47/2$; $n_{AnxA5-KO} = 45/2$), n represents the number of mitochondria/biological replicates. (L) Schematic illustration depicts the spatial cristae membrane density (ρCM) measurement within segmented mitochondria. Iterative measurements of ρCM density were conducted in gradually downsized circular segments. (M) shows the spatial ρCM distribution in WT and AnxA5-KO cells by using the methods depicted in (L). The x-axis scale ranges from 0 to 100, with 0 indicating the outermost shell of the mitochondrion and 100 representing the mitochondrial center. Data points represent the mean ± SEM ($n_{WT} = 44/3$; $n_{AnxA5-KO} = 47/3$). n represents the number of cells/biological replicates (minimum of 3 independent experiments). Significant differences were assessed with the unpaired Student's t-test or Kolmogorov–Smirnov test (*$p < 0.05$, **$p < 0.01$, and ns: not significant). Source data are available online for this figure.

removing its band, while VDAC1, fully embedded in the OMM remain protected, indicating effective PK digestion and mitochondrial integrity (Fig. 3A). Interestingly, PK treatment partially digested AnxA5 (Fig. 3A), revealing the minority of AnxA5 being within pure mitochondrial fractions while the majority is in the cytosolic leaflet of the OMM (Fig. EV2B). Because an immunocytochemical staining and fluorescent protein-labeled AnxA5 produced an overwhelming cytosolic signal that masked AnxA5's distinct distribution within the cells, we decided to test the localization of immunogold labeled AnxA5 with TEM. Immunogold particles were detected in the cytosol, OMM, and within the mitochondria (Fig. 3B). Quantification of gold particles within mitochondria was significantly higher when the AnxA5 antibody was used compared to the rabbit antibody, indicating the specificity of the AnxA5 labeling (Fig. EV2C).

## AnxA5 depletion reduces IP₃-mediated Ca²⁺ transfer from the ER into the IMS

After identifying the localization of AnxA5 at the OMM, we monitored the Ca²⁺ concentration in the intermembrane space (IMS) ($[Ca^{2+}]_{IMS}$) levels using our genetically encoded IMS-targeted ratiometric Ca²⁺ sensor MICU1$^{1-140}$-GEMGeCO1 (Waldeck-Weiermair et al, 2019). The resting $[Ca^{2+}]_{IMS}$ levels were comparable in WT and AnxA5-KO in HeLa and EA.hy926 cells (Fig. EV2D,E). After stimulating the ER Ca²⁺ release, the increase in $[Ca^{2+}]_{IMS}$ was significantly reduced in AnxA5-KO HeLa cells (Fig. 3C,D) and AnxA5-KO EA.hy926 cells (Fig. 3E,F). These results suggest that AnxA5 regulates mitochondrial Ca²⁺ signaling at the OMM.

The differential regulation of mitochondrial Ca²⁺ uptake according to the source of the supplied Ca²⁺ has been frequently demonstrated (Collins et al, 2001; Waldeck-Weiermair et al, 2010). Therefore, a comprehensive protocol to monitor IMS Ca²⁺ dynamics by using three distinct Ca²⁺ sources was used: (1) ER-Ca²⁺ leak, evaluated through BHQ treatment; (2) IP₃-mediated ER-Ca²⁺ release, induced by histamine stimulation; and (3) store-operated Ca²⁺ entry (SOCE), initiated by the re-addition of extracellular Ca²⁺ after the depletion of ER-Ca²⁺ stores (Fig. 3G).

Notably, BHQ induced-ER-Ca²⁺ leak led to a slight, comparable $[Ca^{2+}]_{IMS}$ rise in WT and AnxA5-KO cells (Fig. 3H). The IMS Ca²⁺ elevation in response to a subsequent histamine-induced ER-Ca²⁺ was strongly reduced in AnxA5-KO cells (Fig. 3I). In contrast, mitochondrial IMS Ca²⁺ elevation in response to SOCE were similar in WT and AnxA5-KO cells (Fig. 3J). These findings indicate that AnxA5 is specifically involved in the mechanism of Ca²⁺ flux through the OMM in response to ER-Ca²⁺ release.

After establishing the role of AnxA5 in IMS Ca²⁺ homeostasis following ER-Ca²⁺ release, we explored whether AnxA5's involvement depends on a certain Ca²⁺ level. By subjecting cells to histamine stimulation at various concentrations, we induced different levels of ER-Ca²⁺ release (low, moderate, and high) in the absence of extracellular Ca²⁺. Compared to WT, AnxA5-KO cells exhibited reduced $[Ca^{2+}]_{IMS}$ signaling across all applied histamine concentrations, while $[Ca^{2+}]_{cyto}$ remained identical to WT (Fig. EV2F,G). The concentration-response curve for histamine-induced Ca²⁺ elevations revealed a notable difference in the half-maximal effective concentration ($EC_{50}$) for IMS Ca²⁺ signaling, with a value of 1.3 (1–1.6) µM observed in WT, while AnxA5-KO cells showed a shifted $EC_{50}$ of 4.8 (2.7–7.8) µM (Fig. EV2H). In contrast, the $EC_{50}$ values for the increase in cytosolic Ca²⁺ remained unchanged between WT at 2 (1.7–2.3) µM and AnxA5-KO cells at 2 (1.7–2.5) µM (Fig. EV2I).

Next, we investigated whether the Ca²⁺ binding and self-assembly properties of AnxA5 are involved in the regulation of IMS Ca²⁺ signaling. To check this, we used previously characterized AnxA5 mutants that disrupt Ca²⁺ binding through mutations in the Ca²⁺ binding domain of AnxA5, AnxA5-2Mt (D144N, E228Q) (Jin et al, 2004), AnxA5-3Mt (D144N, E228Q, D303N) (Jin et al, 2004). Moreover, a mutant with disrupted AnxA5's self-assembly function while preserving its Ca²⁺ binding ability, AnxA5-5Mt (R18E, R25E, K29E, K58E, K193E) (Bouter et al, 2011), was tested (Fig. 3K). Transient expression of AnxA5 and AnxA5-5Mt in AnxA5-KO cells rescued IMS Ca²⁺ signaling, whereas AnxA5-2Mt and AnxA5-3Mt failed to rescue IMS Ca²⁺ signaling in AnxA5-KO cells (Figs. 3L,M and EV2J). These findings suggest that the Ca²⁺ binding of AnxA5 is essential for regulating IMS Ca²⁺ signaling, whereas the self-assembly of AnxA5 is not required.

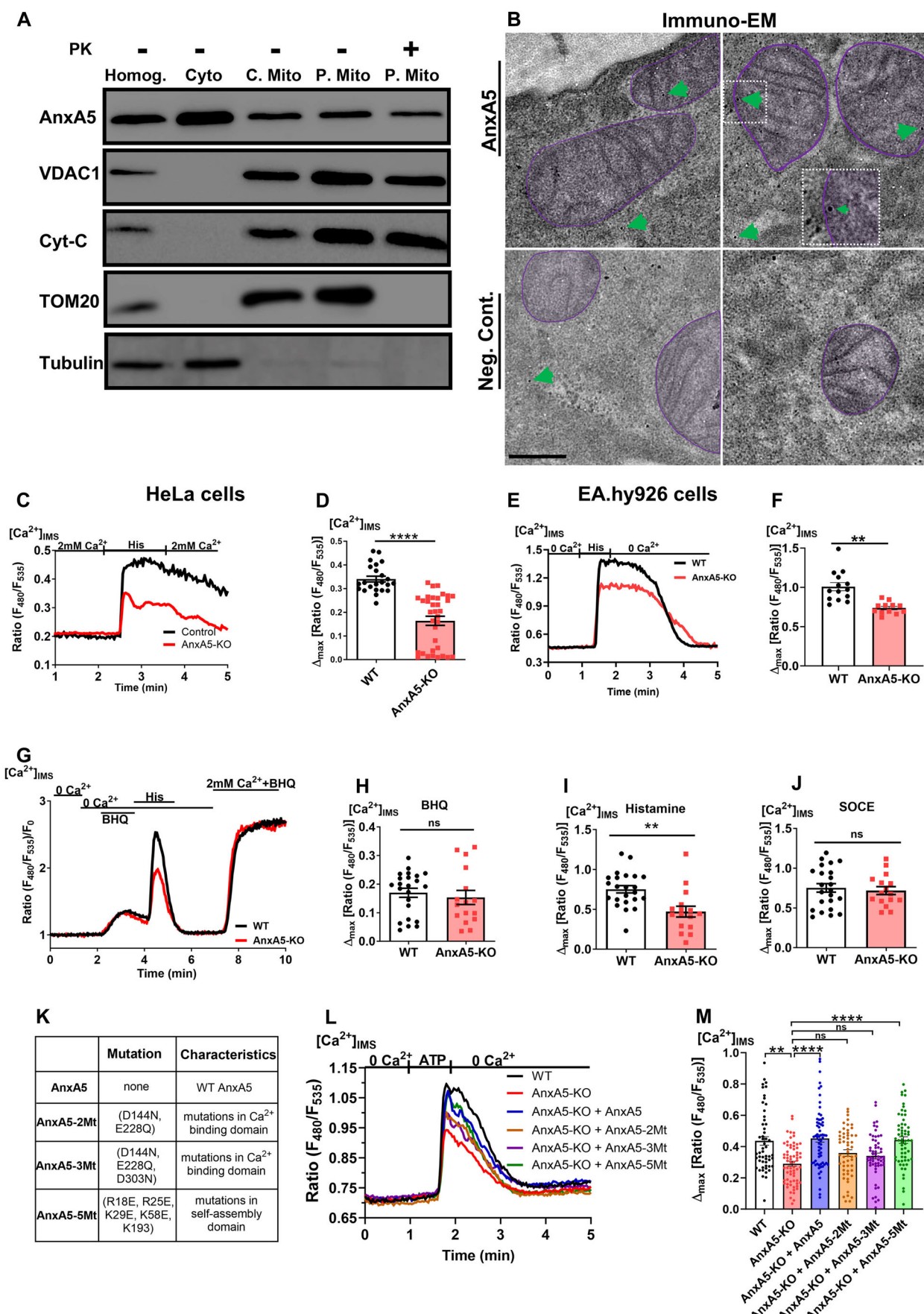

Figure 3. AnxA5 selectively modulates Ca$^{2+}$ signaling on the OMM depending on the source of Ca$^{2+}$.

(A) Representative immunoblots show protein components of subcellular fractions including Homogenate (Homog), cytosol (Cyto), crude mitochondria (C. Mito), and pure mitochondria (P. Mito) obtained from HeLa cells. Marker proteins indicate OMM (VDAC1), the cytosolic leaflet of the OMM (TOM20), IMS (cytochrome-C, Cyt-C), and cytosol (tubulin). To digest proteins localized on the cytosolic leaflet of the OMM, the pure mitochondrial fraction was incubated with 50 µg/ml Proteinase K (PK) for 15 min ($n = 3$). Uncropped blots are provided in the Source Data. (B) Representative Immuno-electron micrographs of mitochondria from HeLa cells. In the upper panel, individual gold particles indicate the localization of the AnxA5, while the lower panel serves as a negative control where the primary antibody was omitted. Mitochondria were highlighted with magenta and individual gold particles are marked with green arrows (scale bar = 250 nm). (C) Representative time courses of the 100 µM histamine-induced [Ca$^{2+}$]$_{IMS}$ responses in WT (black) and AnxA5-KO (red) in HeLa cells measured in Ca$^{2+}$ (2 mM) containing buffer. (D) Bar graphs show the histamine-induced maximum [Ca$^{2+}$]$_{IMS}$ levels in WT (black) and AnxA5-KO (red). Data points represent the mean ± SEM ($n_{WT} = 23/7$; $n_{AnxA5-KO} = 34/8$). The $p$-value was calculated using the two-tailed Kolmogorov–Smirnov test, yielding $p < 0.0001$ (****$p < 0.0001$). (E) Mean time courses of the 100 µM histamine-induced [Ca$^{2+}$]$_{IMS}$ responses of WT (black) and AnxA5-KO (red) EA.hy926 cells measured in Ca$^{2+}$-free buffer (containing 100 µM of EGTA). (F) Bar graph shows the histamine-induced maximum [Ca$^{2+}$]$_{IMS}$ levels in WT (black) and AnxA5-KO (red) cells. Data points represent the mean ± SEM ($n_{WT} = 14/6$; $n_{AnxA5-KO} = 12/6$). The $p$-value was calculated using a two-tailed unpaired Student's t-test, $p = 0.0034$ (**$p < 0.01$). (G) Mean time courses of 15 µM BHQ and 100 µM histamine-induced [Ca$^{2+}$]$_{IMS}$ responses in the absence of extracellular Ca$^{2+}$ and upon subsequent store-operated Ca$^{2+}$ entry in WT (black) and AnxA5-KO (red) HeLa cells. (H) Bar graphs show the BHQ-, (I) histamine-, and (J) SOCE-induced maximum [Ca$^{2+}$]$_{IMS}$ responses in WT (black) and AnxA5-KO (red) cells. Data points represent the mean ± SEM ($n_{WT} = 23/5$; $n_{AnxA5-KO} = 16/4$). The $p$-value for (I) was calculated using a two-tailed unpaired t-test yielding $p = 0.001$ (**$p < 0.01$). (K) Table displaying the list of AnxA5 mutation sites and the characteristics of the mutations. L Average time courses of 100 µM ATP-induced [Ca$^{2+}$]$_{IMS}$ responses were measured in HeLa cells under Ca$^{2+}$-free conditions (containing 100 µM of EGTA). Studied cell groups: WT, AnxA5-KO, AnxA5-KO + AnxA5 (WT AnxA5), AnxA5-KO + AnxA5-2Mt (mutations in the Ca$^{2+}$-binding domain: D144N, E228Q), AnxA5-KO + AnxA5-3Mt (mutations in the Ca$^{2+}$-binding domain: D144N, E228Q, D303N), AnxA5-KO + AnxA5-5Mt (mutations in the self-assembly domain: R18E, R25E, K29E, K58E, K193). Cells were transfected with either an empty plasmid in WT and AnxA5-KO cells or the respective WT or mutant version of AnxA5 in AnxA5-KO cells. (M) Bar graphs show the ATP-induced maximum [Ca$^{2+}$]$_{IMS}$ levels in WT (black), AnxA5-KO (red), AnxA5-KO + AnxA5 (blue), AnxA5-KO + AnxA5-2Mt (orange), AnxA5-KO + AnxA5-3Mt (magenta), AnxA5-KO + AnxA5-5Mt (green). Data points represent the mean ± SEM ($n_{WT} = 53/10$; $n_{AnxA5-KO} = 67/12$; $n_{Rescue} = 64/11$; $n_{AnxA5-KO+2Mt} = 55/11$; $n_{AnxA5-KO+3Mt} = 58/11$; $n_{AnxA5-KO+5Mt} = 60/12$). $n$ represents the number of cells/biological replicates (minimum of 3 independent experiments). The $p$-values were calculated using a Kruskal–Wallis test, from left to right: $p = 0.0011$ (**$p < 0.01$), $p < 0.0001$ (****$p < 0.0001$), $p = 0.1488$ (ns), $p = 0.9038$ (ns), $p < 0.0001$ (****$p < 0.0001$). Significant differences were assessed using either one-way ANOVA with Kruskal–Wallis test (**$p < 0.01$, ****$p < 0.0001$ and ns: not significant) and with the unpaired Student's t-test or Kolmogorov–Smirnov test (**$p < 0.01$, ****$p < 0.0001$ and ns: not significant). Source data are available online for this figure.

## AnxA5 by modulating IMS Ca$^{2+}$ signaling, contributes to the dynamicity of the IMM

In IMS, Ca$^{2+}$ binds to MICU1, the gatekeeper of MCU, triggering a conformational change (Wang et al, 2014) that induces its de-oligomerization. To investigate the impact of AnxA5-mediated IMS Ca$^{2+}$ homeostasis on MICU1 rearrangements, we used the MICU1-FRET sensor (Waldeck-Weiermair et al, 2015) (Fig. 4A). In WT cells, ER-Ca$^{2+}$ release resulted in a significant reduction in the MICU1-FRET ratio (Fig. 4B). However, in AnxA5-KO cells, the decrease in the FRET ratio was markedly reduced, indicating the impact of AnxA5 on the level of MICU1 dimerization as a consequence of the reduced IMS Ca$^2$ signaling (Fig. 4B,C).

Previously, we reported that IP$_3$-induced Ca$^{2+}$ rise in IMS activates MICU1 dimerization, leading to spatial opening of the cristae junction (Gottschalk et al, 2018) consequently decelerating the dynamics of the mitochondrial cristae membrane in MERCs (Gottschalk et al, 2022) (Fig. 4D). Therefore, we used structured illumination microscopy (SIM) to investigate the dynamics of cristae membrane in the whole mitochondria and the MERCs regions of WT and AnxA5-KO cells. Upon ER-Ca$^{2+}$ release, cristae membrane dynamics were unaffected throughout the entire mitochondria in both WT and AnxA5-KO cells (Fig. EV3A). However, the dynamics of cristae membrane in the vicinity of MERCs were significantly reduced in WT, while no change was observed in AnxA5-KO cells (Fig. 4E). Notably, the transient expression of AnxA5 in KO cells led to a similar effect to that observed in WT cells (Fig. 4E). These findings highlight the crucial role of AnxA5-facilitated IMS Ca$^{2+}$ signaling in regulating the dynamics of cristae junctions and membrane kinetics within MERCs. Furthermore, we investigated whether reducing cristae junction openings in AnxA5-KO cells leads to hampered Ca$^{2+}$ transfer to the cristae. Therefore, we used our genetically encoded Ca$^{2+}$ sensor explicitly localized within the cristae (Waldeck-Weiermair et al, 2019). Compared to WT, IP$_3$-induced Ca$^{2+}$ release significantly reduced [Ca$^{2+}$]$_{cristae}$ signaling in AnxA5-KO cells (Fig. 4F,G) while maintaining identical basal levels (Fig. EV3B).

It has been demonstrated that histamine-induced cristae junction openings redistribute MCU from the cristae lumen to inner boundary membrane (IBM) (Gottschalk et al, 2019). To study the role of AnxA5-mediated IMS Ca$^{2+}$ signaling in MCU shuttling, we transfected cells with MCU-mCherry (localized in the IMM) and MICU1-YFP (localized in the IBM), to dynamically follow MCU redistribution in response to histamine (Fig. 4H). In WT cells, ER-Ca$^{2+}$ release resulted in a significant transfer of MCU towards the IBM, while this process was hindered in AnxA5-KO cells (Fig. 4I). Furthermore, histamine stimulation led to measurable changes in mitochondrial area and aspect ratio after 90 s, with WT cells exhibiting a significant reduction in both parameters. In contrast, AnxA5-KO cells showed no change in mitochondrial area and a slight reduction in aspect ratio after histamine stimulation (Fig. EV3C,D). These findings indicate that AnxA5-mediated IMS Ca$^{2+}$ signaling regulates MCU redistribution from cristae lumen to IBM during cristae junction openings and is essential for mitochondrial morphology remodeling during ER-Ca$^{2+}$ release.

## Annexin-A5 is in the proximity of VDAC1 and is fundamental for the Ca$^{2+}$ permeability of VDAC1 and VDAC2

Having established the regulatory role of AnxA5 in IMS Ca$^{2+}$ signaling, we further investigated its potential interaction and/or localization close to VDAC1. We did not detect physical interaction

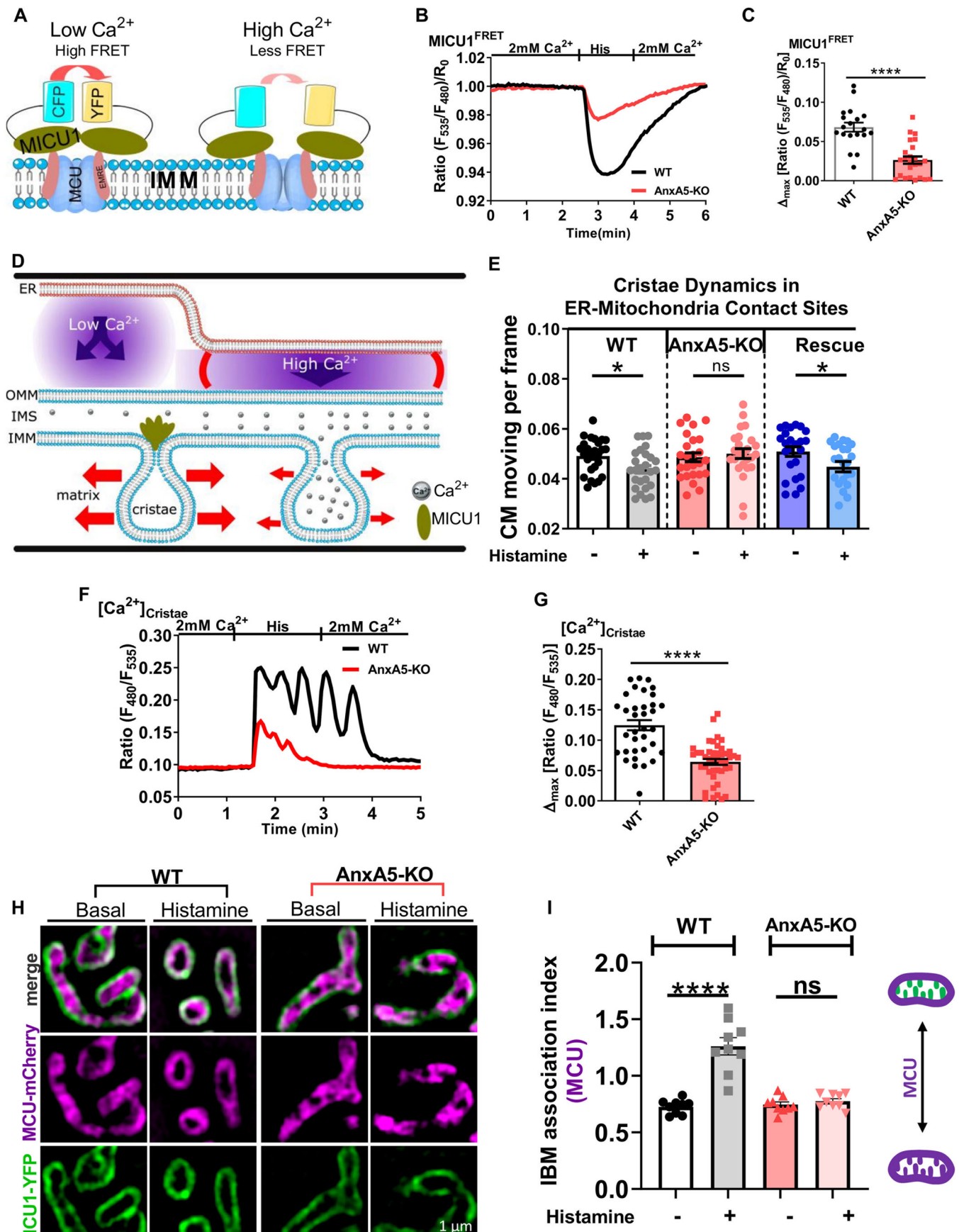

**Figure 4. AnxA5-mediated IMS Ca²⁺ signaling maintains cristae Ca²⁺, membrane dynamics, and MCU redistribution.**

(A) The schematic illustration depicts the histamine-induced rearrangement of MICU1-CFP and MICU1-YFP FRET, which serve as $[Ca^{2+}]_{IMS}$ sensors. (B) Mean time course of 100 μM histamine-induced MICU1 FRET ratio in WT (black) and AnxA5-KO (red) HeLa cells measured in Ca²⁺ (2 mM) containing buffer. (C) Bar graphs show the histamine-induced change in the MICU1$^{FRET}$ ratio in WT (black) and AnxA5-KO (red) cells. Data points represent the mean ± SEM ($n_{WT}$ = 19/6; $n_{AnxA5-KO}$ = 24/6). The p-value for (C) was calculated using a two-tailed unpaired Student's t-test, $p < 0.0001$ (****$p < 0.0001$). (D) The schematic illustration of cristae membrane dynamics under basal (low) and upon IP₃-induced $[Ca^{2+}]_{ER}$ release (high Ca²⁺). (E) Bar graph shows cristae membrane movements per frame in MERCs under basal and upon IP₃-induced $[Ca^{2+}]_{ER}$ release in WT (black), AnxA5-KO (red), and Rescue (blue) HeLa cells. Data points represent the mean ± SEM ($n_{WT}$ = 26/6; $n_{AnxA5-KO}$ = 25/6; $n_{Rescue}$ = 24/6). The p-values were calculated using a two-tailed unpaired Student's t-test, from left to right: $p = 0.0124$ (*$p < 0.05$), $p = 0.5845$ (ns), and $p = 0.0360$ (*$p < 0.05$). (F) Representative time course of 100 μM histamine-induced $[Ca^{2+}]_{cristae}$ levels in WT (black) and AnxA5-KO (red) HeLa cells measured in Ca²⁺ (2 mM) containing buffer. (G) Bar graph shows the histamine-induced maximum $[Ca^{2+}]_{cristae}$ levels in WT (black) and AnxA5-KO (red) cells. Data points represent the mean ± SEM ($n_{WT}$ = 35/6; $n_{AnxA5-KO}$ = 44/6). The p-value was calculated using a two-tailed unpaired Student's t-test, $p < 0.0001$ (****$p < 0.0001$). (H) Representative SIM images of HeLa cells depict the expression of MICU1-YFP (green) and MCU-mcherry (magenta) in WT (left panel) and AnxA5-KO (right panel) before and 90 s after histamine stimulation. (I) Bar graph shows the IBM association index of MCU in WT and AnxA5-KO before and 90 s after $[Ca^{2+}]_{ER}$ release. Higher IBM association index values correspond to an increased translocation of MCU from cristae to IBM, as illustrated in the schematic. Data points represent the mean ± SEM ($n_{WT}$ = 84/9; $n_{AnxA5-KO}$ = 88/9). n represents the number of cells/biological replicates (minimum of 3 independent experiments). The p-values were calculated using a one-way ANOVA with Tukey's multiple comparison test, from left to right: $p < 0.0001$ (****$p < 0.0001$) and $p = 0.9586$ (ns). Significant differences were assessed using either one-way ANOVA with Tukey's multiple comparison tests (****$p < 0.0001$ and ns: not significant) or with the unpaired Student's t-test (*$p < 0.05$, ****$p < 0.0001$ and ns: not significant). Source data are available online for this figure.

between AnxA5 and VDAC1 using the co-immunoprecipitation method (Appendix Fig. S2A). However, our experiments using the in-situ proximity ligation assay (PLA), which enables visualizing proximity of two proteins that are in less than 30 nm distance (Söderberg et al, 2006), pointed to localization of AnxA5 close to VDAC1 (Fig. 5A). Notably, knockout of AnxA5 abolished the PLA signal. As a control, VDAC1 knockdown in WT cells resulted in a substantial 70% reduction in PLA signals and a 60% decrease in protein levels, confirming the method's specificity (Fig. 5A,B), (Appendix Fig. S2B,C). The VDAC1-IP₃R interaction served as a positive control in this assay (Appendix Fig. S2D,E). These results indicate a proximity of AnxA5 and VDAC1.

Next, we investigated whether the proximity of AnxA5 to VDAC1 contributes to mitochondrial Ca²⁺ signaling. To test this, we knocked down VDAC1 in both WT and AnxA5-KO cells and measured mitochondrial matrix Ca²⁺ signaling following IP₃-induced ER Ca²⁺ release (Fig. 5C). While VDAC1 knockdown significantly reduced mitochondrial Ca²⁺ levels in WT cells, it did not cause further reduction in AnxA-5KO cells, indicating mutual interdependence between AnxA5 and VDAC1 in mitochondrial Ca²⁺ signaling (Fig. 5C,D).

Subsequently, we explored whether AnxA5 regulates the Ca²⁺ permeability of the other VDAC isoforms VDAC 2 and VDAC 3 of those the expression levels are much lower than VDAC1 (Appendix Fig. S2F). We transiently expressed FLAG-tagged VDAC2 and VDAC3 in siVDAC1-treated WT and AnxA5-KO cells. Mitochondrial localization was confirmed by high-resolution fluorescence microscopy (Appendix Fig. S2G). Subsequently, we tested whether these isoforms could rescue the lack of VDAC1 in terms of mitochondrial Ca²⁺ signaling. Transient expression of VDAC2, but not VDAC3, rescued the reduced mitochondrial Ca²⁺ signaling caused by VDAC1 knockdown in WT cells (Fig. 5C,D). This isoform-specific difference indicates that only VDAC2 mimics VDAC1 regarding Ca²⁺ transit through the OMM, possibly due to the co-localization of VDAC1 and VDAC2, whereas VDAC3 displays a distinct spatial distribution (Neumann et al, 2010). Notably, the rescuing effect of VDAC2 on IMS Ca²⁺ in VDAC1-depleted cells critically depended on the presence of AnxA5 (Fig. 5C,D). These data suggest that AnxA5 regulates the Ca²⁺ permeability of VDAC1 and VDAC2, while VDAC3 does not mimic VDAC1 in terms of its Ca²⁺ function in our conditions.

## AnxA5 accumulates in the OMM upon ER Ca²⁺ release

AnxA5 serves as an intracellular Ca²⁺ sensor, capable of detecting increases in Ca²⁺ levels and effectively tracking high Ca²⁺ microdomains to bind negatively charged phospholipids (Monastyrskaya et al, 2007). Therefore, we investigated whether IP₃-induced ER Ca²⁺ release, that is known to generate local Ca²⁺ hotspots of up to 16.42 μM in the MERCs (Giacomello et al, 2010), leads to the accumulation of AnxA5 in mitochondrial membranes. To address this, we employed cryo-fixation to avoid potential artefacts caused by a disturbance of ion homeostasis during chemical fixation, combined with immunogold labeling of AnxA5 to track its localization with TEM under basal conditions and 20 s after ER-Ca²⁺ release. While this method is more time-consuming, it bypasses the limitation of immunofluorescence labeling caused by the intense cytosolic signal obscuring the visualization of AnxA5 localization on mitochondria. In both conditions, we observed gold particles within the mitochondria and in the cytosol (Fig. 5E). Interestingly, ER Ca²⁺ release resulted in the accumulation of gold particles in the 20 nm vicinity of the OMM from both the cytosolic and the IMS leaflet of the OMM (indicated with a red rectangle) (Fig. 5E,F; Appendix Fig. S2H). Consequently, we propose a model where ER-Ca²⁺ release dynamically triggers the Ca²⁺-dependent accumulation of AnxA5 to the OMM (Fig. 5F,G).

## AnxA5 is fundamental for the occurrence and the open probability of an OMM Ca²⁺ current

To further elucidate whether the hampered IMS Ca²⁺ signaling in AnxA5-KO cells was due to reduced Ca²⁺ permeability of the OMM, single-channel currents of intact mitochondria, isolated from WT and AnxA5-KO cells were investigated by employing the mitochondria-attached configuration of the patch clamp technique. By delivering hyperpolarizing voltage steps from the holding potential 0 mV with the 20 mV increment, we were able to identify a single Ca²⁺ channel activity of the inward currents within the test potentials ranging from −60 to −120 mV in the presence of 10 μM free Ca²⁺ and 140 mM K⁺ in the pipette. These channels displayed a single-channel conductance of 35 pS and an increased probability of single-channel opening (Po) as a function of voltage (Fig. EV4A,B). In the absence of Ca²⁺ in the

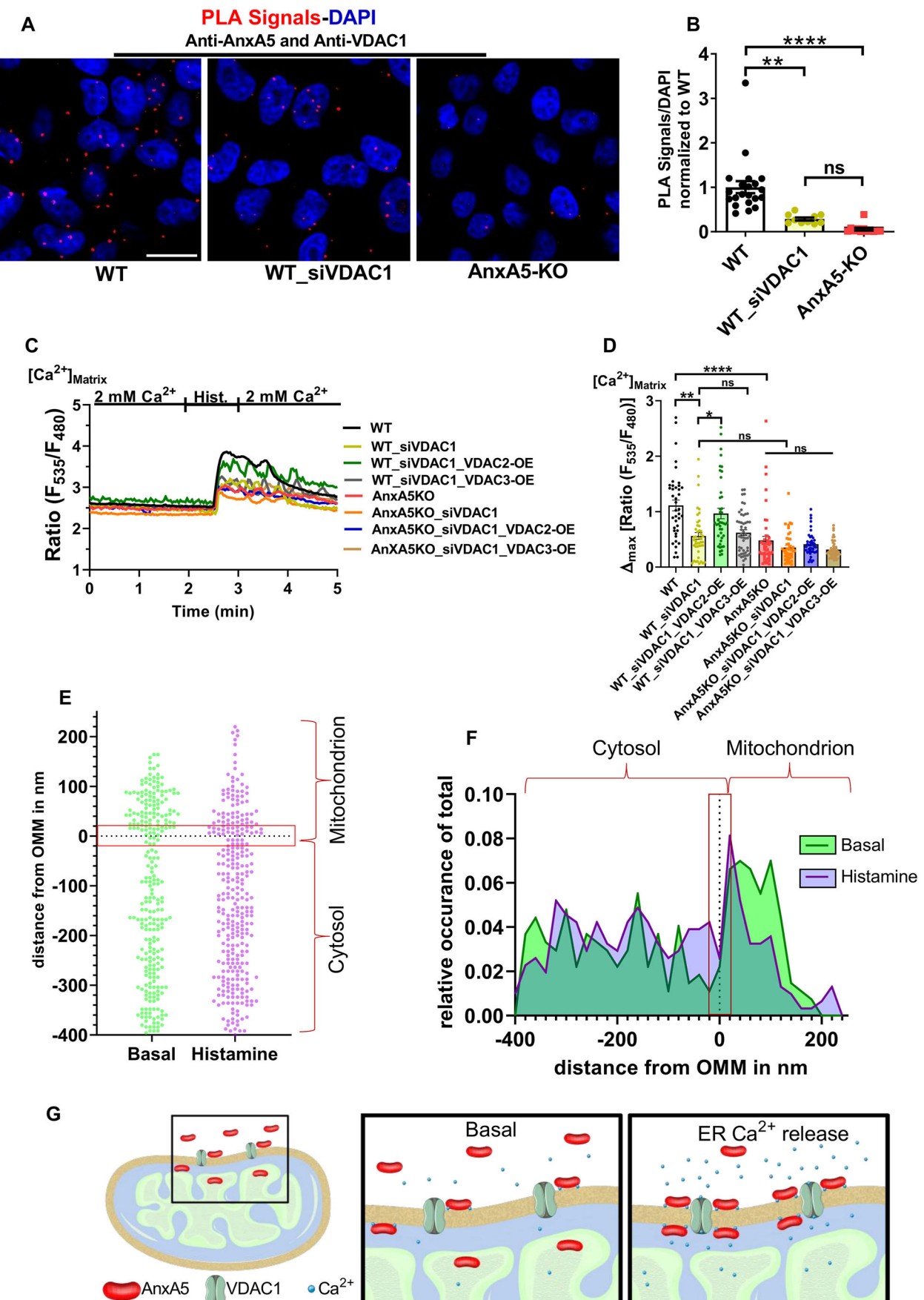

**Figure 5. AnxA5 localizes in the vicinity of VDAC1 and ER Ca²⁺ release triggers its accumulation on the OMM.**

(A) Representative image of PLA assay indicating the protein-protein proximity between AnxA5 and VDAC1 in siNeg- and siVDAC1-treated WT, and in AnxA5-KO HeLa cells (scale bar = 20 μm). (B) The bar graph shows the ratio of total PLA signals to the number of cells. Data points represent the mean ± SEM ($n_{siNeg} = 4$; $n_{siVDAC1} = 3$; $n_{AnxA5-KO} = 4$). The p-values were calculated using a Kruskal–Wallis test, from left to right: $p = 0.0025$ (**$p < 0.01$), $p < 0.0001$ (****$p < 0.0001$), and $p = 0.2271$ (ns). (C) Representative time courses of 100 μM histamine-induced $[Ca^{2+}]_{Matrix}$ responses in WT_siNeg, AnxA5-KO_siNeg, WT_siVDAC1, AnxA5-KO_siVDAC1, WT_siVDAC1 + VDAC2-FLAG, AnxA5-KO_siVDAC1 + VDAC2-FLAG, WT_siVDAC1 + VDAC3-FLAG, and AnxA5-KO_siVDAC1 + VDAC3-FLAG were measured in a Ca²⁺ (2 mM)-containing buffer. (D) Bar graphs show the histamine-induced maximum $[Ca^{2+}]_{Matrix}$ levels in WT_siNeg (black), AnxA5-KO_siNeg (red), WT_siVDAC1 (dark yellow), AnxA5-KO_siVDAC1 (orange), WT_siVDAC1 + VDAC2-FLAG (green), AnxA5-KO_siVDAC1 + VDAC2-FLAG (blue), WT_siVDAC1 + VDAC3-FLAG (gray), and AnxA5-KO_siVDAC1 + VDAC3-FLAG (brown). Data points represent the mean ± SEM ($n_{WT\_siNeg} = 40/9$; $n_{AnxA5\_KO\_siNeg} = 47/9$; $n_{WT\_siVDAC1} = 44/9$; $n_{AnxA5\_KO\_siVDAC1} = 43/7$; $n_{WT\_siVDAC1+VDAC2-FLAG} = 42/6$; $n_{AnxA5\_KO\_siVDAC1+VDAC2-FLAG} = 43/8$; $n_{WT\_siVDAC1+VDAC3-FLAG} = 50/7$; $n_{AnxA5\_KO\_siVDAC1+VDAC3-FLAG} = 48/8$). The p-values were calculated using a Kruskal–Wallis test, from left to right (bottom to top): $p = 0.0014$ (**$p < 0.01$), $p < 0.0001$ (****$p < 0.0001$), $p = 0.1353$ (ns), $p = 0.0401$, *$p < 0.05$), $p > 0.9999$ (ns) and $p > 0.9999$ (ns). (E) The distribution of AnxA5-labeled gold particles within the whole cell was analyzed, showing their localization on mitochondria and cytosol under basal conditions (green) and 20 s after histamine stimulation (magenta). Positive values indicate the distance of the gold particles from the OMM to the mitochondrial matrix side, while negative values represent the distance from the OMM to the cytosol. Data points represent the subcellular localization of the gold particles ($n_{WT-Basal} = 10/3$; $n_{WT-Histamine} = 10/3$). (F) The relative occurrence of the gold particles was calculated based on the data presented in Fig. 5 (E). n represents the number of cells/biological replicates (minimum of 3 independent experiments). (G) The schematic illustration depicts the basal localization and histamine-induced translocation of AnxA5 to OMM. Significant differences were assessed using one-way ANOVA with Kruskal–Wallis test (*$p < 0.05$, **$p < 0.01$, ****$p < 0.0001$ and ns: not significant). Source data are available online for this figure.

pipette, voltage steps within the same voltage range failed to elicit any single-channel activity (Fig. EV4C). Notably, we observed the 35 pS channel activity in mitochondria isolated from WT and AnxA5-KO HeLa cells (Fig. 6A). However, our findings showed decreased channel occurrence in mitochondria of AnxA5-KO cells compared to WT mitochondria (Fig. 6B). Addition of recombinant AnxA5 to patch pipette rescued channel occurrence in AnxA5-KO mitochondria, suggesting that AnxA5 promotes the recruitment of silent channels in the OMM (Fig. 6B). Furthermore, the knockdown of VDAC1 in WT mitochondria slightly reduced the 35 pS channel occurrence and the knockout of AnxA5 further reduced the occurrence (Fig. 6B).

To investigate the effect of AnxA5 on transmitochondrial Ca²⁺ fluxes further, we compared the Po values at the test potential −80 mV (Fig. 6C). Depletion of AnxA5 significantly reduced the Po of the 35 pS channel, and AnxA5 addition to AnxA5-KO mitochondria rescued the effect (Fig. 6A–C). VDAC1 depletion resulted in an apparent reduction in the channel's Po (Fig. 6B,C). Notably, the knockout of AnxA5 did not cause a further decrease in Po in VDAC1 knockdown cells. While these data suggest that VDAC1 influences the biophysical properties of the 35 pS Ca²⁺ channel that, by itself, also essentially depends on AnxA5, a fundamental contribution of VDAC1 to the observed channel cannot be conclusively determined, possibly due to the modest knockdown efficiency. Collectively, these results indicate that AnxA5 contributes to the appearance and activity of the 35 pS Ca²⁺ permeable channel in the OMM.

## AnxA5's presence in the VDAC1 microenvironment mitigates cisplatin-induced VDAC1 dimerization and safeguards against cell death

Numerous studies demonstrated that various apoptotic stimuli including cisplatin, induce an increase in intracellular Ca²⁺ levels in HeLa cells (Weisthal et al, 2014; Keinan et al, 2013). To investigate the role of AnxA5 on the impact of cisplatin on subcellular Ca²⁺ rise, we monitored mitochondrial and cytosolic Ca²⁺ levels in WT and AnxA5-KO cells. After 12 h of cisplatin treatment (5 and 10 μM), we observed a strongly reduced elevation in the mitochondrial $[Ca^{2+}]_{Matrix}$ and $[Ca^{2+}]_{IMS}$ levels in AnxA5-KO cells compared with that observed in WT HeLa (Fig. 7A,B). Moreover, the administration of cisplatin resulted in a comparable slight

increase in $[Ca^{2+}]_{Cyto}$ levels in both WT and AnxA5-KO cells (Fig. 7C), suggesting that cisplatin-induced Ca²⁺ elevations predominantly target the mitochondria. Next, we questioned the effect of mitochondrial Ca²⁺ overload on cell viability and apoptosis. Therefore, we used flow cytometry and in situ live-cell imaging with Annexin5-FITC/PI staining. Cisplatin induces cell death from 12–14 h onward (Appendix Fig. S3A). Consequently, at 24 and 48 h, AnxA5-KO cells exhibited more cell death (14–18%) compared to WT in a concentration-dependent manner (Fig. 7D; Appendix Fig. S2B). Furthermore, AnxA5-KO cells exhibited increased susceptibility to cisplatin-induced cell death through apoptosis (Fig. 7E; Appendix Fig. S3C–E). These findings indicate that despite the knockout of AnxA5 reduced mitochondrial Ca²⁺ accumulation in response to cisplatin treatment, AnxA5-KO cells exhibit enhanced susceptibility for cisplatin-induced apoptosis. Ca²⁺-dependent apoptotic stimuli induce the oligomerization of VDAC1, a process linked to mitochondrial-mediated apoptosis (Keinan et al, 2013). However, inhibition of VDAC1 oligomerization with a novel compound, VBIT-4, sheltered cells from apoptosis (Ben-Hail et al, 2016). To pinpoint whether the increased susceptibility of AnxA5-KO cells to cisplatin treatment depends on differences in the VDAC1 oligomerization, we treated the cells with cisplatin in the presence and absence of VBIT-4. Co-treatment with cisplatin and VBIT-4 effectively inhibited cell death (Fig. 7F) and apoptosis (Fig. 7G) in WT and AnxA5-KO cells (Appendix Fig. S3F–I), suggesting that the sensitivity of AnxA5-KO cells to cisplatin-induced cell death is based on cisplatin-induced VDAC1 oligomerization.

Finally, we conducted cross-linking and immunoblotting assays to examine potential differences in cisplatin-induced VDAC1 dimerization between WT and AnxA5-KO cells at different time points. Although cisplatin-induced VDAC1 dimerization was undetectable at 12 h, probably due to the absence of observed cell death (Appendix Fig. S3A,J), 24 h of treatment resulted in a 3.2-fold increase in WT cells and a 5.2-fold increase in AnxA5-KO cells compared to the DMSO-treated WT (Fig. EV5A,B). After 48 h, WT cells had an 8.4-fold increase, and AnxA5-KO cells had a 9.9-fold increase in VDAC1 dimeric level, indicating that cisplatin induces a higher VDAC1 dimerization in AnxA5-KO cells (Fig. 7H,I). Co-administration of cisplatin and VBIT-4 significantly

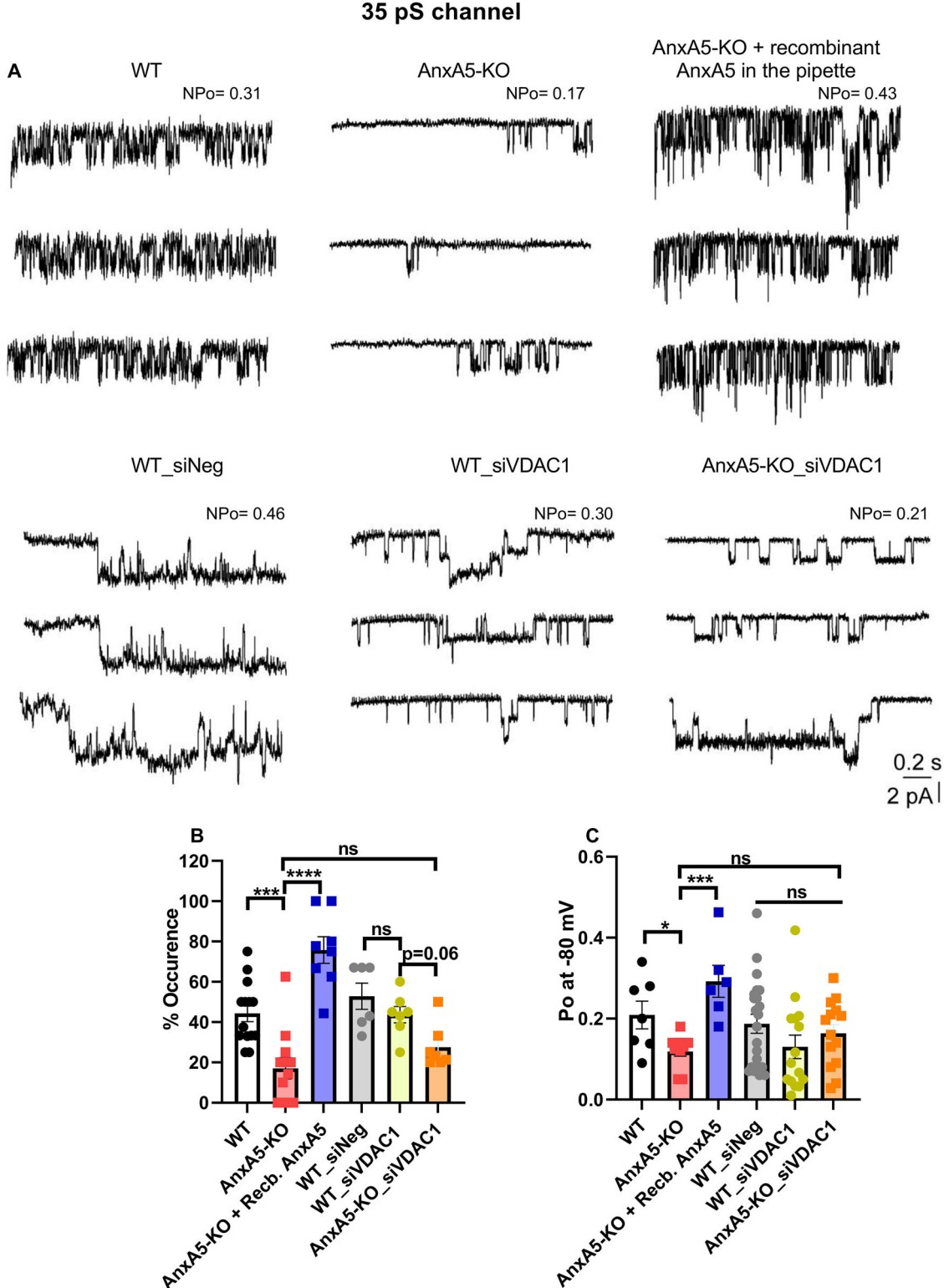

reduced VDAC1 dimerization in both WT and AnxA5-KO cells (Figs. 7H,I and EV5A,B). Moreover, we investigated whether the enhanced apoptotic cell death and VDAC1 dimerization observed in AnxA5-KO cells is specific for cisplatin. To address this point, we tested selenite, another well-known inducer of apoptosis via

VDAC1 dimerization (Ben-Hail et al, 2016). Upon 48 h of selenite treatment, WT cells exhibited a 4-fold increase in dimeric VDAC1 compared to DMSO-treated WT controls, while AnxA5-KO cells showed an 8.5-fold increase (Fig. EV5C,D). Hence, selenite induced more cell death in AnxA5-KO cells (Fig. EV5E). Notably, co-

**Figure 6.   Effect of AnxA5 on the single-channel activity of the OMM.**

(A) Representative single-channel traces showing the 35 pS channel in intact mitochondria isolated from the WT, AnxA5KO, AnxA5-KO mitochondria supplemented with recombinant AnxA5 (200 ng/ml added to the patch pipette), WT_siNeg, WT_siVDAC1, and AnxA5KO_siVDAC1. (B) Bar graphs show channel occurrence, calculated as the percentage of patches showing single-channel activity relative to the total number of patches on a given experimental day in WT (black), AnxA5-KO (red), AnxA5-KO mitochondria supplemented with recombinant AnxA5 (blue), WT_siNeg (gray), WT_siVDAC1 (dark yellow) and AnxA5KO_siVDAC1 (orange). Data points represent the mean ± SEM ($n_{WT} = 14$; $n_{AnxA5-KO} = 12$; $n_{AnxA5-KO + Recombinant\ AnxA5} = 8$; $n_{WT\_siNeg} = 6$; $n_{WT\_siVDAC1} = 7$; $n_{AnxA5-KO\_siVDAC1} = 7$). The p-values, from left to right: $p = 0.0009$ (***$p < 0.001$), $p < 0.0001$ (****$p < 0.0001$), $p = 0.4726$ (ns), $p = 0.4094$ (ns) and $p = 0.066$ (ns). (C) Bar graphs show NPo at −80 mV in WT (black), AnxA5-KO (red), AnxA5-KO mitochondria supplemented with recombinant AnxA5 (blue), WT_siNeg (gray), WT_siVDAC1 (dark yellow), and AnxA5KO_siVDAC1 (orange). Data points represent the mean ± SEM ($n_{WT} = 7$; $n_{AnxA5-KO} = 13$; $n_{AnxA5-KO + Recombinant\ AnxA5} = 6$; $n_{WT\_siNeg} = 22$; $n_{WT\_siVDAC1} = 15$; $n_{AnxA5-KO\_siVDAC1} = 14$). The p-values, from left to right: $p = 0.0327$ (*$p < 0.05$), $p = 0.0002$ (***$p < 0.001$), $p = 0.6267$ (ns) and $p = 0.2877$ (ns). n represents the number of cells/biological replicates (minimum of 3 independent experiments). Significant differences were assessed using one-way ANOVA with Tukey's multiple comparison tests (*$p < 0.05$, ***$p < 0.001$, ****$p < 0.0001$ and ns: not significant). Source data are available online for this figure.

treatment with VBIT4 effectively reduced dimerization levels and prevented cell death (Fig. EV5C–E).

Additionally, as an alternative method, we transiently expressed tetracysteine-tagged VDAC1 (Pilic et al, 2024) in WT and AnxA5-KO cells and visualized the cells 12 h after cisplatin and cisplatin + VBIT-4 co-treatment (Fig. EV5F). We observed clustered VDAC1 in WT and AnxA5-KO cells without cisplatin treatment. Consistent with immunoblot experiments, cisplatin-induced larger VDAC1 clusters in AnxA5-KO cells and VBIT-4 reduced the cluster size (Fig. EV5F,G). These results support our previous immunoblot results that AnxA5 localization near to VDAC1 microenvironment acts as a protective barrier, controlling the level of cisplatin/selenite-induced VDAC1 dimerization, and subsequently regulates the cell's susceptibility to apoptosis (Fig. 8).

## Discussion

Previous studies have provided evidence for the involvement of AnxA5 in intracellular $Ca^{2+}$ signaling (Gerke & Moss, 2002; Kubista et al, 1999). Nevertheless, our understanding of the molecular function of AnxA5 in contribution to mitochondrial $Ca^{2+}$ homeostasis remains limited. Therefore, we investigated the role of AnxA5 in mitochondrial $Ca^{2+}$ signaling. We found a regulatory role of AnxA5 in the VDAC1 established $Ca^{2+}$ permeability upon ER $Ca^{2+}$ release. Hence, by preventing excessive VDAC1 oligomerization, AnxA5 safeguards the cells from cisplatin/selenite-induced apoptosis.

This paper provides a comprehensive analysis using dynamic $Ca^{2+}$ measurements in the mitochondrial matrix of HeLa, EA.hy926, and isolated perivascular cells from AnxA5-KO mice, and the rescue experiments by AnxA5 in AnxA5-KO HeLa cells have provided compelling evidence for the vital role of AnxA5 in mitochondrial but not cytosolic $Ca^{2+}$ signaling upon IP₃-generating agonists. Importantly, our data using sensors targeted to the mitochondrial matrix and the IMS further revealed that AnxA5 is fundamental for both mitochondrial compartments, thus pointing to a dominant role of AnxA5 in the $Ca^{2+}$ transit through the OMM. Our research further revealed that AnxA5 knockout cells maintain normal $\Psi_{mito}$, mitochondrial morphology, and mitochondrial-ER tethering, thus indicating that the observed changes in mitochondrial $Ca^{2+}$ signaling occur independently from changes in mitochondrial structure, their membrane potential, or the tethering between the ER and the mitochondria.

We applied electron micrographs after cryo-fixation and immunogold labeling to decipher the location of AnxA5 before and after ER $Ca^{2+}$ release and proved the presence of AnxA5's at the OMM and within mitochondria and revealed a further accumulation of AnxA5 at the OMM following ER $Ca^{2+}$ release. The latter findings were expected as the high $Ca^{2+}$ microdomains, reaching up to 16 μM (Giacomello et al, 2010) within the MERCs, should be sufficient to induce the dynamic binding of AnxA5 to the OMM at this location. However, the presence of AnxA5 on the OMM of isolated mitochondria that were typically isolated using EGTA-containing buffers points either to a partial $Ca^{2+}$-independent mitochondrial localization of AnxA5 at the OMM or to a very slow dissociation of AnxA5 from the OMM after its $Ca^{2+}$-triggered membrane accumulation.

Our findings that in AnxA5-KO cells, MICU1 dimerization and cristae junction openings are reduced, further highlight a role of AnxA5 in governing IMS $Ca^{2+}$ signaling and an active participation in the dynamic process of mitochondrial architecture modeling. These data are in agreement with reports showing that the IMS $Ca^{2+}$ levels play a pivotal role in regulating the mitochondrial ultrastructure by triggering the process of MICU1 dimerization, which subsequently leads to the opening of cristae junction and ultimately causes mitochondrial fission (Gottschalk et al, 2019, 2022, 2018). Finally, our data that ER $Ca^{2+}$ release induces mitochondrial fission exclusively in WT but not in AnxA5-KO cells completes our analysis, revealing an active contribution of AnxA5 to the regulation of the $Ca^{2+}$ transit through the OMM into the IMS upon ER $Ca^{2+}$ release.

The fluorimetric $Ca^{2+}$ measurements and structural evaluations, were complemented by electrophysiological experiments of single $Ca^{2+}$ channels in the OMM of freshly isolated mitochondria applying the mitochondria-attached configuration. Interestingly, these experiments revealed a 35 pS conductance $Ca^{2+}$ channel, while we did not observe a high conductance single-channel activity described in previous studies using reconstituted VDAC1 in artificial bilayers or vesicles (Pavlov et al, 2005). Nevertheless, the lack of high-conductance single channels in this study is consistent with earlier studies on intact mice liver mitochondria, which reported a 30 pS $Ca^{2+}$ channel but no high conductance in the OMM under similar experimental conditions to ours (Catia Sorgato et al, 2008; Moran et al, 1992). Such discrepancy between the $Ca^{2+}$ conductances in the OMM of freshly isolated mitochondria and that found in artificial lipid bilayers with the exclusive presence of VDAC1, may be due to the differences in the environment of

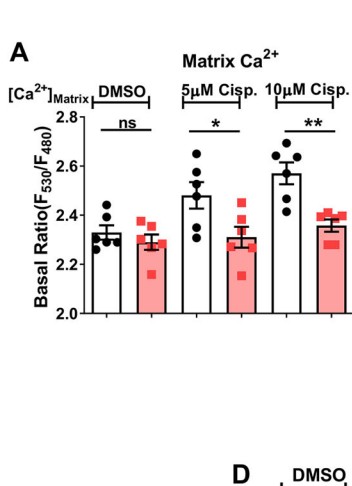

**A** Matrix Ca²⁺

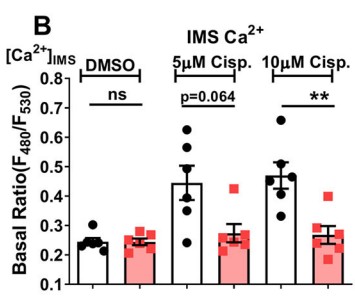

**B** IMS Ca²⁺

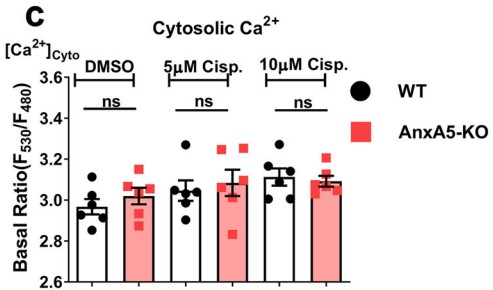

**C** Cytosolic Ca²⁺

● WT
■ AnxA5-KO

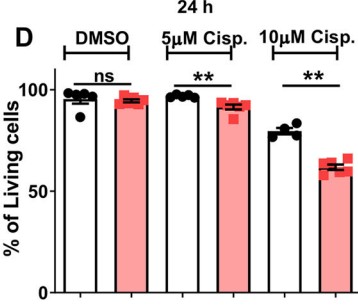

**D** 24 h

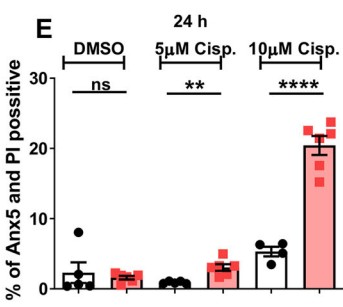

**E** 24 h

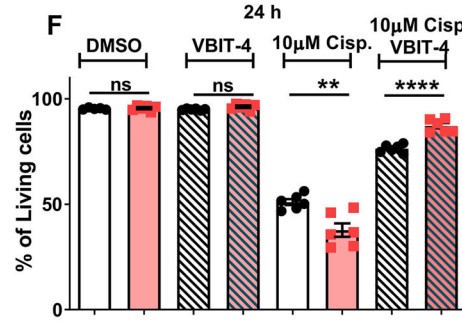

**F** 24 h

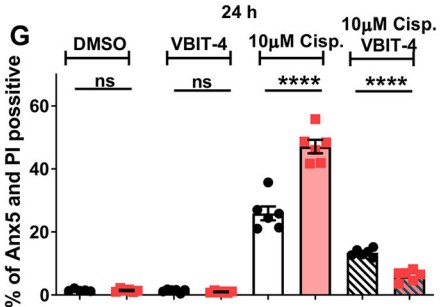

**G** 24 h

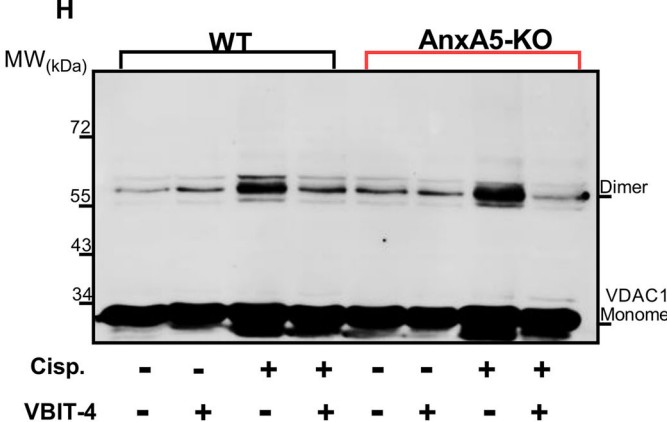

**H**

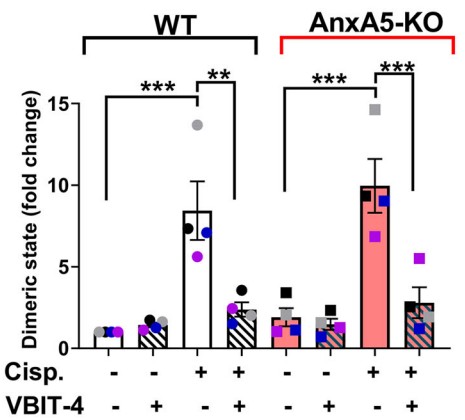

**I**

**Figure 7.   AnxA5 controls cisplatin-induced VDAC1 dimerization.**

(A) Bar graphs show the basal $[Ca^{2+}]_{Matrix}$, (B) $[Ca^{2+}]_{IMS}$, and (C) $[Ca^{2+}]_{Cyto}$ level in WT (black) and AnxA5-KO (red) cells upon 12 h DMSO, 5 μM and 10 μM cisplatin treatment. Data points represent the mean ± SEM for $[Ca^{2+}]_{Matrix}$ ($n_{WT-DMSO} = 63/6$; $n_{AnxA5-KO-DMSO} = 75/6$, $n_{WT-5μM-cisplatin} = 67/6$; $n_{AnxA5-KO-5μM-cisplatin} = 71/6$, $n_{WT-10μM-cisplatin} = 53/6$; $n_{AnxA5-KO-10μM-cisplatin} = 54/6$), and for $[Ca^{2+}]_{IMS}$ ($n_{WT-DMSO} = 64/6$; $n_{AnxA5-KO-DMSO} = 66/6$, $n_{WT-5μM-cisplatin} = 60/6$; $n_{AnxA5-KO-5μM-cisplatin} = 74/6$, $n_{WT-10μM-cisplatin} = 64/6$; $n_{AnxA5-KO-10μM-cisplatin} = 63/6$), and $[Ca^{2+}]_{Cyto}$ ($n_{WT-DMSO} = 64/6$; $n_{AnxA5-KO-DMSO} = 68/6$, $n_{WT-5μM-cisplatin} = 74/6$; $n_{AnxA5-KO-5μM-cisplatin} = 84/6$, $n_{WT-10μM-cisplatin} = 55/6$; $n_{AnxA5-KO-10μM-cisplatin} = 83/6$). The $p$-values for (A), from left to right, are: $p = 0.3780$ (ns), $p = 0.0320$ (*$p < 0.05$), and $p = 0.0022$ (**$p < 0.01$); for (B): $p = 0.9716$ (ns), $p = 0.0640$ (ns), and $p = 0.0039$ (**$p < 0.01$). (D) Bar graphs show the percentage of living cells and (E) late apoptosis, assessed by Annexin5-FITC/PI staining and FACS analysis in WT (black) and AnxA5-KO (red) cells upon 24 h DMSO, 5 μM and 10 μM cisplatin treatment. (F) Bar graphs show the percentage of living cells and (G) late apoptosis in WT (black) and AnxA5-KO (red) cells upon 24 h DMSO, 20 μM VBIT-4, 10 μM cisplatin, and cisplatin+VBIT-4 treatment. Data points represent the mean ± SEM ($n_{WT-DMSO} = 5$; $n_{AnxA5-KO-DMSO} = 6$, $n_{WT-20μM-VBIT-4} = 6$; $n_{AnxA5-KO-20μM-VBIT-4} = 6$, $n_{WT-10μM-cisplatin} = 6$; $n_{AnxA5-KO-10μM-cisplatin} = 6$, $n_{WT-cisplatin+VBIT-4} = 6$; $n_{AnxA5-KO-cisplatin+VBIT-4} = 6$). The $p$-values for (D), from left to right, are: $p = 0.1775$ (ns), $p = 0.0043$ (**$p < 0.01$), and $p = 0.0095$ (**$p < 0.01$); for (E): $p = 0.6040$ (ns), $p = 0.0028$ (**$p < 0.01$), and $p < 0.0001$ (****$p < 0.0001$); for (F): $p = 0.6787$ (ns), $p = 0.0582$ (ns), and $p = 0.0037$ (**$p < 0.01$), and $p < 0.0001$ (****$p < 0.0001$); for (G): $p = 0.8063$ (ns), $p = 0.1840$ (ns), $p < 0.0001$ (****$p < 0.0001$), and $p < 0.0001$ (****$p < 0.0001$). (H) Representative immunoblot shows monomeric and dimeric VDAC1 levels. Uncropped blots are provided in the Source Data. (I) Bar graph shows the quantification of the immunoblot in WT (black) and AnxA5-KO (red) cells upon 48 h DMSO, 20 μM VBIT-4, 10 μM cisplatin, and cisplatin+VBIT-4 treatment (each color represents the experiments from the same day). Data points represent the mean ± SEM ($n_{WT-all} = 4$; $n_{AnxA5-KO-all} = 4$). The $p$-values from left to right, are: $p = 0.0003$ (***$p < 0.001$), $p = 0.0037$ (**$p < 0.01$), and $p = 0.0001$ (***$p < 0.001$), and $p = 0.0005$ (***$p < 0.001$). $n$ represents the number of cells/biological replicates (minimum of 3 independent experiments). Significant differences were assessed using either one-way ANOVA with Tukey's multiple comparison tests (**$p < 0.01$, ***$p < 0.001$ and ns: not significant) and with the two-tailed unpaired Student's t-test or Kolmogorov–Smirnov test (*$p < 0.05$, **$p < 0.01$, ****$p < 0.0001$ and ns: not significant). Source data are available online for this figure.

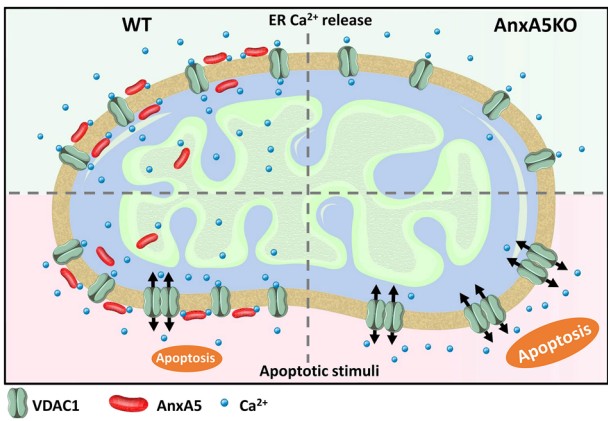

**Figure 8.   Schematic depiction of AnxA5-modulated VDAC1 dimerization in response to cisplatin treatment.**

AnxA5 regulates mitochondrial $Ca^{2+}$ signaling upon ER $Ca^{2+}$ release (upper panel). Apoptotic stimuli induce VDAC1 dimerization and apoptotic cell death (lower panel). Anx5 regulates the dimerization state by localizing in the VDAC1 microenvironment, thus affecting the level of apoptosis (lower left panel). In AnxA5-KO cells, cisplatin/selenite induces enhanced VDAC1 dimerization, leading to elevated apoptosis (lower right panel).

VDAC1, like neighboring/interacting proteins, and/or the different lipid composition in the two experimental models.

Our present experiments revealed that AnxA5 had no impact on the biophysical characteristics (conductance and gating) of the 35 pS $Ca^{2+}$ channel. This observation, and the findings that in mitochondria isolated from AnxA5-KO cells, the channel Po and occurrence were strongly reduced, and could be rescued by recombinant AnxA5 in the pipette, exclude the contribution of AnxA5 as a pore-forming component and points to AnxA5 as a positive regulator of the $Ca^{2+}$ permeable state of VDAC1 (Báthori et al, 2006). In its $Ca^{2+}$-permeable state, VDAC1 allows the $Ca^{2+}$ flux across the OMM and, thus, establishes IMS $Ca^{2+}$ signaling (Shoshan-Barmatz et al, 2017; Waldeck-Weiermair et al, 2019). The proximity between AnxA5 and VDAC1, as demonstrated by the proximity ligation assays shown, suggests that AnxA5 localizes in

the vicinity of the VDAC1 while facilitating its $Ca^{2+}$ permeable state. This assumption is supported by our co-immunoprecipitation experiments where no direct interaction between AnxA5 and VDAC1 was found. Although these data do not entirely rule out a transient interaction between the two proteins, we consider an alternative function of AnxA5 in facilitating VDAC1 $Ca^{2+}$ permeability. In this context, the binding of AnxA5 to negatively charged phospholipids in the presence of high $Ca^{2+}$ deserves consideration (Bouter et al, 2015; Monastyrskaya et al, 2007). The presence of negatively charged phospholipids such as phosphatidylinositol, phosphatidylserine, and cardiolipin in the OMM (Ardail et al, 1990) establishes a lipid microenvironment that promotes the binding of AnxA5 to those regions in the presence of high $Ca^{2+}$ levels (Monastyrskaya et al, 2007). At present, we do not know which specific phospholipid(s) in the OMM AnxA5 binds. Considering the number of various phospholipids and their possible combinations, the task to figure this out appears endless. However, based on the present data, we propose that in the presence of high $Ca^{2+}$ (e.g., IP₃-induced ER $Ca^{2+}$ release) AnxA5 binds to the negatively charged phospholipid microdomain that already harbors VDAC1, thus stabilizing the singular $Ca^{2+}$ permeable state of VDAC1. Such regulation of ion channels by AnxA5 binding to membranes has been discussed (Gerke & Moss, 2002; Kubista et al, 1999).

Despite the 75% sequence similarity among the three VDAC isoforms (Naghdi & Hajnóczky, 2016), only VDAC2 but not VDAC3 could rescue the IMS $Ca^{2+}$ signal in VDAC1-depleted cells. AnxA5 also regulates the $Ca^{2+}$ permeability of VDAC2. This suggests that VDAC1 and VDAC2 may coexist in a lipid environment enriched with AnxA5, which is distinct from that of VDAC3 (Neumann et al, 2010).

Our findings using cisplatin further supported the stabilization of the singular $Ca^{2+}$-permeable state of VDAC1 by AnxA5, which has been shown to translocate to mitochondria upon cisplatin treatment (Jeong et al, 2014). In line with previous reports (Xu et al, 2015), cisplatin elevated $Ca^{2+}$ levels in the mitochondrial matrix and IMS of WT HeLa cells but not in AnxA5-KO cells, emphasizing the crucial role of AnxA5 in preventing cisplatin-induced mitochondrial $Ca^{2+}$ elevation. On the other hand, AnxA5-

KO cells displayed enhanced susceptibility to cisplatin-induced apoptotic cell death, suggesting that $Ca^{2+}$ alone does not solely govern apoptosis. These findings point to an unknown role of AnxA5 in protecting against cisplatin-induced apoptosis. Previous studies showed that cisplatin-induced VDAC1 oligomerization triggers the release of pro-apoptotic proteins into the cytosol, initiating mitochondria-mediated apoptosis (Keinan et al, 2013; Shoshan-Barmatz et al, 2020). Notably, inhibition of VDAC1 oligomerization by VBIT-4 prevents apoptosis and cell death in several cell lines (Ben-Hail et al, 2016). In line with these studies, in our work, co-treatment of cisplatin and VBIT-4 successfully counteracted cell death in both WT and AnxA5-KO cells, thus providing evidence that the increased cell death observed in AnxA5-KO cells may rely on the level of VDAC1 oligomerization. Our experiments further revealed that compared to WT, AnxA5-KO cells exhibited an app. 60% and 20% increase in VDAC1 dimeric levels after 24 and 48 h of cisplatin treatment, respectively. In accordance with these data, in AnxA5-KO cells, selenite treatment induced an app. 110% increase in VDAC1 dimeric levels compared to WT cells. These data highlight a regulatory role of AnxA5 in VDAC1 oligomerization in response to different apoptotic stimuli, possibly by its capability to self-assemble into 2D arrays to reduce the lateral diffusion of phospholipids (Pigault et al, 1994). On the other hand, VDAC1 is capable of organizing lipids around itself, leading to the formation of specific assemblies regulating its oligomerization (Lafargue et al, 2024), a process that might be shaped by AnxA5. Notably, because AnxA5 may regulate VDAC1 dimerization through its phospholipid-binding/stabilization function, VBIT-4 inhibits dimerization by directly binding to VDAC1 (Ben-Hail et al, 2016). Accordingly, our data allows us to suggest that the localization of AnxA5 at the OMM plays a crucial role in stabilizing the lipid microenvironment of singular $Ca^{2+}$ permeable VDAC1, while effectively preventing excessive VDAC1 dimerization upon apoptotic stimuli (Fig. 8).

Contrary to our findings, a previous study (Jeong et al, 2014) showed that AnxA5 facilitates VDAC1 oligomerization upon cisplatin treatment, enhancing the mitochondrial apoptotic pathway. However, in the same study and unlike our present work, AnxA5 depletion alone significantly decreased the expression levels of VDAC1. Cisplatin-induced VDAC1 dimerization was reduced by 60% in AnxA5-depleted cells, whereas AnxA5 depletion alone resulted in a 70% reduction in basal VDAC1 dimer levels (Jeong et al, 2014). Therefore, reduced VDAC1 dimerization upon cisplatin treatment in AnxA5-depleted cells may result from decreased VDAC1 expression rather than the direct effect of AnxA5. Notably, the expression level of VDAC1 regulates the equilibrium between monomeric and oligomeric states upon cisplatin treatment, thereby influencing cell death (Arif et al, 2014; Abu-Hamad et al, 2006). Because we show that AnxA5-KO cells exhibit enhanced VDAC1 oligomerization without affecting VDAC1 expression, we suggest that in our model system, AnxA5 serves as a protective factor against cisplatin-induced apoptosis by stabilizing VDAC1 and preventing excessive VDAC1 dimerization.

Collectively, our data highlight the crucial role of AnxA5 in controlling $Ca^{2+}$ fluxes across the OMM by stabilizing the singular, $Ca^{2+}$ permeable state of VDAC1 upon ER $Ca^{2+}$ release, thus counteracting cisplatin-/selenite-induced VDAC1 dimerization and subsequent apoptosis.

# Methods

## Reagents and tools table

| Reagent/Resource | Reference or Source | Identifier or Catalog Number |
|---|---|---|
| **Experimental models** | | |
| HeLa S3 cells (*H. sapiens*) | ATCC | CCL-2.2 |
| EA.hy926 cells (*H.sapiens*) | Bauer et al (1992) (J Cell Physiol) | Dr. C.J.S. Edgell, University of North Carolina, NC, USA |
| Perivascular cells (*M. musculus*) | Brachvogel et al (2007) (Exp Cell Res) | Prof. Dr. Bent Brachvogel, University of Cologne, Germany |
| **Recombinant DNA** | | |
| AnxA5-2Mt (D144N, E228Q), PCDNA3.1 (-) | Gene Universal Inc. in Newark, USA | Custom-synthesized DNA construct |
| AnxA5-3Mt (D144N, E228Q, D303N), PCDNA3.1 (-) | Gene Universal Inc. in Newark, USA | Custom-synthesized DNA construct |
| AnxA5-5Mt (R18E, R25E, K29E, K58E, and K193E), PCDNA3.1 (-) | Gene Universal Inc. in Newark, USA | Custom-synthesized DNA construct |
| Control shRNA Plasmid | Santa Cruz Biotech | sc-108060 |
| AnxA5 shRNA Plasmid | Santa Cruz Biotech | sc-29686-SH |
| pGS-U6-gAnxA5 | Genscript | N/A |
| pcDNA3.3-Cas9-2A-eGFP | Genscript | N/A |
| VDAC2-FLAG | Neumann et al (2010) (Biophysics) | N/A |
| VDAC3-FLAG | Neumann et al (2010) (Biophysics) | N/A |
| VDAC1-TC | Pilic et al (2024) (PLOS One) | NA |
| MICU1-CFP, PCDNA3.1 (-) | Gottschalk et al (2019) (Nat Commun) | N/A |
| MICU1-YFP, PCDNA3.1 (-) | Gottschalk et al (2019) (Nat Commun) | N/A |
| ER-RFP, PCDNA3.1 (-) | Gottschalk et al (2019) (Nat Commun) | N/A |
| MCU-mCherry, PCDNA3.1 (-) | Gottschalk et al (2019) (Nat Commun) | N/A |
| MICU1 FRET, PCDNA3.1 (-) | Waldeck-Weiermair et al (2015) (Scientific Reports) | N/A |
| MICU1$^{1-140}$-GEMGeCO1, PCDNA3.1 (-) | Waldeck-Weiermair et al (2019) (Front Cell Neurosci) | N/A |
| ROMO-GEMGeCO1, PCDNA3.1 (-) | Waldeck-Weiermair et al (2019) (Front Cell Neurosci) | N/A |
| 4mtD3cpv, PCDNA3.1 (-) | Palmer et al (2006) (Chem Biol) | N/A |
| D1ER, PCDNA3.1 (-) | Palmer et al (2006) (Proc Natl Acad Sci USA) | N/A |
| jGCaMP7c | Dana et al (2019) (Nat Methods) | N/A |
| ERAT4.03 NA, PCDNA3.1 (-) | Imamura et al (2009) (Proc Natl Acad Sci USA) | N/A |
| SPLICS | Addgene | Plasmid #164107 |
| AKAP-RFP-CAAX | Csordás et al (2006) (J Cell Biol) | N/A |
| **Antibodies** | | |
| Annexin A5 antibody. mouse monoclonal. Reacts with: Mouse, Rat, Human. Dilution = 1:1000 | Santa Cruz | sc-74438 |
| Annexin A5 antibody. Rabbit polyclonal. Reacts with: Mouse, Rat, Human. Isotype: IgG Dilution = 1:1000 | Abcam | ab14196 |

| Reagent/Resource | Reference or Source | Identifier or Catalog Number |
|---|---|---|
| VDAC antibody. recombinant monoclonal mouse. Reacts with: Mouse, Rat, Human. Isotype: IgG2a. Dilution = 1:5000 | Abcam | ab186321 |
| Tubulin antibody. Mouse Monoclonal. Reacts with: Human, Mouse, Rat. Host Species Mouse ImmunogenT. Dilution = 1:1000 | BioLegend | 801202 |
| MICU1 antibody. Rabbit monoclonal. Reacts with: Human Mouse Rat Monkey. Source/Isotype Rabbit IgG. Dilution = 1:1000 | Cell Signaling Technology | D4P8Q |
| EMRE antibody. Rabbit polyclonal. Reacts with: Mouse, Human, Horse. Dilution = 1:1000 | SantaCruz | sc-86337 |
| MICU2 antibody. Rabbit polyclonal. Reacts with: Mouse, Human, Horse. Dilution = 1:1000 | Abcam | ab101465 |
| UCP2 antibody. Reacts with: human Mouse Rat Monkey. Source/Isotype Rabbit IgG Dilution = 1:1000 | Cell Signaling Technology | D105V |
| MCU antibody. Reacts with: Human Mouse Rat Monkey. Monoclonol. Source/Isotype Rabbit IgG. Dilution = 1:1000 | Cell Signaling Technology | D2Z3B |
| β-Actin monoclonal antibody produced in mouse. Reacts with: rabbit, sheep, cat, guinea pig, rat, human. Dilution = 1:1000 | Sigma | A5316 |
| Histone H3 antibody. Reactivity: Human Mouse Rat Monkey. Source/ Isotype Mouse IgG3. Dilution = 1:1000 | Cell Signaling Technology | 1B1B2 |
| Cytochrome C antibody. Reactivity: Human Mouse Rat. Source: Rabbit. Dilution = 1:500 | Cell signaling | 4272 |
| TOM20 antibody. Reactivity: Human Mouse Rat. Source: Rabbit IgG Dilution = 1:1000 | Cell Signaling | D8T4N |
| **Oligonucleotides and other sequence-based reagents** | | |
| Control siRNA (5′-3′) UUCUCCGAACGUGUCACGU | Microsynth | Waldeck-Weiermair et al (2019) (Front Cell Neurosci) |
| VDAC1 siRNA (5′-3′) ACACUAGGCACCGAGAUUA | Microsynth | Waldeck-Weiermair et al (2019) (Front Cell Neurosci) |
| AnxA9 siRNA (5′-3′) GCAGUCUACAAACACAAUUtt | Microsynth | Lu et al (2023) (Hum Cell) |
| AnxA8 siRNA (5′-3′) | Sigma-Aldrich | SASI_Hs01_00091429. |
| The gRNA sequence of AnxA5 (5′-3′) GATGCAGAAACTCTTCGGA | Genscript | This study |
| **Chemicals, Enzymes and other reagents** | | |
| DMEM | Sigma-Aldrich | D5523 |
| PolyJet™ | SignaGen Laboratories | SL100688 |
| TransFast | Promega | E2431 |
| RNeasy® Mini Kit | Qiagen | 74104 |
| Embedding resin | TAAB | C032 |
| Proximity ligation assay | Sigma-Aldrich | DUO92101 |
| Mounting medium | Immuno bioscience | AR-6500-01 |
| Pierce™ BCA Protein Assay Kit | ThermoFisher | 23225 |
| SuperSignal™ West Pico PLUS Chemiluminescent Substrate | ThermoFisher Scientific | A38554 |
| Proteinase K | Qiagen | RP103B |
| Ethylene glycol bis(succinimidyl succinate) | Thermo Fisher | 21565 |
| Cisplatin | Sigma | P4394 |
| Selenite | Sigma | S5261 |

| Reagent/Resource | Reference or Source | Identifier or Catalog Number |
|---|---|---|
| FlAsH-EDT$_2$ | Cayman chemical | 212118-77-9 |
| FITC Annexin V Apoptosis Detection Kit I | BD Biosciences | 556547 |
| Percoll | Sigma | P1644 |
| PMSF | Sigma | P7626 |
| Recombinant AnnexinA5 | Abcam | ab255714 |
| **Software** | | |
| Graphpad Prism 9.5.1 | https://www.graphpad.com/ | N/A |
| Fiji: ImageJ | https://fiji.sc/ | N/A |
| **Other** | | |
| Olympus IX73 | Olympus | |
| Axio Observer.Z1 | Zeiss | |
| Tecnai G2 | Thermo Fisher Scientific | |
| Nikon-Structured Illumination Microscopy | Nikon | |

## Cell culture and transfection

HeLa S3 (organism: *Homo sapiens*, clone number:S3, supplier: ATCC, catalog-number: CCL-2.2, RRID number: CVCL_0058), EA.hy926 (organism: *Homo sapiens*, RRID number: CVCL_3901, provided by Dr. C.J.S. Edgell, University of North Carolina, USA), and Perivascular cells (purified from 10 adult brain meninges of 4–6-month-old AnxA5-LacZ+ mice with a mixed genetic background C57/BL6x129/SvJ), provided from Prof. Dr. Bent Brachvogel, University of Cologne, Germany), were cultivated in DMEM (D5523, Sigma-Aldrich, Vienna, Austria) containing 10% FCS (Gibco, Thermo Fisher Scientific), penicillin (100 U/ml), streptomycin (100 μg/ml) and amphotericin B (1.25 μg/ml) (Gibco) in a humidified incubator (37 °C, 5% CO$_2$, 95% air). Cells were regularly checked for mycoplasma contamination and were found negative.

For microscopic measurements, cells were seeded either on 30 mm glass coverslips or 1.5 H high-precision glass coverslips (Marienfeld-Superior) and transfected at 40% confluence (HeLa), 60% confluence perivascular cells, and 70–80% confluence (EA.hy926) with 1 μg respective plasmid by using PolyJet™ (SignaGen Laboratories, Rockville, MD, USA) transfection reagent in 1 ml full culture medium. 5 h after transfection, the medium was replaced with 2 ml of full culture medium. For knockdown experiments, HeLa cells (50–60% confluence), were transfected either with siRNA or siRNA and plasmid combination by using 3 μl of TransFast transfection reagent (Promega, Madison, WI, USA) in 1 ml serum- and antibiotic-free medium for 14–16 h. Afterward, the transfection medium was replaced with 2 ml of full culture medium. Experiments were performed 40–48 h after transfection. Before each experiment, cells were adjusted to room temperature while keeping the cells in a storage buffer. VDAC1 siRNA sequence (sense strands, 5′-3′) ACACUAGGCACCGAGAUUA (siVDAC1), scrambled Control siRNA sequence UUCUCCGAAC-GUGUCACGU (siControl) and AnxA9 siRNA sequence (sense strands, 5′-3′) GCAGUCUACAAACACAAUUtt-3 Microsynth (Balgach, Switzerland) were used. AnxA8 siRNA (SASI_Hs01_00091429) was purchased from Sigma-Aldrich, St. Louis, MO. The knockdown level of siVDAC1-treated cells was verified with a western blot while

the knockdown level of siAnxA8 and siAnxA9 treated cells were verified with qPCR.

## Generation of AnxA5-KO cells

For generating AnxA5-KO HeLa cells, we ordered pcDNA3.3-Cas9-2A-eGFP and pGS-U6-gAnxA5 from Genscript (Thermo Fischer Scientific). The gRNA sequence of AnxA5 is 5′-GATGCA-GAAACTCTTCGGA-3′. Next, plasmids were co-transfected into HeLa cells in a ratio of 1:1. 48 h post-transfection, GFP-positive cells were single-sorted into 96-well plates and further cultivated. In the end, around 50 clones were analyzed with PCR. To validate whether each allele of 5 clones was mutated, we made bacterial sub-clones of amplified PCR fragments and sequenced them using the Sanger method (Microsynth, Balgach, Switzerland). For the generation of AnxA5-KO in EA.hy926, cells were co-transfected with pcDNA3.3-Cas9-2A-eGFP and pGS-U6-gAnxA5 in the ratio of 1:1. Next day GFP-positive cells were sorted and, respectively, 200–800 cells were seeded to 10 cm dishes. Later, 24 clones were picked and further cultivated. Clones were analyzed by PCR and Western blot. WT and AnxA5-KO perivascular cells were isolated from mice as described elsewhere (Bouter et al, 2011).

## Constructs

AnxA5 mutants, including AnxA5-2Mt (D144N, E228Q), AnxA5-3Mt (D144N, E228Q, D303N) (Jin et al, 2004), and AnxA5-5Mt (R18E, R25E, K29E, K58E, and K193E) (Bouter et al, 2011), were purchased from Gene Universal Inc. in Newark, USA. VDAC2-FLAG and VDAC3-FLAG were a kind gift from Prof. Stefan Jakobs' lab.

## Short hairpin RNA (shRNA) mediated AnxA5 silencing

HeLa cells were cultured in 6-well plates under standard conditions and grown to a confluency of 30%. Cells were then transfected using 10 µL of shRNA plasmid DNA (Control shRNA Plasmid-A, sc-108060, for control cells; AnxA5 shRNA Plasmid, sc-29686-SH, for AnxA5 knockdown cells; all plasmids purchased from Santa Cruz Biotechn.). 48 h post-transfection, the transfection medium was replaced by a fresh medium containing 1.0 µg/ml puromycin. 48 h later 90% of the cells died and the selective medium was changed every two days. Colonies were picked using a pipette filled with trypsin and cultured under selective conditions in separate plates.

## mRNA isolation and quantitative PCR

Total mRNA was extracted employing the RNeasy® Mini Kit (Qiagen, Hilden, Germany), and reverse transcription was carried out using the Applied Biosystems High-Capacity cDNA Reverse Transcription kit (Thermo Fisher Scientific Baltics UAB, Vilnius, Lithuania). For qPCR, the Promega GOTaq® qPCR Master Mix (Madison, WI, USA) was utilized.

## Buffers and solutions

Before the microscopic measurements, cells were adjusted to room temperature in storage buffer: 2 mM $Ca^{2+}$, 138 mM NaCl, 1 mM $MgCl_2$, 5 mM KCl, 10 mM HEPES, 2.6 mM $NaHCO_3$, 0.44 mM $KH_2PO_4$, amino acid, and vitamins mix, 10 mM glucose, 2 mM L-glutamine, 1% penicillin/streptomycin, 1.25 µg/mL amphotericin B and pH adjusted to 7.4. Live-cell imaging experiments were performed either in the Hepes-buffered $Ca^{2+}$ containing experimental buffer (2CaNa): 2 mM $CaCl_2$, 138 mM NaCl, 1 mM $MgCl_2$, 5 mM KCl, 10 mM Hepes, and 10 mM D-glucose at pH 7.4 or a nominal $Ca^{2+}$ free solution (0CaNa) containing 0.1 mM EGTA instead of $CaCl_2$. Cells were stimulated with 100 µM ATP or Histamine, and BHQ (15 µM) in either 2CaNa, or 0CaNa buffer.

## Live-cell imaging experiments

Experiments were conducted using an Olympus IX73 inverted microscope equipped with a UApoN340 40× oil immersion objective (Olympus, Tokyo, Japan) and a CCD Retiga R1 camera (Q-imaging, Surrey, BC, Canada). Illumination was provided by LedHUB® (Omnicron, Germany) with 340, 385, 455, 470, and 550 nm LEDs combined with a CFP/YFP/RFP (CFP/YFP/mCherry-3X, Semrock, New York, NY, USA) filter set. Data acquisition was performed using VisiView 4.2.01 (Visitron, Puchheim, Germany). Cells were placed into a flow chamber for the life cell imaging experiments. A gravity-based perfusion system PS-9D (NGFI, Graz, Austria) was used to perfuse cells during measurements. The valve with nine positions was connected to reservoirs, and the desired reservoir was activated automatically using perfusion control software. Cells were randomly selected based on the expression of genetically encoded biosensors.

## Detection of sub-mitochondrial $Ca^{2+}$ levels

Mitochondrial matrix $Ca^{2+}$ level was measured using genetically encoded, FRET-based, mitochondrial matrix targeted ratiometric $Ca^{2+}$ sensor 4mtD3cpv (Palmer et al, 2006). Mitochondrial IMS $Ca^{2+}$ measurements were performed with a genetically encoded ratiometric green emitting IMS-targeted $Ca^{2+}$ sensor IMS-GEM-GECO1 (Waldeck-Weiermair et al, 2019). Mitochondrial cristae measurement was performed with genetically encoded ratiometric cristae-targeted $Ca^{2+}$ sensor ROMO-GEM-GECO1 (Waldeck-Weiermair et al, 2019). Dynamic FRET measurements between MICU1-CFP and MICU1-YFP were performed with the MICU1 FRET sensor (Waldeck-Weiermair et al, 2015). 4mtD3cpv sensor was illuminated with 455 nm LED every 2 or 3 s using a 300 ms exposure time. Emission was collected at 480 nm and 530 nm by using a CFP/YFP/mCherry-3X filter set and 505dcxr beam-splitter. IMS-GEM-GECO1 and ROMO-GEM-GECO1 sensors were illuminated with 425 nm LED every 2 or 3 s using a 200 ms exposure time. Emission was collected at 480 nm and 530 by using a CFP/YFP filter set and with 505dcxr beam-splitter. All recordings were background subtracted and corrected for bleaching using an extrapolation of an exponential decay fit of the initial basal phase.

## Detection of cytosolic $Ca^{2+}$ levels

Cytosolic $Ca^{2+}$ levels were measured either with genetically encoded cytosol-targeted intensiometric $Ca^{2+}$ sensor jGCaMP7c (Dana et al, 2019) or ratiometric cytosolic $Ca^{2+}$ dye Fura-2 acetoxy-methyl-ester (Fura-2 AM) (TEFLabs, Austin, TX). For cytosolic $Ca^{2+}$ measurements with jGCaMP7c, cells were transfected as described above. For cytosolic $Ca^{2+}$ measurements with Fura-2 AM, cells were incubated with 3.3 µM (60% confluency for HeLa and 80%

confluency for perivascular cells and EA.hy926 cells) FURA-2 AM in the storage buffer for 30 min. After incubation, cells were 2 times washed with a storage buffer and measurements were done either in 2CaNa or 0CaNa buffer. All recordings were background subtracted.

jGCaMP7c sensor was illuminated with 470 nM LED for every 2 s using a 200 ms exposure time. Emission was collected at 510. For Fura-2 AM measurements, cells were sequentially illuminated with 340 and 385 nm LED every 2–3 s using a 300 ms exposure time. Emission was collected at 510 nm. All recordings were background subtracted.

## Detection of ER Ca²⁺ levels

ER, $Ca^{2+}$ levels were measured with genetically encoded, FRET-based, ER-targeted ratiometric $Ca^{2+}$ sensor D1ER (Palmer et al, 2004). This sensor was illuminated with a 455 nm LED every 3 s using a 300-ms exposure time. Emission was collected at 480 nm and 530 nm by using a CFP/YFP/mCherry-3X filter set and 505dcxr beam-splitter. All recordings were background subtracted and corrected for bleaching using an extrapolation of an exponential decay fit of the initial basal phase.

## Mitochondrial membrane potential measurements

Mitochondrial membrane potential experiments were done as described previously (Oflaz et al, 2022). Briefly, Tetramethylrhodamine methyl ester perchlorate (TMRM) (Invitrogen™ T668; Vienna, Austria) dye was used for mitochondrial membrane potential experiments. TMRM dye was excited with 550 nm LED every 3 s using a 200 ms exposure time. Emission was collected at 600 nm using CFP/YFP/mCherry-3X filter set. Briefly, cells were incubated at room temperature for 30 min with a storage buffer containing 25 nM (cells were 50–60% confluent) of TMRM dye. After incubation cells were perfused with 2CaNa for 2 min and basal values were recorded. Next, cells were perfused for 4 min with 2CaNa buffer containing 1 μM of carbonyl cyanide-p-trifluoromethoxyphenylhydrazone (FCCP) (Abcam, Cambridge, UK) to completely depolarize the mitochondria. Background subtracted mitochondria to nucleus fluorescence ratio was used for calculating mitochondrial basal membrane potential in WT and AnxA5-KO cells.

## ER-mitochondria co-localization and 3D-morphological analysis of mitochondria

To label ER, WT, and AnxA5-KO (HeLa) cells were transfected with ERAT4.03 NA (NGFI, Graz, Austria) and mitochondria were stained with Mitotracker Red CMXROS (MTR-CMX) (Invitrogen, Thermo Fischer Scientific, Vienna, Austria). Alternatively, the split-GFP-based ER-mitochondria contact site sensor (Cieri et al, 2018), together with the AKAP-RFP-CAAX construct (Csordás et al, 2006) (used to tag mitochondria to the cell membrane), were used to assess the ER-mitochondria contact sites. Cells were imaged with a confocal spinning disk microscope (Axio Observer.Z1 from Zeiss, Gottingen, Germany). The microscope is equipped with a 100x objective (Plan-Fluor x100/1.45 Oil, Zeiss), a motorized filter wheel (CSUX1FW, Yokogawa Electric Corporation, Tokyo, Japan), and an AOTF-based laser merge module (laser lines: 405, 445, 473, 488, 514, and 561 nm (Visitron Systems)). ERAT4.03 NA or SPLICS$_L$ (in

the case of using SPLICS technology) and MTR-CMX were sequentially excited with 488 and 561 and emission was acquired at 530 and 600 nm, respectively, by using a charged CCD camera (CoolSNAP-HQ, Photometrics, Tucson, AZ, USA). Z-stacks with 0.2 μm z-height increments were imaged. VisiView software (Universal Imaging, Visitron Systems) was used to acquire the imaged data. Images were blind deconvoluted (NIS-Elements, Nikon, Vienna), and co-localization between ER (ERAT4.03 NA) and mitochondria (MTR-CMX) was determined on a single-cell level using ImageJ and the plugin coloc2. Pearson's coefficient was calculated as described elsewhere (Koshenov et al, 2021).

For morphology analysis of mitochondria, custom-made ImageJ macro was used as described elsewhere (Gottschalk et al, 2019). Briefly, after deconvolution, the rolling ball method was applied for background subtraction to further increase the contrast. A global auto Otsu threshold (stack histogram) and local Otsu threshold (radius of 640 nm, single slice histogram) were applied to the stack and merged. Mitochondria were segmented and binarized by using the ImageJ plugin 3D manager. Mitochondrial volume and surface were determined with the 3D Geometrical Measure plugin and elongation and flatness parameters were obtained with the 3D Ellipsoid Fitting plugin. Mitochondrial branching was quantified by calculating the ratio of the mitochondrial volume to the 3D ellipse volume fitted to the respective mitochondrion.

## Morphological analysis of mitochondria imaged with SIM

Cells were co-transfected with MICU1-YFP and MCU-mCherry constructs. Single 3D-SIM and time-lapsed images of MCU-mCherry, before and after the ER $Ca^{2+}$ release (with 100 μM of Histamine) were used for morphological analysis of mitochondria as described previously (Gottschalk et al, 2019). Briefly, ImageJ Plugin (Mosaic Suite, background subtractor, NIH) was applied to images for background correction and binarization was done by using a Yen auto threshold. Mitochondrial count, area, minor, and major axes of the mitochondria were obtained by using the ImageJ particle analyzer. The ratio of the major to minor axis of mitochondria was used to calculate the aspect ratio.

## Cristae membrane kinetics quantification

Quantification of cristae membrane--kinetics in whole mitochondria and the MERCs was done as described elsewhere (Gottschalk et al, 2022, 2018). Briefly, WT and AnxA5-KO (HeLa) cells were transfected with ER-RFP and stained with Mitotracker Green/FM (MTG) and were recorded with live dual-color SIM imaging. Local and global automated thresholding of the MTG-stained cristae membrane kinetics were quantified, before and after ER $Ca^{2+}$ release (with 100 μM of histamine stimulation), by measuring the cristae membrane alterations per frame within the mitochondria. To detect the cristae membrane-kinetics in MERCs, the partially delated area of the overlap of ER-RFP and MTG staining was used as a mask.

## Analysis of MCU shuttling to IBM

Analysis of MCU shuttling to IBM in WT and AnxA5-KO (HeLa) cells was done as described previously (Gottschalk et al, 2019). Briefly, cells were co-transfected with MICU1-YFP (localized in the

IBM) and MCU-mCherry (localized in the IMM) constructs and imaged before and after ER Ca²⁺ release. Two masks of the cristae membrane and IBM were used to measure the cristae membrane and IBM fluorescence intensities of MCU-mCherry. The IBM association index was calculated by the intensity ratio of IBM to cristae membrane. The higher the value of the IBM association index, the higher the distribution of MCU shifts towards IBM. The freeware program ImageJ was used for the image analysis.

## Electron microscopy

For chemical fixation, WT and AnxA5-KO (HeLa) cells were seeded onto Aclar fluoropolymer foils and fixed with a mixture of buffered 2% paraformaldehyde and 2.5% glutaraldehyde. After osmification, the cells were dehydrated in a range of ethanol solutions (50% to 100%). The transition solvent propylene oxide was then incubated for 30 min, followed by propylene oxide and TAAB embedding resin (TAAB, Berkshire, UK) mixture incubations of 1:1 and 1:2. Finally, the samples were incubated in pure resin and polymerized at 60 °C for 3 days, as described previously (Reichmann et al, 2015).

High-pressure freezing and freeze substitution were done as described previously (Sobol et al, 2010). Briefly, HeLa cells were plated onto sapphire discs, which included untreated WT cells and cells that were stimulated with histamine for 20 s. The sapphire discs containing the cells were then placed into aluminum carriers and were high-pressure frozen using a Leica HPM 100 high-pressure freezer. Freeze substitution (FS) was conducted first by incubating in acetone for 44 h at −90 °C with one acetone exchange after 24 h. This was followed by acetone with 0.5% glutaraldehyde and 1.5% distilled water for 6 h during which the temperature was gradually raised to −60 °C, incubated for 8 h at −60 °C in the same substitution solution, a gradual temperature rise to −40 °C for the next 4 h, and 2 h at −40 °C. The final step of FS was undertaken in pure acetone, with the temperature being raised from −40 °C to 0 °C for 3.3 h and held for 30 min. To eliminate acetone, the substituted samples were rinsed in ice-cold 96% ethanol four times. Resin infiltration was carried out on ice using a ratio of 96% ethanol to LR white of 2:1 and then 1:2, each step lasting 30 min. The samples were left overnight in pure LR White resin at 4 °C and then polymerized for three days using UV light at 4 °C.

For immunogold labeling, the freeze-substituted cells that had been embedded in LR White resin were ultrathin sectioned and placed on pioloform-covered nickel grids. After blocking in 5% normal goat serum for 30 min, they were incubated for 2 h in 1:100 Anti-Annexin V antibody (Abcam ab14196) at room temperature, and for 1 h with 10 nm gold coupled goat anti-rabbit IgG (BB International EM). The sections were thoroughly washed between each incubation step. A negative control in which the primary antibody had been omitted remained free of staining. Electron micrographs were made with a Thermo Fisher FEI Tecnai G2 20 with an Ametek Gatan US1000 camera. Whole cells were visualized with magnification 5000x.

## Analysis of cristae membrane, density, and cristae density distribution

Analysis of the cristae membrane density was done as described previously (Gottschalk et al, 2022). Briefly, mitochondria were

selected by free-hand selection. Next, from the selected mitochondria, cristae were segmented by free hand. Mitochondria and cristae area and perimeter were measured. The ratio of the perimeter of cristae to the area of mitochondria indicates the density of cristae within single mitochondria. The ratio of cristae perimeter to mitochondria perimeter indicates the ratio of cristae to mitochondria membrane content. Cristae density distribution was analyzed as described elsewhere (Gottschalk et al, 2022). Briefly, mitochondrial segments were converted to binary masks and applied as masks to binarized cristae membranes to calculate the percentage of cristae perimeter within the mitochondrial area. The mitochondrial mask was gradually eroded in 2-pixel increments (5.88 nm) while measuring the corresponding cristae coverage. These measurements provided cristae density in circular segments, starting from the outer mitochondrial membrane and extending toward the mitochondrial center. To account for variations in mitochondrial size and shape, resulting in different numbers of ring segments, cristae densities were normalized using linear interpolation to 100 segments.

### Analysis of the immunogold staining

Electron micrographs that had been immunolabelled for AnxA5 were manually segmented to yield masks defining the areas of cytosol, nucleus, and mitochondria. Further, the cell was screened for gold particles and the x/y-localizations were saved for further analysis. Gold granules were recognized by their size, perfectly round shape, and electron lucent halo in under focus. A custom-made macro was used to determine the gold particle localization according to the cell compartments cytosol, nucleus, and mitochondria. Additionally, the shortest distance of each gold particle, localized in the cytosol or mitochondria, to the outer mitochondrial membrane was measured.

## Proximity ligation assay

Proximity ligation assay (PLA) was performed as described previously (Tubbs & Rieusset, 2016). Briefly, WT and AnxA5-KO cells were seeded in 12-well plates. Cells were transfected either with neg. siRNA or siVDAC1. 48 h after transfection, cells were washed with 1% PBS and fixed with 10% formaldehyde for 10 min (room temperature). After 10 min the reaction was stopped by 1 M of glycine solution (pH 2.2) and incubated with 100 mM of glycine for 15 min. Next, cells were washed with PBS and permeabilized for 15 min with 0.1% Triton X-100 and washed 2 times with PBS. For blocking, 50 μL of blocking solution (Duolink In Situ PLA Probe, Sigma-Aldrich) was added to each well, and the plate was incubated for 30 min at 37 °C in a humidified incubator. After incubation, the blocking solution was removed from each well. Next, primary antibodies for Annexin 5 (Abcam, ab14196, 1:200) and VDAC1 (Abcam, 186321, 1:400) or IP₃R (Abcam, ab5804, 1:200) and VDAC1 (Abcam, 186321, 1:400) were diluted in PBS, added to each well, and incubated overnight at 4 °C. Next day, cells were washed two times with Tris Buffered Saline with 0.01% Tween (TBS-T). Next, diluted (1:5) proximity ligation assay probes (Duolink In Situ PLA Probe, Sigma-Aldrich), were added to each well and incubated at 37 °C in a humidified incubator for an hour. After incubation, cells were washed two times with TBS-T. For the ligation, an in situ detection reagent (Texas red kit (Texas Red Oligo Labeling, Bio-Synthesis (Zotal)) was added to wells and incubated 30 min at 37 °C, and cells were washed two times with TBS-T. Next, an

amplification solution (provided with KIT), was added to each sample, and the plate was incubated in a humidity chamber for 100 min at 37 °C. After incubation, coverslips were washed and mounted on slides with a DAPI-containing mounting medium (Immuno Bioscience, AR-6500-01). Slides were imaged with confocal microscopy (Olympus 1X81) 2 h after mounting. For counting single PLA signals, a custom-made macro in image-J Fiji was used. A certain threshold was set for PLA signals and the signals that are above the determined threshold are considered as positive PLA dots. The number of cells was counted based on the DAPI-containing cells. The ratio of PLA dots to the number of cells was used as a readout. Finally, ratios were normalized to WT cells.

## Western blot

To compare the level of proteins that are involved in mitochondrial $Ca^{2+}$ uptake, HeLa and AnxA5-KO cells were seeded in 6-well plates. Next, cells were harvested in RIPA buffer (25 mM Tris-HCl pH 7.6, 150 mM NaCl, 5 mM EDTA, 1% Triton X-100, 1% sodium deoxycholate, 0.1% SDS) supplemented with protease inhibitor cocktail (#P8340 Sigma, Vienna, Austria). The concentration of extracted proteins was determined with Pierce™ BCA Protein Assay Kit (ThermoFisher Scientific, Waltham, MA, USA) on a CLARIOstar Plus (BMG Labtech, Ortenberg, Germany). Samples were resolved on SDS-PAGE and transferred to the membrane. Next, membranes were incubated with primary antibodies Annexin 5 (Santa Cruz, sc-74438 1:1000), VDAC1 (Abcam, 186321, 1:5000), MICU1 (Cell Signaling Technology, D4P8Q, 1:1000), UCP2 (Cell Signaling Technology, D105V, 1:1000), MCU (Cell Signaling Technology, D2Z3B, 1:1000), β-Actin (Sigma, A5316, 1:1000), Histone H3 (Cell Signaling Technology, 1B1B2, 1:1000) at 4 °C overnight. The next day, membranes were incubated with appropriate HRP-conjugated secondary antibodies for 1 h at room temperature. Washed membranes were incubated with SuperSignal™ West Pico PLUS Chemiluminescent Substrate (ThermoFisher Scientific) and the signal was captured with ChemiDoc MP Imaging System (Biorad). Signal intensities were quantified by Image-J Fiji (Schindelin et al, 2012) and normalized to intensities of the appropriate β-actin or Histone H3 signal which are used as a loading control.

For the cell fractionation assay, the concentration of cytosolic, crude mitochondria, and pure mitochondria (+/-Proteinase K) fractions were determined with the BCA Protein Assay Kit (ThermoFisher Scientific, Waltham, MA, USA) on a CLARIOstar Plus (BMG Labtech, Ortenberg, Germany). Annexin 5 (Annexin 5 Abcam, ab14196, 1:1000), VDAC1 (Abcam, 186321, 1:5000), Cytochrome C (Cell signaling, 4272, 1:500), TOM20 (Cell Signaling, D8T4N, 1:1000), Tubulin (BioLegend, 801202, 1:1000) antibodies and HRP-conjugated corresponding secondary antibodies were used for immunoblotting.

For KD validation of VDAC1, HeLa cells were seeded in 6-well plates and transfected with respective siRNAs (neg. siRNA or siVDAC1) and harvested 48 h post-transfection. VDAC1 and β-actin antibodies and HRP-conjugated corresponding secondary antibodies were used for immunoblotting.

## Co-immunoprecipitation

Co-immunoprecipitation was performed as described previously (Gottschalk et al, 2019). Briefly, AnxA5-KO Hela cells were

transfected either with empty plasmid (pcDNA3.1 (-)) or AnxA5-Flag plasmid 48 h before the experiment. Next, cells were lysed in 0.7 mL lysis buffer (100 mM NaCl, 20 mM Tris, 1 mM EGTA, 5 mM n-Dodecyl β-D-*maltoside*, pH 7.5-HCl) supplemented with protease inhibitors (10 μM phenylmethylsulfonyl fluoride, and 1 μg/ml each aprotinin, leupeptin, and pepstatin). After a short freeze-thaw cycle lysates were centrifuged (12,000 × g, 10 min) and 60 μl of supernatants were removed for protein determination using a BCA assay (Thermo-Fisher) and total cell lysate analyses (input). The rest of the lysates (∼1.5 mg total protein) were incubated with 30 μl of anti-Flag M2 affinity gel (Sigma-Aldrich; beads were prepared as recommended by the manufacturer) for 30 min on a rotary shaker. Then, beads were collected by centrifugation (8000 × g, 30 s) and washed three times with TBS. All steps were performed on ice or 4 °C. Elution of FLAG-fusion proteins was performed in 50 μl of 2x Lämmli sample buffer with 5% β-mercaptoethanol and protein samples were boiled for 10 min before SDS-PAGE.

For cross-linking fixation, cells were washed with PBS, and incubated with 10% formaldehyde (10 min, RT, under agitation). The reaction was stopped with 1 M glycine (pH 8), cells were washed with PBS and 100 mM glycine was added for 15 min at RT under agitation. Then cells were prepared for co-immunoprecipitation as described above.

For immunoblotting, whole cell lysates (input, 40 μg (VDAC1) and 20 μg (Annexin V) of total protein) and co-immunoprecipitation samples were subjected to SDS-PAGE. PVDF membranes were probed at 4 °C overnight with primary anti-VDAC1 (Abcam, 186321, 1:1000) and AnxA5 (Annexin 5 Abcam, ab14196, 1:1000). To avoid interference from denatured heavy and light chain IgG, the HRP-conjugated immunoblot reagent Veriblot (Abcam; 1:000; ab131366) was used for the detection of co-immunoprecipitated proteins. Immunoreactive bands were visualized using Immobilon Western HRP Substrate (Thermo Fisher) and the chemiluminescence detection system ChemiDoc (Bio-Rad).

## Chemical cross-linking

Cross-linking experiments were done as described previously (Weisthal et al, 2014). Briefly, cells (1.5 mg/ml) were harvested and incubated with 300 μM cross-linking reagent, ethylene glycol bis(succinimidyl succinate) (Thermo Fisher, 21565) in PBS (pH 8.3) at 30 °C for 20 min. Samples (30 μg of protein) were subjected to SDS-PAGE and immunoblotting using mouse anti-VDAC1 antibody (Abcam, 186321, 1:5000). Quantitative analysis of the VDAC1 dimer level was calculated by using the image-J Fiji (Schindelin et al, 2012).

## Cisplatin, selenite, and VBIT-4 treatment

WT and AnxA5-KO HeLa cells were treated with either 5 and 10 μM cisplatin (Sigma, P4394), 10 μM selenite (Sigma, S5261), or co-treated with 10 μM cisplatin/selenite plus 20 μM VBIT-4 (VBIT-4 was a generous gift from Prof. Dr. Varda Shoshan-Barmatz, Ben-Gurion University of the Negev, Beer-Sheva, Israel). In the case of co-treatment, cells were initially pre-incubated with VBIT-4 for 2 h and subsequently exposed to cisplatin/selenite and VBIT-4.

## VDAC1 cluster size experiments and analysis

Cells expressing tetracysteine-tagged VDAC1 (VDAC1-TC) were stained with 1 μM FlAsH-EDT$_2$ (Cayman Chemical, Michigan, USA) in EHL for 15 min at 37 °C. The cells were washed with 100 μM BAL (2,3-dimercaptopropanol or British anti-Lewisite) in EHL for 15 min at 37 °C and kept in EHL before confocal imaging.

High-resolution imaging was performed with an array confocal laser scanning microscope (Axiovert 200 M, Zeiss) with a 100×/1.45 NA oil immersion objective (Plan-Fluor, Zeiss) and a Nipkow-based confocal scanner unit (CSU-X1, Yokogawa Electric Corporation, Tokyo, Japan). Laser light of diode lasers (Visitron Systems) served as an excitation light source. VDAC1-TC and mitoDsRed were exited at 488 nm and 561 nm, respectively. Emissions were captured with a CoolSNAP HQ2 CCD Camera (Photometrics Tucson, Arizona, USA) using the following emission filters (Chroma Technology Corporation, VT, USA): ET535/30 m for VDAC1-TC and ET630/75 m for mitoDsRed.

The analysis of the VDAC1 cluster size was conducted using Fiji software. To enhance cluster visibility, a Gaussian blur preprocessing technique was applied. The Watershed method was then used to separate closely positioned clusters, followed by the application of the MaxEntropy thresholding method for accurate cluster identification.

## Flow cytometry

After 24 or 48 h of treatment with cisplatin, VBIT4, and combination, cells and cell supernatant were harvested and stained using the FITC Annexin V Apoptosis Detection Kit 1 (BD Biosciences) according to the manufacturer's instructions. Flow cytometric analyses were performed using a CytoFLEX S flow cytometer (Beckman Coulter Life Sciences) and analyzed by CytExpert Software (Beckman Coulter).

## Mitochondria isolation

For patch clamp experiments, crude mitochondria were isolated with differential centrifugation steps as described previously (Frezza et al, 2007). Briefly, one to two days before the isolation, WT and AnxA5-KO cells were seeded on a 20 cm dish. On the day of isolation, cells (90–100% confluency) were trypsinized, harvested, and washed with PBS. All the following steps were performed on ice. The cell pellet was suspended in 1 ml of IMBc(+PI) buffer and homogenized with a glass-Teflon potter with 80 strokes. The suspension was placed in a 1.5 ml Eppendorf tube and centrifuged at $600 \times g$ for 10 min at 4 °C (This step was repeated two times to fully remove the cell debris). The supernatant was collected into a new pre-chilled Eppendorf tube and further centrifuged at 9000 g for 15 min at 4 °C. After centrifugation, the supernatant was removed and the pellet (containing mitochondria) was resuspended with 200 μM of IMBc(+PI). Mitochondrial resuspension was centrifuged at $9000 \times g$ for 15 min at 4 °C and the pellet was kept on ice for further experiments.

Pure mitochondrial fractions were isolated by Percoll (Sigma, P1644) density centrifugation as previously described (Wieckowski et al, 2009). Briefly, HeLa cells were seeded on a T175 flask two days before the experiment. On the day of isolation, cells (90–100% confluency) were trypsinized, harvested, and washed with PBS.

Next, the cell pellet was resuspended in IB$_{cells}$–1 buffer (all buffer compositions are described in (Wieckowski et al, 2009) and homogenized by using a Teflon potter. Homogenate was centrifuged at $600 \times g$ for 5 min at 4 °C. Supernatant (containing mitochondria) was collected and further centrifuged at $7000 \times g$ for 10 min. The supernatant was used for cytosolic fraction isolation (supernatant was centrifuged at $100,000 \times g$ to collect organelle-free cytosolic fraction) and the pellet (containing mitochondria) was resuspended in MRB buffer. Next, Percoll medium was added to the ultracentrifuge tube (thin-wall ultra-clear tube, 344060 Beckman Coulter). Mitochondrial suspension layered on Percoll medium and MRB buffer was added on top of the mitochondrial fraction and centrifuged at $95,000 \times g$ for 35 min (SW40 rotor, Beckman). Pure mitochondrial fractions were collected with a Pasteur pipette, centrifuged at $6300 \times g$, resuspended in MRB buffer, and centrifuged at $6300 \times g$ to collect pure mitochondrial pellets. All steps were performed on ice and centrifugations were carried out at 4 °C.

For Proteinase K treatment, the pure mitochondrial fraction was incubated with 50 μg/ml of proteinase K for 15 min at 37 °C. To terminate proteinase K activity, the mitochondrial fraction was incubated with 2 mM of Phenylmethylsulfonyl fluoride PMSF (Sigma, P7626) for 10 min at room temperature.

## Calculation of the mitochondrial localized percentage of AnxA5

For calculating AnxA5 distribution in the cytosol, pure mitochondria, and PK-treated pure mitochondria, we applied formula 1:

$$AnxA5_{p.mito}\% = \frac{AnxA5_{p.mito}}{AnxA5_{p.mito} + AnxA5_{cyto}} \cdot 100 \qquad (1)$$

with $AnxA5_{cyto}$ and $AnxA5_{p.mito}$ being defined as formulas 2 and 3:

$$AnxA5_{cyto} = I_{blot} \cdot D_1 \cdot D_2 \qquad (2)$$

$$AnxA5_{p.mito} = I_{blot} \cdot D_1 \cdot D_2 \qquad (3)$$

$D_1$ is the loading dilution factor and $D_2$ is the factor needed to compensate for the fraction of the complete purification process.

The same calculations were applied to determine the ratio of pure mitochondria and PK-treated mitochondria

Further, to compensate for the loss of mitochondrial mass during the purification process of mitochondria for the determination of AnxA5 distribution we determined the loss of Cytochrome C from crude to pure mitochondria and included it as a factor for purification loss ($K_{pl}$) into formula 1.

$$AnxA5_{p.mito}\% = \frac{AnxA5_{p.mito}}{AnxA5_{p.mito} + AnxA5_{cyto}} \cdot K_{pl} \cdot 100 \qquad (4)$$

Using formulas 4 and 5, the proportions of cytosolic and mitochondrial AnxA5 were determined:

$$AnxA5_{cyto}\% = 100 - AnxA5_{p.mito}\% \qquad (5)$$

To compensate for mitochondria with damaged OMM integrity, we used VDAC1 lanes to account for the loss of AnxA5 during PK treatment because of damaged mitochondria and included it as a

factor for damaged mitochondria ($K_{dm}$) into formula 1.

$$AnxA5_{p.mito}\% = \frac{AnxA5_{p.mito}}{AnxA5_{p.mito} + AnxA5_{p.mito+PK}} \cdot K_{dm} \cdot 100 \quad (6)$$

Using formulas 6 and 7, the AnxA5 proportions between the inner and outer leaflet of the outer mitochondrial membrane were determined:

$$AnxA5_{p.mito+PK}\% = 100 - AnxA5_{p.mito}\% \quad (7)$$

## Patch clamp of intact mitochondria

Single-channel measurements were performed in the intact mitochondria-attached configuration allowing to record ionic current passing the outer mitochondrial membrane. Patch pipettes were pulled from the glass capillaries using a Narishige puller (Narishige Co., Ltd., Tokyo, Japan), fire-polished, and had a resistance of 12–15 MΩ, when filled with a solution containing (in mM): 100 K gluconate, 40 KCl, 3 EGTA, 10 HEPES. Free $Ca^{2+}$ concentration was adjusted to 10 μM by adding the appropriate amount of $CaCl_2$ calculated by the program CaBuff, and pH was adjusted to 7.2 by KOH. Recombinant AnxA5 (Abcam, ab255714) was added to the pipette at the final concentration of 200 ng/ml. Mitochondria were bathed in the solution of the same composition. In some experiments, a $Ca^{2+}$-free pipette solution of the following composition was used: 100 mM K gluconate, 40 mM KCl, 0.3 mM EGTA, 10 mM HEPES, with pH adjusted to 7.2 by KOH. Single-channel currents were recorded at fixed test potentials indicated in the respective figures. Voltage steps of 2 s duration from the holding potential 0 mV were delivered to the test potentials every 5 s. Test potentials are indicated relative to the inner membrane surface. Mitochondria were approached by Sensapex micromanipulator (SMX series, Sensapex, Oulu, Finland). Currents were recorded using a patch-clamp amplifier Axopatch 200B (Molecular Devices, Sunnyvale, CA, USA). Data collection was performed using Clampex software of pClamp (V9.0, Molecular Devices, Sunnyvale, CA, USA). Signals obtained were low pass filtered at 1 kHz and digitized with a sample rate of 10 kHz using a Digidata 1322A A/D converter (Molecular Devices, Sunnyvale, CA, USA). Data were analyzed using Clampfit 10.3 software of pClamp (Molecular Devices, Sunnyvale, CA, USA).

## Statistical analysis and reproducibility

The number of independent experiments was represented as "n = single-cell/independent experiment" in each figure legend, along with the used statistical test and $p$-value. The normality distribution of the data was assessed, and in instances where the data deviated from a normal distribution, appropriate tests were employed. Statistical analyses (Student's t-test or Mann-Whitney U-Test and analysis of variance (ANOVA) with Tukey post hoc test or Kruskal–Wallis test) and data visualization were performed on GraphPad Prism software version 9.3.1 (GraphPad Software, San Diego, CA, USA) or Microsoft Excel (Microsoft Office 2013). Bar graphs with individual values (represented as single dots) were used to show the data distribution and mean value of the data where the differences with $p < 0.05$ were considered statistically significant.

## Data availability

The data supporting this research's findings can be obtained from the authors upon reasonable request; please refer to the author's contributions for details regarding specific datasets. Source data are provided with this paper. No large-scale data amenable to data repository depositions were generated in this study.

The source data of this paper are collected in the following database record: biostudies:S-SCDT-10_1038-S44318-025-00454-9.

## Peer review information

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

## Acknowledgements

This research was funded in whole or in part by the Austrian Science Fund (FWF) [https://doi.org/10.55776/W1226]. WFG is grateful to the Austrian Science Fund (FWF) for excellence cluster 10.55776/COE14, that also supported FEO and the revision of this study. For open access purposes, the author has applied a CC BY public copyright license to any author-accepted manuscript version arising from this submission. We thank the Medical University of Graz for financial support. This work was supported by the FWF (DK-MCD W1226 to WFG, P28529, and I3716 to RM), the MEFO Graz (to WFG), Nikon Austria (to WFG), the Deutsche Forschungsgesellschaft (DFG) (FOR2722-407146744/TP1; BR2304/12-1), and by the Israel Science Foundation (ISF) (3260/23 to VS-B). FEO and ZK are doctoral fellows in the doctoral program Metabolic and Cardiovascular Disease (MCD) (FWF, DKplus W 1226-B18) at the Medical University of Graz. MH is a fellow of the Molecular Medicine (MolMed) doctoral program at the Medical University of Graz supported by MEFO Graz and AIB was supported by ALLEA EFDS-FL1-25. The SIM equipment is part of the Nikon Center of Excellence, Graz, and is supported by the Austrian infrastructure program 2013/2014, Nikon Austria Inc., and BioTechMed. We appreciate the technical assistance from Anna Schreilechner, Nina Schlögl, Luca Alfred Schmid, Sowmya Sunkara, Anna Shteinfer-Kuzmine, Yusuf Ceyhun Erdogan, Sweta Trishna, and Mercedes Maier.

## Author contributions

**Furkan E Oflaz**: Data curation; Formal analysis; Validation; Investigation; Visualization; Methodology; Writing—original draft; Writing—review and editing. **Alexander I Bondarenko**: Data curation; Formal analysis; Validation;

Investigation; Methodology. **Michael Trenker**: Investigation. **Markus Waldeck-Weiermair**: Investigation. **Benjamin Gottschalk**: Data curation; Software; Validation; Investigation; Visualization; Methodology. **Eva Bernhart**: Investigation; Methodology. **Zhanat Koshenov**: Supervision; Investigation; Methodology. **Snježana Radulović**: Methodology. **Rene Rost**: Investigation; Methodology. **Martin Hirtl**: Investigation. **Johannes Pilic**: Investigation. **Aditya Karunanithi Nivedita**: Investigation. **Adlet Sagintayev**: Investigation. **Gerd Leitinger**: Resources. **Bent Brachvogel**: Resources. **Susanne Summerauer**: Methodology. **Varda Shoshan-Barmatz**: Visualization; Writing—review and editing. **Roland Malli**: Writing—review and editing. **Wolfgang F Graier**: Conceptualization; Supervision; Funding acquisition; Visualization; Writing—original draft; Project administration; Writing—review and editing.

Source data underlying figure panels in this paper may have individual authorship assigned. Where available, figure panel/source data authorship is listed in the following database record: biostudies:S-SCDT-10_1038-S44318-025-00454-9.

## Disclosure and competing interests statement

The authors declare no competing interests.

# Expanded View Figures

**Figure EV1. Characterization of AnxA5-KO cells.** ▶

(A) Immunoblots show the AnxA5 expression in WT and AnxA5-KO in HeLa and (B) EA.hy926 cells (Experiments are performed in Clone 21 indicated as a red rectangle). Uncropped blots are provided in the Source data. Panel (A) shows a redisplay of content from Fig. EV2J. (C) Graphical representation of genetically encoded FRET-based mitochondrial matrix targeted $Ca^{2+}$ sensor (4mtD3cpv). (D) Representative image of HeLa cells transfected with (4mtD3cpv). The cells have been pseudocolored to represent mitochondrial $Ca^{2+}$ levels as a ratio under basal (left panel) conditions or upon histamine stimulation (left panel) (Scale bar = 5 μm). (E) Average time courses of the 100 μM histamine-induced $[Ca^{2+}]_{Matrix}$ responses in WT (black) and AnxA5-KO (red) in EA.hy926 cells measured in $Ca^{2+}$-free buffer (containing 100 μM EGTA). (F) Bar graphs show the basal $[Ca^{2+}]_{Matrix}$ and (G) histamine-induced maximum $[Ca^{2+}]_{Matrix}$ levels in WT (black) and AnxA5-KO (red). Data points represent the mean ± SEM ($n_{WT} = 9/6$; $n_{AnxA5-KO} = 12/6$). The p-value for (G) is $p = 0.0011$ (**$p < 0.01$). (H) Mean time courses of the histamine-induced $[Ca^{2+}]_{Cyto}$ responses in WT (black) and AnxA5-KO (red) in EA.hy926 cells measured in $Ca^{2+}$-free buffer (containing 100 μM EGTA). (I) Bar graphs show the basal $[Ca^{2+}]_{Cyto}$ and (J) histamine-induced maximum $[Ca^{2+}]_{Cyto}$ levels in WT (black) and AnxA5-KO (red). Data points represent the mean ± SEM ($n_{WT} = 101/6$; $n_{AnxA5-KO} = 88/6$). The p-value for (J) is $p = 0.0094$ (**$p < 0.01$). (K) Representative Immunoblot shows the expression level of AnxA5 transfected either with shControl or shAnxA5. Uncropped blots are provided in the Source Data. (L) Bar graph represents immunoblot analysis of AnxA5 expression as mean ± SEM ($n_{shControl} = 3$; $n_{shAnxA5} = 3$). (M) Average time courses of the 100 μM histamine-induced $[Ca^{2+}]_{Matrix}$ responses in the presence (dashed lines) and absence (solid lines) of CGP37157 in shControl (black) and shAnxA5 (red) in HeLa cells. Data points represent the mean ± SEM ($n_{shControl} = 29/3$; $n_{shAnxA5} = 28/4$; $n_{shControl-CGP37157} = 56/3$; $n_{shAnxA5-CGP37157} = 43/4$). Significant differences were assessed with the two-tailed unpaired Student's t-test (**$p < 0.01$ and ns: not significant). Source data are available online for this figure.

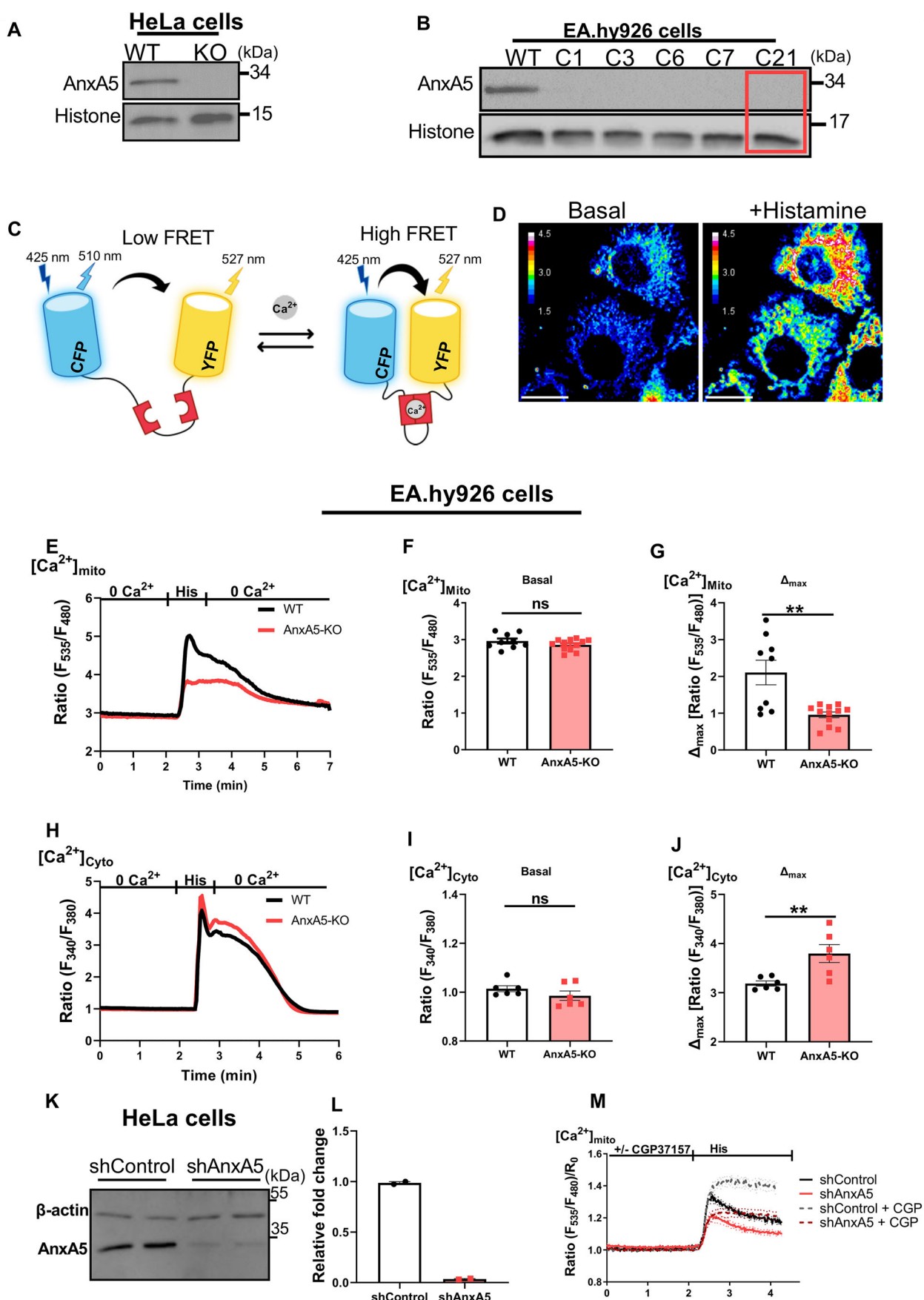

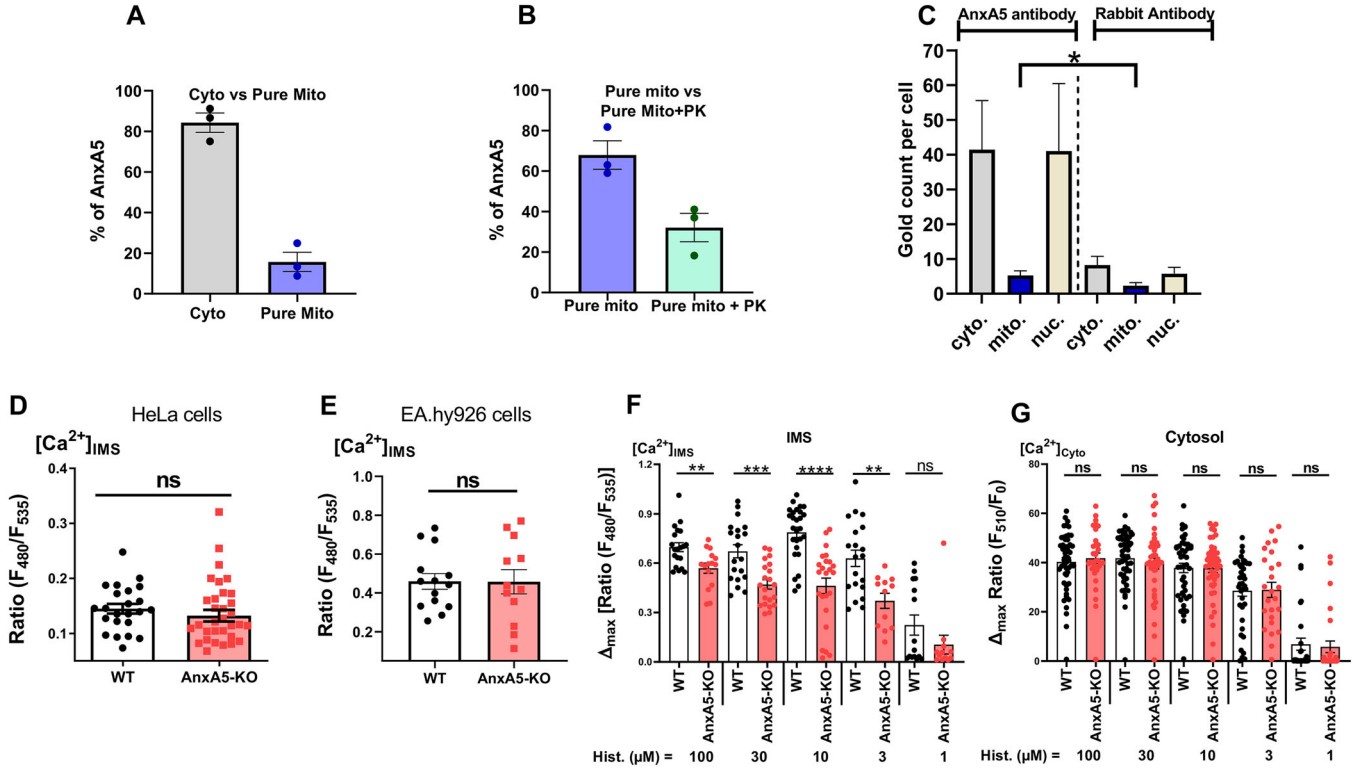

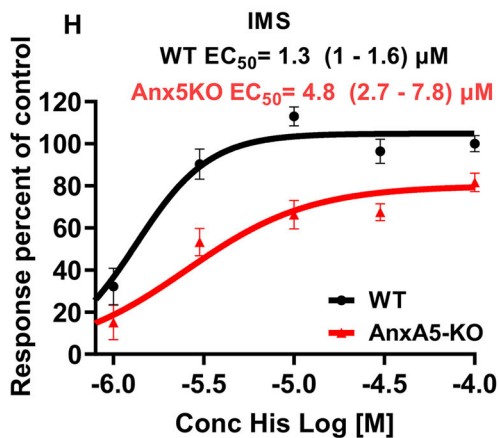

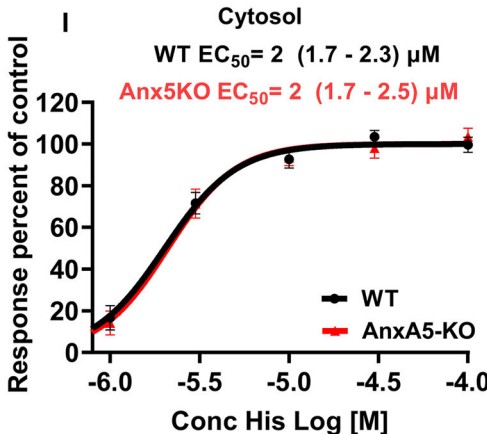

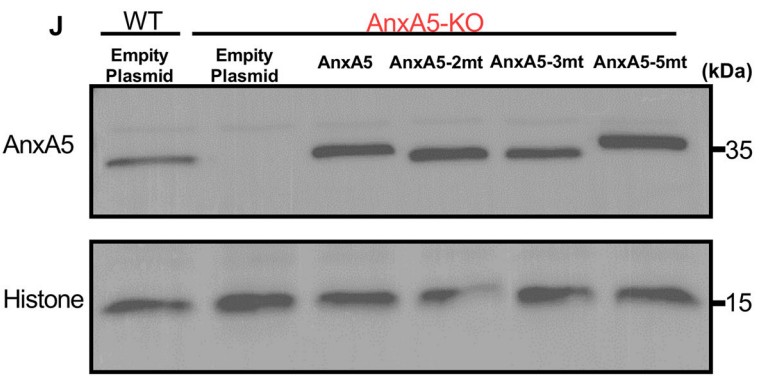

◄   **Figure EV2.   AnxA5 localizes on and within the mitochondria and regulates IMS Ca²⁺ signaling.**

(**A**) Bar graphs show the calculated percentage localization of AnxA5 in cytosol versus pure mitochondria, (**B**) pure mitochondria versus pure mitochondria + proteinase K (PK) treatment in mitochondria isolated from HeLa cells. Data points represent the mean ± SEM in HeLa cells ($n = 3$). (**C**) Bar graphs show the distribution of gold particles in cytosol, mitochondria, and the nucleus using AnxA5 antibody or rabbit antibody as a negative control. Data points represent the mean ± SEM in HeLa ($n_{AnxA5}$ $_{antibody} = 13/3$; $n_{Rabbit\ antibody} = 15/3$). The *p*-value is $p = 0.0337$ (*$p < 0.05$). (**D**) Bar graphs show the basal $[Ca^{2+}]_{IMS}$ levels in WT (black), AnxA5-KO (red), and Rescue (blue) in HeLa and (**E**) EA.hy926 cells. Data points represent the mean ± SEM in HeLa ($n_{WT} = 23/7$; $n_{AnxA5-KO} = 34/8$; $n_{Rescue} = 30/9$) and in EA.hy926 cells ($n_{WT} = 14/6$; $n_{AnxA5-KO} = 12/6$). (**F**) Bar graphs show a histamine-induced maximum $[Ca^{2+}]_{IMS}$ and (**G**) $[Ca^{2+}]_{cyto}$ elevation in WT (black) and AnxA5-KO (red) HeLa cells measured in Ca²⁺-free buffer. The *p*-values for (**F**), from left to right, are: $p = 0.0037$ (**$p < 0.01$), $p = 0.0002$ (***$p < 0.001$), $p < 0.0001$ (****$p < 0.0001$), $p = 0.0014$ (**$p < 0.01$), and $p = 0.1758$ (ns). (**H**) Concentration-response curve of histamine (1, 3, 10, 30, 100 μM) shows maximum $[Ca^{2+}]_{IMS}$ and (**I**) $[Ca^{2+}]_{Cyto}$ rise in WT (black) and AnxA5-KO (red) cells measured in Ca²⁺-free buffer. The values were calculated from panels (**F**) and (**G**). Data points represent the mean ± SEM in IMS ($n_{WT-100\ μM-Hist} = 19/4$; $n_{AnxA5-KO-100\ μM-Hist} = 14/5$; $n_{WT-30μM-Hist} = 19/5$; $n_{AnxA5-KO-30\ μM-Hist} = 20/5$; $n_{WT-10\ μM-Hist} = 26/6$; $n_{AnxA5-KO-10\ μM-Hist} = 23/6$; $n_{WT-3μM-Hist} = 19/6$; $n_{AnxA5-KO-3\ μM-Hist} = 11/6$; $n_{WT-1\ μM-Hist} = 14/3$; $n_{AnxA5-KO-1\ μM-Hist} = 11/3$) and in the cytosol ($n_{WT-100\ μM-Hist} = 50/5$; $n_{AnxA5-KO-100\ μM-Hist} = 33/4$; $n_{WT-30\ μM-Hist} = 50/5$; $n_{AnxA5-KO-30\ μM-Hist} = 40/5$; $n_{WT-10\ μM-Hist} = 46/4$; $n_{AnxA5-KO-10\ μM-Hist} = 46/4$; $n_{WT-3\ μM-Hist} = 39/4$; $n_{AnxA5-KO-3\ μM-Hist} = 25/4$; $n_{WT-1\ μM-Hist} = 30/5$; $n_{AnxA5-KO-1\ μM-Hist} = 28/5$). (**J**) Representative immunoblots show the expression of AnxA5 in WT and AnxA5-KO cells transfected either with an empty plasmid (in WT and AnxA5-KO cells) or with AnxA5, AnxA5-2mt, AnxA5-3mt, and AnxA5-5mt (in AnxA5-KO cells). Uncropped blots are provided in the Source Data. Significant differences were assessed using either one-way ANOVA with Tukey's multiple comparison tests or Kruskal–Wallis test (ns: not significant) and with the two-tailed unpaired Student's t-test or Kolmogorov–Smirnov (*$p < 0.05$, **$p < 0.01$, ***$p < 0.001$, ****$p < 0.0001$, and ns: not significant). Source data are available online for this figure.

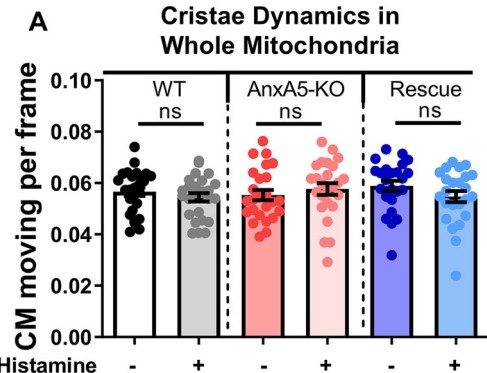

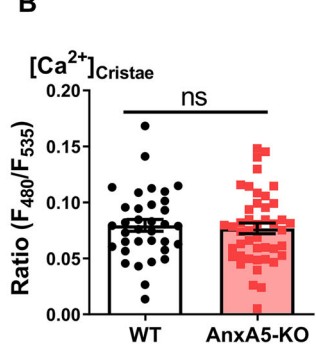

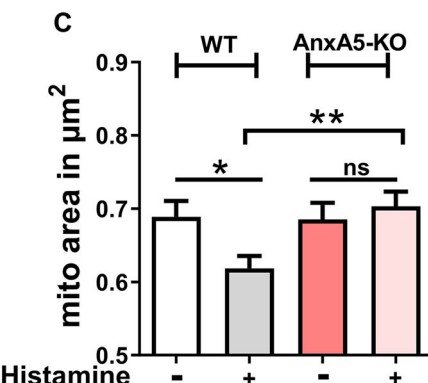

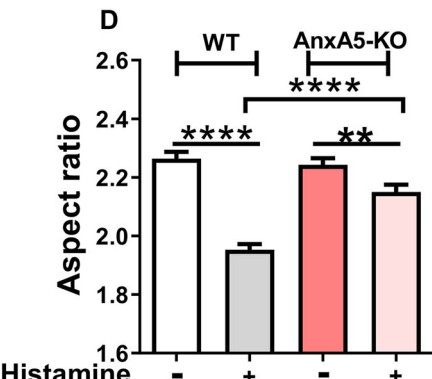

**Figure EV3. AnxA5 contributes to $Ca^{2+}$-induced remodeling of mitochondrial morphology.**

(A) Bar graph shows cristae membrane movements per frame in the whole mitochondria under basal and upon $IP_3$-induced $[Ca^{2+}]_{ER}$ release in WT (black), AnxA5-KO (red), and Rescue (blue) in HeLa cells. Data points represent the mean ± SEM ($n_{WT} = 26/6$; $n_{AnxA5-KO} = 25/6$; $n_{Rescue} = 24/6$). (B) Bar graph shows the basal $[Ca^{2+}]_{cristae}$ levels in WT (black) and AnxA5-KO (red) cells. Data points represent the mean ± SEM ($n_{WT} = 35/6$; $n_{AnxA5-KO} = 44/6$). (C) Bar graphs show the mitochondrial area and (D) aspect ratio in WT and AnxA5-KO before and 90 s after $[Ca^{2+}]_{ER}$ release. Data points represent the mean ± SEM ($n_{WT} = 84/9$; $n_{AnxA5-KO} = 88/9$). The p-values, from left to right, are for (C): $p = 0.0109$ (*$p < 0.05$), $p = 0.0015$ (**$p < 0.01$) and $p = 0.5559$ (ns); and for (D): $p < 0.0001$ (****$p < 0.0001$), $p < 0.0001$ (****$p < 0.0001$), and $p = 0.0066$ (**$p < 0.01$). Significant differences were assessed using either one-way ANOVA with Tukey's multiple comparison tests or the Kruskal–Wallis test and with the two-tailed unpaired Student's t-test (*$p < 0.05$, **$p < 0.01$, ****$p < 0.0001$, and ns: not significant). Source data are available online for this figure.

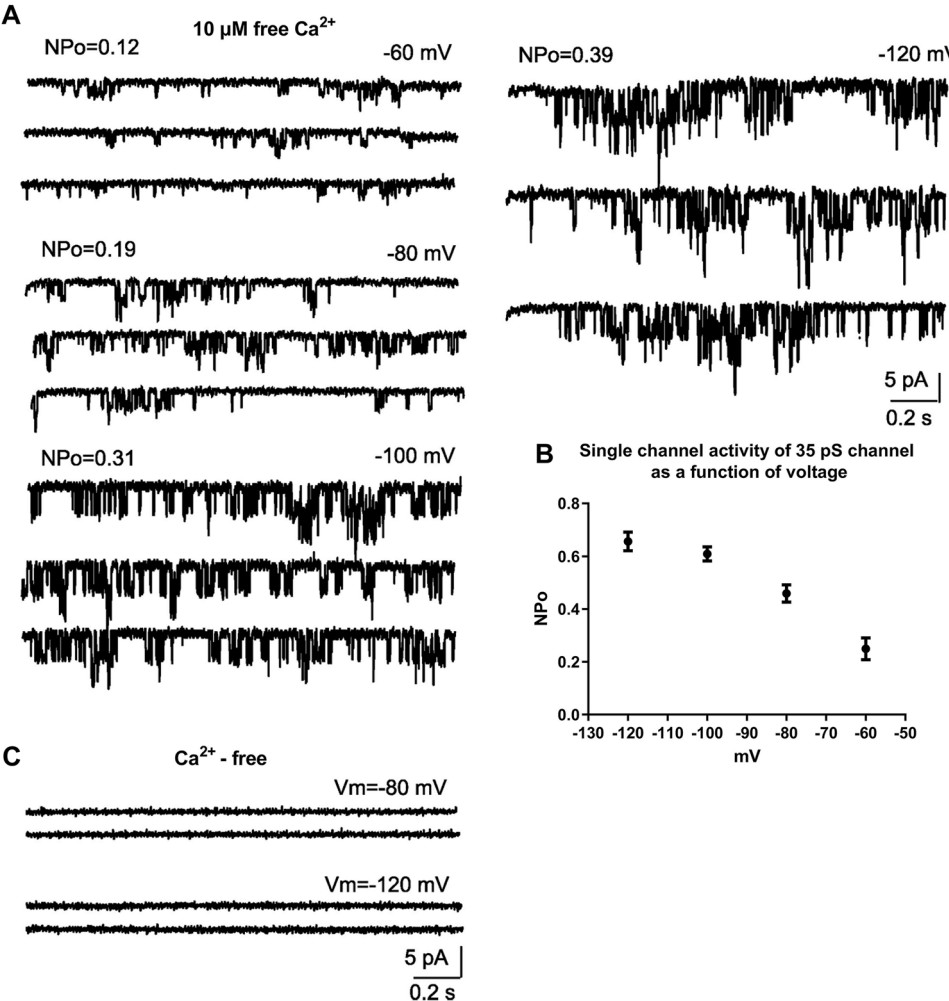

**Figure EV4. Characterization of the 35 pS channel at the OMM.**

(A) Representative single-channel traces showing the 35 pS channel in different voltages in intact mitochondria isolated from the WT HeLa cells (10 μM free $Ca^{2+}$ in the pipette). (B) Graph shows the mean NPo of the 35 pS channel at −60, −80, −100, and −120 mV. (C) Representative single-channel traces showing no channel activity in intact mitochondria isolated from the WT HeLa cells ($Ca^{2+}$-free). Data points represent the mean ± SEM ($n_{WT-60} = 3$; $n_{WT-80} = 7$; $n_{WT-100} = 7$; $n_{WT-120} = 10$). Source data are available online for this figure.

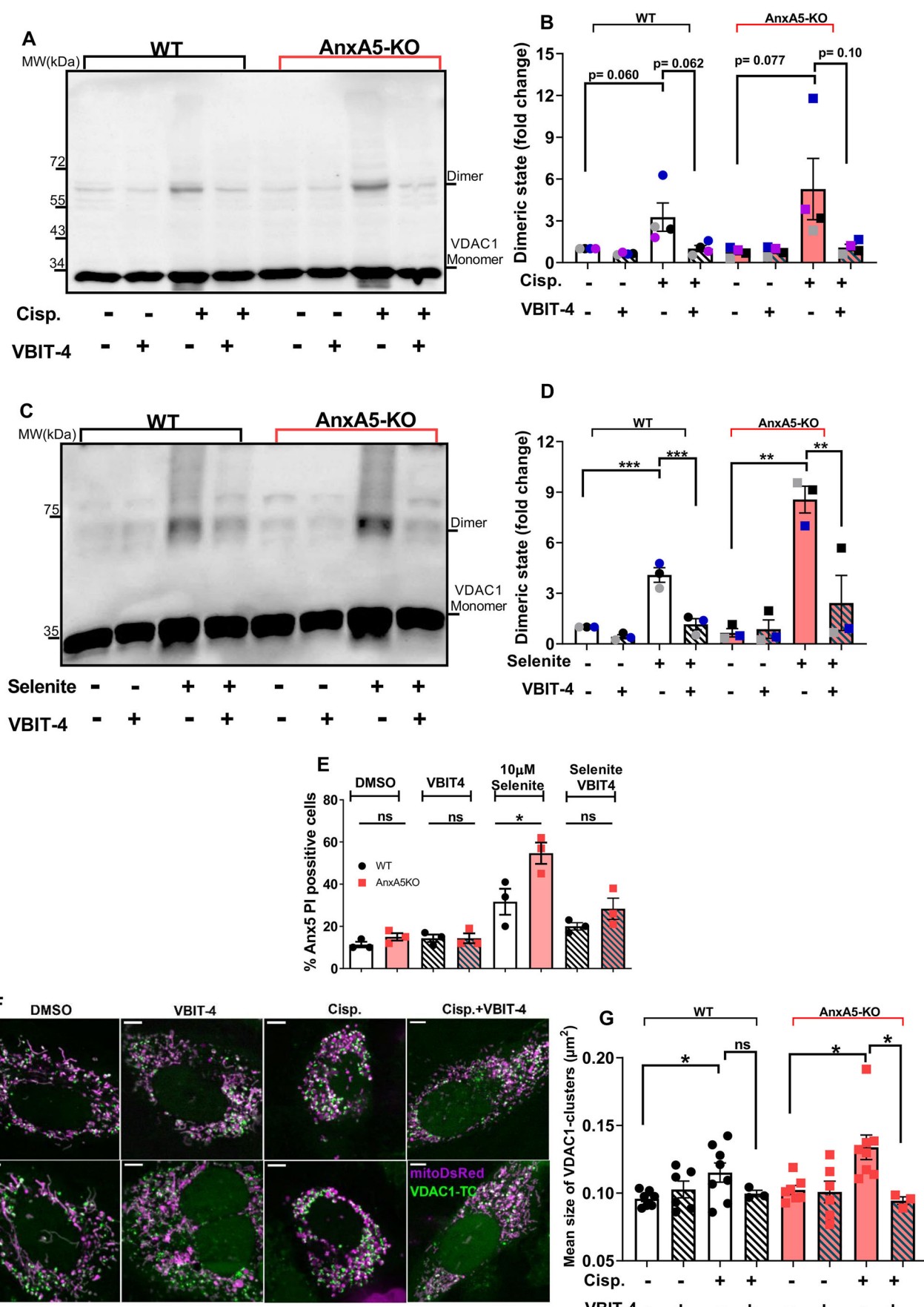

**Figure EV5. AnxA5 regulates cisplatin/selenite-induced VDAC1 dimerization.**

(A) Representative immunoblot shows monomeric and dimeric VDAC1 levels. Uncropped blots are provided in the Source data. (B) Bar graph shows the quantification of the immunoblot in WT (black) and AnxA5-KO (red) cells upon 24 h DMSO, 20 μM VBIT-4, 10 μM cisplatin, and cisplatin+VBIT-4 treatment (each color represents the experiments from the same day). Data points represent the mean ± SEM ($n_{\text{WT-all}} = 4$; $n_{\text{AnxA5-KO-all}} = 4$). (C) Representative immunoblot shows monomeric and dimeric VDAC1 levels in response to 48-hour DMSO, 20 μM VBIT-4, 10 μM selenite, and selenite + VBIT-4 treatment. (D) Bar graph shows the quantification of the immunoblot in panel (C). Data points represent the mean ± SEM ($n_{\text{WT-all}} = 3$; $n_{\text{AnxA5-KO-all}} = 3$). The *p*-values, from left to right, are: $p = 0.0002$ (***$p < 0.001$), $p = 0.0003$ (***$p < 0.001$), $p = 0.0017$ (**$p < 0.01$), and $p = 0.0082$ (**$p < 0.01$). (E) Bar graphs show the percentage of late apoptosis in WT (black) and AnxA5-KO (red) cells upon 48 h DMSO, 20 μM VBIT-4, 10 μM selenite, and selenite+VBIT-4 treatment. Data points represent the mean ± SEM ($n_{\text{WT-all}} = 3$; $n_{\text{AnxA5-KO-all}} = 3$). The *p*-values, from left to right, are: $p = 0.1688$ (ns), $p > 0.9999$ (ns), $p = 0.0448$ (*$p < 0.05$), and $p = 0.1932$ (ns). (F) Representative confocal images of WT and AnxA5-KO HeLa cells, expressing VDAC1-TC (green) and mitoDsRed (red), were captured (Scale bar = 5 μm). (G) Bar graph shows the quantification of the obtained confocal images indicating VDAC1 cluster size in μm² in WT (black) and AnxA5-KO (red) cells upon 12 h DMSO, 20 μM VBIT-4, 10 μM cisplatin, and cisplatin+VBIT-4 treatment. Data points represent the mean ± SEM ($n_{\text{WT-DMSO}} = 7$; $n_{\text{WT-VBIT-4}} = 6$; $n_{\text{WT-cisp.}} = 8$; $n_{\text{WT-cisp.+VBIT-4}} = 3$; $n_{\text{AnxA5-KO-DMSO}} = 7$; $n_{\text{AnxA5-KO-VBIT-4}} = 6$; $n_{\text{AnxA5-KO-cisp.}} = 8$; $n_{\text{AnxA5-KO-cisp.+VBIT-4}} = 3$). The *p*-values, from left to right, are: $p = 0.0494$ (*$p < 0.05$), $p = 0.2773$ (ns), $p = 0.0126$ (*$p < 0.05$), and $p = 0.0168$ (*$p < 0.05$). Significant differences were assessed using one-way ANOVA with Tukey's multiple comparison tests or with the unpaired Student's t-test (*$p < 0.05$, **$p < 0.01$, ***$p < 0.005$, and ns: not significant). Source data are available online for this figure.

