## [Peer Review File · The EMBO Journal]

Annexin A5 controls VDAC1-dependent mitochondrial Ca²⁺ homeostasis and determines cellular susceptibility to apoptosis

Furkan Enes Oflaz, Alexander Bondarenko, Michael Trenker, Markus Waldeck-Weiermair, Benjamin Gottschalk, Eva Bernhart, Zhanat Koshenov, Snjezana Radulovic, Rene Rost, Martin Hirtl, Johannes Pilic, Aditya Karunani Nivedita, Adlet Sagintayev, Gerd Leitinger, Bent Brachvogel, Susanne Summerauer, Varda Shoshan-Barmatz, Roland Malli, and Wolfgang Graier

Corresponding author: Wolfgang Graier (wolfgang.graier@medunigraz.at)

Review Timeline:

Submission Date:	10th Sep 24
Editorial Decision:	25th Oct 24
Revision Received:	25th Feb 25
Editorial Decision:	12th Mar 25
Revision Received:	21st Mar 25
Accepted:	11th Apr 25

Editor: Daniel Klimmeck

Transaction Report:

Dear Dr. Graier,

Thank you again for the submission of your manuscript (EMBOJ-2024-118437) to The EMBO Journal. Please accept my apologies for getting back to you with unusual delay due to protracted referee input and detailed discussion in the editorial team. As mentioned earlier, your study was assessed by two reviewers with expertise in mitochondrial calcium biology and cell death signaling, whose comments are enclosed below. Please note that we expected a third report, however this referee got delayed and was in the end not able to send his-her comments. We have now concluded to base our decision on the existing reviews.

As you will see from the experts' reports, the referees acknowledge the analysis and potential interest of your results. However, they also express major concerns regarding mechanistic completeness and robustness of the findings, which need to be addressed thoroughly to make them supportive of publication in the EMBO Journal. The reviewers also raise issues related to the data presentation, additional controls and improved methods annotation required, statistics applied and overall discussion of related literature, that would need to be conclusively addressed to achieve the level of robustness and clarity needed for The EMBO Journal.

Given the overall interest stated and broader angle of your findings, we are able to invite you to revise your manuscript experimentally to address the referees' comments. I need to stress though that we do require strong support from the referees on a revised version of the study in order to move on to publication of the work.

In light of the extensive experimentation requested, I would appreciate if you could contact me during the next weeks for exchange e.g. a video call to discuss your perspective on the comments and potential plan for revisions.

Please feel free to contact me if you have any questions or need further input on the referee comments.

When submitting your revised manuscript, please carefully review the instructions below.

Please feel free to approach me any time should you have additional questions related to this.

Thank you for the opportunity to consider your work for publication.

I look forward to your revision.

Best regards,

Daniel Klimmeck

Daniel Klimmeck, PhD
Senior Editor
The EMBO Journal

Instruction for the preparation of your revised manuscript:

- 1) a .docx formatted version of the manuscript text (including legends for main figures, EV figures and tables). Please make sure that the changes are highlighted to be clearly visible.
- 2) individual production quality figure files as .eps, .tif, .jpg (one file per figure).
- 3) a .docx formatted letter INCLUDING the reviewers' reports and your detailed point-by-point response to their comments. As part of the EMBO Press transparent editorial process, the point-by-point response is part of the Review Process File (RPF), which will be published alongside your paper.
- 4) a complete author checklist, which you can download from our author guidelines (<https://wol-prod-cdn.literatumonline.com/pb->

assets/embo-site/Author Checklist%20-%20EMBO%20J-1561436015657.xlsx). Please insert information in the checklist that is also reflected in the manuscript. The completed author checklist will also be part of the RPF.

6) It is mandatory to include a 'Data Availability' section after the Materials and Methods. Before submitting your revision, primary datasets produced in this study need to be deposited in an appropriate public database, and the accession numbers and database listed under 'Data Availability'. Please remember to provide a reviewer password if the datasets are not yet public (see <https://www.embopress.org/page/journal/14602075/authorguide#datadeposition>).

7) Our journal encourages inclusion of *data citations in the reference list* to directly cite datasets that were re-used and obtained from public databases. Data citations in the article text are distinct from normal bibliographical citations and should directly link to the database records from which the data can be accessed. In the main text, data citations are formatted as follows: "Data ref: Smith et al, 2001" or "Data ref: NCBI Sequence Read Archive PRJNA342805, 2017". In the Reference list, data citations must be labeled with "[DATASET]". A data reference must provide the database name, accession number/identifiers and a resolvable link to the landing page from which the data can be accessed at the end of the reference. Further instructions are available at .

8) At EMBO Press we ask authors to provide source data for the main and EV figures. Our source data coordinator will contact you to discuss which figure panels we would need source data for and will also provide you with helpful tips on how to upload and organize the files.

Numerical data can be provided as individual .xls or .csv files (including a tab describing the data). For 'blots' or microscopy, uncropped images should be submitted (using a zip archive or a single pdf per main figure if multiple images need to be supplied for one panel). Additional information on source data and instruction on how to label the files are available at .

9) We replaced Supplementary Information with Expanded View (EV) Figures and Tables that are collapsible/expandable online (see examples in <https://www.embopress.org/doi/10.15252/emj.201695874>). A maximum of 5 EV Figures can be typeset. EV Figures should be cited as 'Figure EV1, Figure EV2' etc. in the text and their respective legends should be included in the main text after the legends of regular figures.

11) For data quantification: please specify the name of the statistical test used to generate error bars and P values, the number (n) of independent experiments (specify technical or biological replicates) underlying each data point and the test used to calculate p-values in each figure legend. The figure legends should contain a basic description of n, P and the test applied. Graphs must include a description of the bars and the error bars (s.d., s.e.m.).

We realize that it is difficult to revise to a specific deadline. In the interest of protecting the conceptual advance provided by the work, we recommend a revision within 3 months (23rd Jan 2025). Please discuss the revision progress ahead of this time with the editor if you require more time to complete the revisions.

Referee #1:

Summary:

Oflaz et al. demonstrate that Annexin A5 regulates VDAC oligomerization during apoptosis. The authors use several advanced biochemical and biophysical techniques to support their conclusions. The authors deleted AnxA5 and showed differential Ca²⁺ homeostasis in multiple mitochondrial compartments (cristae, matrix, IMS). The authors show VDAC1 and AnxA5 co-localization without direct physical interaction. Using patch-clamping the authors show that AnxA5 regulates the open probability of an outer mitochondrial membrane channel. Annexin A5 is protective against cisplatin-induced cell death through altered VDAC oligomerization. This is an interesting manuscript describing a novel mechanism for Annexin A5's involvement in Ca²⁺ homeostasis and mitochondrial polarization/directionality for calcium responsiveness.

Major Concerns:

1. In figure 4 the authors use a MICU1-dependent FRET system to interrogate IMS Ca signaling and use FRET as a proxy for MICU1 oligomerization on/at the mitochondrial calcium uniporter. The authors have previously published that MICU1 is essential for uniporter re-localization in response to Ca stimulation and cristae maintenance (Gottschalk et al, Nature Communications 2019). Other publications have since implicated MICU1 in cristae maintenance and shown this to occur even in the absence of MCU (Tomar et al. Science Signaling. 2023). The authors show that AnxA5 deletion alters both IMS and Cristae Ca²⁺ and alters mitochondrial ultrastructure. Are the alterations in FRET signal the authors show in figure 4 due to changes in ultrastructure, Ca²⁺ homeostasis, or both? This question could be answered using a mutant that eliminates uniporter interaction of MICU1 while maintaining interaction with cristae organization proteins, utilization of MCU knockout cell lines, or other experiments.
2. Does Annexin A5 regulate other VDAC's or outer membrane Ca²⁺ channels? If patch clamp experiments were repeated in VDAC1 KO cells, would Annexin-A5 rescue cause no change in currents?

Minor Concerns:

1. The title of figure 1 is "AnxA5 regulates mitochondrial matrix Ca²⁺ homeostasis in different cell lines" however the title in the manuscript of the section describing figure 1 results is "AnxA5 controls the Ca²⁺ signaling in all mitochondrial compartments". The authors show in supplemental figure that Anx5a-KO's have differential IMS Ca using a GEMGeCO1 fused to IMS protein MICU1. Figure 1 could be improved by adding recordings of IMS Ca²⁺ in response to ATP as they have done for other compartments. Alternatively, the data presented in figure 4f could be moved to figure 1.
2. Are alterations in mitochondrial ultrastructure shown in figure 2 Ca²⁺ dependent? i.e. if cells were treated with ATP, Histamine, or EGTA prior to acquiring EM images would differences between AnxA5-KO and ctrl be exacerbated or reduced?
3. The authors could change the title of figure 6 to something that indicates that Anx5a increases open probability of an OMM channel as opposed to just modulates.
4. The argument being made in figure 7 could be made stronger by using additional apoptotic stimuli besides cisplatin
5. If experiments performed in figure 7 were performed on a shorter timeline (<1 hour instead of 24/48 hours) of cisplatin treatment would the Anx5a KO show a differential phenotype? Performing these experiments on a shorter timeline would support this being a primary mechanism regulating VDAC oligomerization.
6. Could the authors add molecular weight markers to panel 7h, supplemental 1k, and any other western blots without kD markers annotated?
7. With reference to figure 2c: Does loss of AnxA5 cause altered expression of uniporter components besides MICU1? (MICU2, MICU3, EMRE)

Referee #2:

The manuscript entitled "Annexin-A5 is fundamental for VDAC1-dependent mitochondrial Ca²⁺ homeostasis and determines the susceptibility to apoptosis" presents exciting new data demonstrating selective regulation of OMM permeability to Ca²⁺ by Annexin A5. General thinking in the field is that the OMM permeability to Ca²⁺ is constitutive and that it is only entry into the matrix that is regulated. Strong evidence is provided here revealing selective passage of Ca²⁺ through the OMM under the control of AnxA5, with clear implications to initiation of apoptosis. Overall, I find the paper very convincing, however, I did have

some specific questions as outlined below. In particular, the additivity of AnxA5 deletion and VBIT-4 confused me, as I would have anticipated AnxA5-mediated control of VDAC-1 to be wiped out by VBIT-4. While I do not think that the fact that this did not happen is a fatal flaw, I do think that the relationship between AnxA5 and VDAC-1 requires further consideration. VBIT-4 inhibits dimerization, without interfering with the effect of AnxA5 of the same endpoint. What does this say about the mechanisms of action of these compounds?

Comments:

1. In figure 2, the potential contribution of AnxA5 to MMP was discounted without depleting the ER of Ca²⁺. Considering that the effect of AnxA5 on mitochondrial Ca²⁺ loading was only observed after PLC activation, measuring MMP under similar conditions would be ideal to support this claim.

2. It is notable that the effects of AnxA5 deletion and VBIT-4 seemed fundamentally independent - dimerization and survival were affected by both changes additively and, hence, independently. Why wouldn't the effect of AnxA5 on VDAC1 dimerization be blocked by VBIT-4? The fact that they occur independent points to unrelated mechanisms.

Minor Comments:

1. In figure 3, Protein K digested Tom20, but not VDAC1. Given the location of VDAC1 in the OMM, please provide an explanation for non-experts as to why there is selective digestion of Tom20.

Response to the referees:

We sincerely thank both referees for the very constructive and valuable points they raised. We have tried to address each individual point by clarifying, rewriting, or restructuring our manuscript and, mostly, by performing novel experiments. We assure the referees that we have made all possible efforts to address their point as clearly as possible while we were facing a strict time plan. Moreover, we further elaborated the manuscript for easier reading and gaining clarity. We hope that our efforts are acknowledged by the referees and thank them once again for their help in improving our manuscript.

Referee #1:

Summary:

Oflaz et al. demonstrate that Annexin A5 regulates VDAC oligomerization during apoptosis. The authors use several advanced biochemical and biophysical techniques to support their conclusions. The authors deleted AnxA5 and showed differential Ca²⁺ homeostasis in multiple mitochondrial compartments (cristae, matrix, IMS). The authors show VDAC1 and AnxA5 co-localization without direct physical interaction. Using patch-clamping the authors show that AnxA5 regulates the open probability of an outer mitochondrial membrane channel. Annexin A5 is protective against cisplatin-induced cell death through altered VDAC oligomerization. This is an interesting manuscript describing a novel mechanism for Annexin A5's involvement in Ca²⁺ homeostasis and mitochondrial polarization/directionality for calcium responsiveness.

RESPONSE: We greatly appreciate Reviewer 1's interest in our study. Furthermore, we are thankful for the valuable feedback provided to enhance the clarity of our manuscript.

Major Concerns:

1. In figure 4 the authors use a MICU1-dependent FRET system to interrogate IMS Ca signaling and use FRET as a proxy for MICU1 oligomerization on/at the mitochondrial calcium uniporter. The authors have previously published that MICU1 is essential for uniporter re-localization in response to Ca stimulation and cristae maintenance (Gottschalk et al, Nature Communications 2019). Other publications have since implicated MICU1 in cristae maintenance and shown this to occur even in the absence of MCU (Tomar et al. Science Signaling. 2023). The authors show that AnxA5 deletion alters both IMS and Cristae Ca²⁺ and alters mitochondrial ultrastructure. Are the alterations in FRET signal the authors show in figure 4 due to changes in ultrastructure, Ca²⁺ homeostasis, or both? This question could be answered using a mutant that eliminates uniporter interaction of MICU1 while maintaining interaction with cristae organization proteins, utilization of MCU knockout cell lines, or other experiments.

RESPONSE: We appreciate this insightful comment. Unfortunately, the suggested experiment with MICU1 not binding to the MCU but to the cristae junction would result in a long-lasting active MCU associated with structural changes by itself. On the other hand, we know from our previous work that in MCU knockout cell lines, MICU1 distribution is strongly altered.

Moreover, deletion of MCU yields loss of fusion capacity and increased fission, thus changing the structure of mitochondria (1).

However, the referee's point is well taken, and thus we sought to measure “morphologically-independently” IMS Ca^{2+} . Therefore, we utilized initially the IMS-targeted GEM-GECO - a single-excitation, dual-emission ratiometric Ca^{2+} reporter - to monitor IMS Ca^{2+} . As shown in figure 3, the IMS Ca^{2+} signaling was reduced in AnxA5KO cells. For the targeting sequence of this sensor, we used the first 140 amino acids of MICU1 (2). This targeting sequence does not include the Ca^{2+} -binding EF-hand domains of MICU1. Hence, this sensor is not FRET-based; therefore, the observed changes in IMS Ca^{2+} levels cannot be attributed to alterations in mitochondrial morphology.

The experiments on MICU1 FRET aimed to confirm our data by utilizing the targeted GEM-GECO described above through a different approach that employed the Ca^{2+} dependency of MICU1 as a “ Ca^{2+} sensor.” Indeed, in our experiments, WT cells but not AnxA5KO cells exhibited a decreased mitochondrial area and aspect ratio that becomes measurable > 90 sec after agonist-induced ER Ca^{2+} release (Fig. EV3 c,d). However, in the same protocol, the signal representing the Ca^{2+} -triggered MICU1 singularization occurred > 1 min faster (Fig. 4b). Taken the temporal difference between the two readouts (morphology vs. MICU1 FRET), we believe that changes in MICU1-FRET are most likely not driven by changes in mitochondrial ultrastructure but reflect changes in IMS Ca^{2+} . Therefore, we concluded that the data in Figure 4, showing a reduced change in the MICU1-FRET signal in AnxA5KO cells compared to WT cells, are confirmative for the results from Figure 3 using a different technology. The changes were included at lines 273 to 275.

2. Does Annexin A5 regulate other VDAC's or outer membrane Ca^{2+} channels? If patch clamp experiments were repeated in VDAC1 KO cells, would Annexin-A5 rescue cause no change in currents?

@ Does Annexin A5 regulate other VDAC's or outer membrane Ca^{2+} channels?

RESPONSE: This is an excellent point. In HeLa cells, the expression levels of VDAC2 and VDAC3 are much lower compared with that of VDAC1 (new Appendix Fig. S2f). To address the referee's point, we investigated whether AnxA5 also affects the Ca^{2+} function of VDAC2 and VDAC3. To test this, we transiently expressed FLAG-tagged VDAC2 and VDAC3 in siVDAC1-treated WT and AnxA5KO cells. Mitochondrial localization of the FLAG-tagged VDAC2 and VDAC3 was confirmed using high-resolution fluorescence microscopy (new Appendix Fig. S2g). Next, we tested whether these isoforms could rescue the lack of VDAC1 in terms of mitochondrial Ca^{2+} signal. Our results show that VDAC2, but not VDAC3, could rescue the reduced mitochondrial Ca^{2+} signaling due to VDAC1 knockdown. Finally, we tested whether the potential rescue by VDAC2 and VDAC3 of the reduced mitochondrial Ca^{2+} signal due to the VDAC1 knockdown, depended on the presence of AnxA5. These experiments revealed that the rescue of VDAC2 in VDAC1 knockdown cells was sensitive to a knockout of AnxA5. These new data suggest that AnxA5 regulates the Ca^{2+} permeability of VDAC1 and VDAC2, while VDAC3 does not mimic VDAC1 in terms of its Ca^{2+} function in our

conditions (new Fig. 5c,d). This difference may be due to the co-localization of VDAC1 and VDAC2, while VDAC3 exhibits a distinct spatial distribution (3). These new findings have been included as figure 5c,d and Appendix Fig. S2f,g, were described in lines 293 to 312, and discussed in lines 482 to 485.

@ Does Annexin A5 regulate other VDAC's or outer membrane Ca²⁺ channels?

RESPONSE: At the current stage, we cannot exclude that AnxA5 affects other proteins in the outer mitochondrial membrane. However, because the correct targeting and the comparable quantity/intensity of mitochondria-targeted sensors were similar in WT and AnxA5 KO cells, we concluded that the protein uptake machinery in the outer mitochondrial membrane is not affected by the KO of AnxA5. Therefore, a general effect on the proteins of the OMM seems unlikely.

We also have analyzed the impact of AnxA5 knockout on a so far unknown current with 18 pS conductance that is distinct to the reported 35 pS Ca²⁺ channel and occurred occasionally during our experiments. In contrast to the 35 pS channel reported in our manuscript (Figure 6), the NPo (Control: 0.20 ± 0.08 vs. AnxA5MO: 0.29 ± 0.10; n=5; P = 0.432) and occurrence (Control: 11.8 ± 4.1 vs. AnxA5MO: 15.9 ± 6.2; n=5; P = 0.610) of this uncharacterized current were slightly but not significantly increased by the AnxA5 knockout (Figure R 1).

Figure R1: Effect of knockout of Annexin 5 on the small unidentified single-channel activity with 18 pS in the OMM. Bar graphs show channel NPo (left panel) and occurrence (right panel), calculated as the percentage of patches showing single-channel activity relative to the total number of patches on a given experimental day in WT (Control) and AnxA5-KO (AnxA5 KO). Data points represent the mean ± SEM (n = 5 for each). Testing significance using one-way ANOVA with Tukey's multiple comparison tests revealed no significance (left: P = 0.432; right: P = 0.610).

These findings let us suggest that AnxA5 does not regulate all proteins / channels in the outer mitochondrial membrane. In line with these findings, mutations in AnxA5's Ca²⁺-binding domains did not rescue IMS Ca²⁺ levels (Fig. 3k-m), suggesting that its Ca²⁺-dependent phospholipid-binding function is crucial for regulating VDAC1 Ca²⁺ permeability.

This indicates that VDAC1 may reside together with VDAC2 (3) in a lipid environment enriched with AnxA5, but is distinct from that of VDAC3. This assumption has been included in lines 482 to 485.

Hence, we attempted to pinpoint the specific negatively charged phospholipid to what AnxA5 binds as an aspect of specificity of the interaction of VDAC1/2 and AnxA5. The idea was to disrupt AnxA5 binding at the OMM, thereby interfering with its contribution to mitochondrial Ca^{2+} signaling. Since AnxA5 binds to cardiolipin-rich microdomains on isolated mitochondria (4, 5), we considered cardiolipin as a potential harboring lipid for AnxA5. Therefore, we knocked down Taffazin (Taf), an enzyme responsible for cardiolipin maturation (6) in both WT and AnxA5-KO cells and monitored IMS Ca^{2+} signaling upon IP_3 -induced ER Ca^{2+} release (Figure R 2a,b). Interestingly, we did not observe a difference in IMS Ca^{2+} signaling in siTaf-treated WT and AnxA5-KO cells (Figure R 2a,b), although siTaf-treated WT and AnxA5 cells showed a nearly 80% reduction in Taffazin mRNA expression levels (Figure R 2c).

To test further test whether cardiolipin is the binding lipid for AnxA5 in regard to VDAC1, we contacted Prof. Ya-Wen Liu and asked for an RFP-tagged mitoPLD construct (a member of the phospholipase D) that localizes to the cytosolic leaflet of the OMM and converts cardiolipin to phosphatidic acid (PA), thus reducing cardiolipin content (7, 8). As a control, we used catalytic-dead mitoPLD6 (mito Δ PLD) (7). We co-transfected the respective constructs with a matrix-targeted Ca^{2+} sensor (4mtD3cpv) in WT and AnxA5-KO cells and monitored mitochondrial Ca^{2+} levels (Figure R 2d). The expression of mitoPLD did not affect mitochondrial Ca^{2+} levels in either WT or AnxA5-KO cells (Figure R 2d,e).

Importantly, one needs to consider that siTaf treatment and the expression of mitoPLD may affect mitochondrial shape and ER-mitochondria contact sites (8, 9), which are crucial for local Ca^{2+} transfer to mitochondria, which might prevent the effect of AnxA5 on VDAC1. However, as the Ca^{2+} signals in WT cells remained comparable with that in cardiolipin reduced WT cells, there is no evidence for dramatic shape changes affecting mitochondrial Ca^{2+} signaling. Therefore, it is tempting to speculate that cardiolipin is not the harboring site of AnxA5. However, the specific phospholipid(s)/lipid composition in the OMM to which AnxA5 binds to exhibit its contribution to VDAC1 activity remain(s) unknown.

Figure R 2: The role of cardiolipin in AnxA5-dependent mitochondrial Ca^{2+} signaling.

a Mean time courses of the 100 μ M histamine-induced $[Ca^{2+}]_{IMS}$ responses of WT (black), AnxA5-KO (red), siTAF treated WT (green), and siTAF treated AnxA5-KO (orange) in HeLa cells measured in Ca^{2+} (2 mM) containing buffer. **b** Bar graphs show histamine-induced maximum $[Ca^{2+}]_{IMS}$ levels in WT (black) and AnxA5-KO (red), siTAF-treated WT (green), and siTAF-treated AnxA5-KO (orange) cells. Data points represent the mean \pm SEM (nWT=12/6; nAnxA5-KO=19/6; nWT+siTaf=12/6; nAnxA5-KO+siTaf=16/6). **c** Bar graphs show the knockdown efficiency of the siTaffazin-treated cells in WT and AnxA5-KO cells. **d** Mean time courses of the 100 μ M histamine-induced $[Ca^{2+}]_{matrix}$ responses of WT expressing mito Δ PLD (black), AnxA5-KO expressing mito Δ PLD (red), WT expressing mitoPLD (green), and AnxA5-KO expressing mitoPLD (orange) in HeLa cells measured in Ca^{2+} (2 mM) containing buffer. **e** Bar graphs show histamine-induced maximum $[Ca^{2+}]_{IMS}$ levels in WT expressing mito Δ PLD (black), AnxA5-KO expressing mito Δ PLD (red), WT expressing mitoPLD (green), and AnxA5-KO expressing mitoPLD (orange) cells. Data points represent the mean \pm SEM (nWT+mito Δ PLD=25/7; nAnxA5-KO+mito Δ PLD=24/6; nWT+mitoPLD=31/8; nAnxA5-KO+mitoPLD=32/7). Significant differences were assessed using one-way ANOVA with Tukey's multiple comparison tests (* $p < 0.05$, ** $p < 0.01$, *** $p < 0.005$, **** $p < 0.001$, and ns: not significant).

@ If patch clamp experiments were repeated in VDAC1 KO cells, would Annexin-A5 rescue cause no change in currents?

RESPONSE: This is an excellent point, and we did our best to address this query in additional experiments. While we immediately started CRISPR/Cas-mediated VDAC1 knockout cells, this procedure, including the electrophysiological, and functional analysis, will take at least half a year from now on - a time range that is not suitable for revision. Because of these foreseen issues and time constraints, we immediately started to perform experiments using VDAC1 knockdown in WT and AnxA5KO cells and are happy to be able to add these findings to our manuscript. We hope the referee agrees with our approach. In our new experiments, the knockdown of VDAC1 slightly reduced the 35 pS **channel occurrence** and the knockout of AnxA5 further reduced the occurrence (new Fig. 6a,b). Unfortunately, given the present result, which might be due to the modest efficiency of VDAC1 knockdown (60% reduction in Western blot analyses), a contribution of VDAC1 in the observed channel cannot be convincingly concluded. **Concerning the channel's Po**, the knockdown of VDAC1 showed an apparent reduction (new Fig. 6a-c). Notably, the knockout of AnxA5 did not further decrease the Po in VDAC1 depleted cells (new Fig. 6a-c).

Altogether, these findings suggest that VDAC1 plays a role in the biophysical properties of the 35 pS Ca^{2+} channel recorded, that, by itself, also essentially depends on AnxA5. These new findings have been included as figure 6-a-c, and were described in lines 345 to 347, and 351 to 356. Nevertheless, despite all these data that point to VDAC1 as being the 35 pS current, we must not conclude that the 35 pS channel is THE VDAC1 channel.

Minor Concerns:

1. The title of figure 1 is "AnxA5 regulates mitochondrial matrix Ca^{2+} homeostasis in different cell lines" however the title in the manuscript of the section describing figure 1 results is "AnxA5 controls the Ca^{2+} signaling in all mitochondrial compartments". The authors show in supplemental figure that AnxA5-KO's have differential IMS Ca^{2+} using a GEMGeCO1 fused to IMS protein MICU1. Figure 1 could be improved by adding recordings of IMS Ca^{2+} in response to ATP as they have done for other compartments. Alternatively, the data presented in figure 4f could be moved to figure 1.

RESPONSE: We thank the referee for this attentive point. The flow of the paper is designed to first point to the general effect of AnxA5 on mitochondrial matrix Ca^{2+} signals, as such measurements are most reported. As such recordings do not allow any conclusion on the point of action of AnxA5, sub-mitochondrial Ca^{2+} measurements (cristae, IMS) were subsequently presented. These findings were further fostered by measurements of MICU1 FRET, cristae dynamics, and MCU movements, guiding the reader to the evident "point of action", the OMM. Therefore, to address the referees' comment, we have updated the title of the manuscript section describing the results of Figure 1 from "AnxA5 controls Ca^{2+} signaling in all mitochondrial compartments" to "AnxA5 controls mitochondrial matrix Ca^{2+} signaling."

2. Are alterations in mitochondrial ultrastructure shown in figure 2 Ca²⁺ dependent? i.e. if cells were treated with ATP, Histamine, or EGTA prior to acquiring EM images would differences between AnxA5-KO and ctrl be exacerbated or reduced?

RESPONSE: Thank you for this excellent point. Indeed, Figures 2e-h (confocal microscopy) and Figures 2i-m (electron microscopy) were acquired under basal conditions. Although this may point to differences between WT and AnxA5KO in the mitochondrial volume and branching without agonist-induced Ca²⁺ elevations, spontaneous Ca²⁺ elevations cannot be ruled out in the time of measurements.

Using structured illumination microscopy, our data on mitochondrial shape (branching/aspect ratio, area) before and 90 sec. after agonist stimulation show that ER Ca²⁺ release significantly reduces mitochondrial area and aspect ratio in WT cells (Fig. EV3c, d). In contrast, AnxA5-KO cells exhibit no change in mitochondrial area and only slightly reduced aspect ratio (Fig. EV3c, d). Overall, these data confirm our concept as they indicate that the mitochondrial Ca²⁺-induced morphological changes upon an agonist stimulation are reduced in AnxA5 knockout cells.

3. The authors could change the title of figure 6 to something that indicates that AnxA5 increases open probability of an OMM channel as opposed to just modulates.

RESPONSE: As suggested by the referee, we changed the title of Figure 6 to "AnxA5 is fundamental for the occurrence and open probability of an OMM Ca²⁺ current." As we cannot exclude that the addition of recombinant AnxA5 recruits more VDAC1 proteins to act as a Ca²⁺ permeable pore, the current changed title is the most appropriate according to our data. We hope the referee agrees with this change.

4. The argument being made in figure 7 could be made stronger by using additional apoptotic stimuli besides cisplatin.

RESPONSE: We thank the referee for helping us to foster our work further. According to the referee's suggestion, we performed further experiments on VDAC1 dimerization and cell viability assays using selenite, as an additional apoptotic stimulus. A 48-hour selenite treatment induced enhanced VDAC1 dimerization (new Fig. EV5c, d) and cell death (new Fig. EV5e) in AnxA5KO cells compared to WT cells, which could be rescued by VBIT4 (new Fig. EV5c-e). Additionally, we conducted parallel dimerization experiments with cisplatin as a control further to validate our findings (Figure R 3). These data were similar to our previous findings with cisplatin and support our conclusions on an anti-apoptotic effect of AnxA5. These new findings have been included as Figure EV5c-e, were described in lines 394 to 400, and discussed in lines 502 to 505.

Figure R 3: AnxA5 regulates cisplatin-induced VDAC1 dimerization. Representative immunoblot shows monomeric and dimeric VDAC1 levels in response to 48-hour DMSO, 20 μ M VBIT-4, 10 μ M cisplatin, and cisplatin + VBIT-4 treatment.

5. If experiments performed in figure 7 were performed on a shorter timeline (<1 hour instead of 24/48 hours) of cisplatin treatment would the Anx5a KO show a differential phenotype? Performing these experiments on a shorter timeline would support this being a primary mechanism regulating VDAC oligomerization.

RESPONSE: This is an interesting question, thank you. To address the referee's point, we tested earlier cisplatin-induced cell death time points. According to these experiments, cell death begins after 12 to 14 hours, with no measurable differences observed before this time (new Appendix Fig. S3a). Further experiments with 5 and 10 μ M cisplatin at 12, 24, and 48 hours revealed no visible dimerization at 12 hours in either WT or AnxA5KO cells (new Appendix Fig. S3j). Dimerization was induced at 24 and 48 hours, with a more pronounced effect observed in AnxA5KO cells (new Appendix Fig. S3j). These new findings have been included as Appendix Fig. S3a,j, were described in lines 369 to 370 and 386 to 388.

6. Could the authors add molecular weight markers to panel 7h, supplemental 1k, and any other western blots without kD markers annotated?

RESPONSE: Thank you for pointing out this mistake. Accordingly, the molecular weight markers have been added to all western blot images in the manuscript.

7. With reference to figure 2c: Does loss of AnxA5 cause altered expression of uniporter components besides MICU1? (MICU2, MICU3, EMRE)

RESPONSE: Thank you for this interesting point. Accordingly, we have tested the expression levels of MICU2 and EMRE in AnxA5KO cells (Fig. 2c,d). We could not find any changes in the expression of these proteins upon knockout of AnxA5. Regarding MICU3, this is a brain-specific isoform that is not expressed in HeLa cells (10).

Referee#2:

The manuscript entitled "Annexin-A5 is fundamental for VDAC1-dependent mitochondrial Ca²⁺ homeostasis and determines the susceptibility to apoptosis" presents exciting new data demonstrating selective regulation of OMM permeability to Ca²⁺ by Annexin A5. General thinking in the field is that the OMM permeability to Ca²⁺ is constitutive and that it is only entry into the matrix that is regulated. Strong evidence is provided here revealing selective passage of Ca²⁺ through the OMM under the control of AnxA5, with clear implications for the initiation of apoptosis. Overall, I find the paper very convincing, however, I did have some specific questions as outlined below. In particular, the additivity of AnxA5 deletion and VBIT-4 confused me, as I would have anticipated AnxA5-mediated control of VDAC-1 to be wiped out by VBIT-4. While I do not think that the fact that this did not happen is a fatal flaw, I do think that the relationship between AnxA5 and VDAC-1 requires further consideration. VBIT-4 inhibits dimerization, without interfering with the effect of AnxA5 of the same endpoint. What does this say about the mechanisms of action of these compounds?

RESPONSE: We thank Reviewer 2 for acknowledging our efforts and appreciate the insightful feedback. We believe that AnxA5 stabilizes VDAC1 in its monomeric form, enhancing its Ca²⁺ permeability and hindering multimerization of VDAC1, thus, promoting cell survival under stress conditions. Therefore, rather than functioning as direct antagonists, AnxA5 and VBIT-4 may be part of a regulatory network in which AnxA5 supports VDAC1 activity by stabilizing its monomeric form without physical interaction. In contrast, VBIT-4 exhibits its effect by direct binding to VDAC1 (11) to inhibit dimerization and subsequent cell death.

Comments:

1. In figure 2, the potential contribution of AnxA5 to MMP was discounted without depleting the ER of Ca²⁺. Considering that the effect of AnxA5 on mitochondrial Ca²⁺ loading was only observed after PLC activation, measuring MMP under similar conditions would be ideal to support this claim.

RESPONSE: We thank the referee for suggesting this insightful question and suggestions of an important experiment. We performed the experiments accordingly. Like expected, stimulating cells with histamine, an IP₃-inducing agonist, resulted in hyperpolarization in

both WT and AnxA5KO cells. The hyperpolarization was slightly enhanced in AnxA5KO compared with WT cells (Figure R 4a-c). The hyperpolarization most likely is caused by Ca^{2+} -triggered activation of the TCA cycle via the dehydrogenases (in particular the IDH), which subsequently fuels the electron transport chain. The enhanced hyperpolarization in AnxA5KO cells despite reduced Ca^{2+} uptake is intriguing, and can result from an optimized balance between the Ca^{2+} -evoked hyperpolarization and the depolarizing effect by the divalent cations *per se*. While this aspect requires a detailed investigation, these results support our conclusion that AnxA5 regulates mitochondrial Ca^{2+} levels without causing mitochondrial membrane depolarization.

Figure R 4: The impact of AnxA5 on mitochondrial membrane potential following ER Ca^{2+} release. A representative time courses of Ψ_{mito} treated with 100 μM histamine and subsequently with 1 μM FCCP in WT (black) and AnxA5-KO (red) HeLa cells, stained with TMRM. **b** Bar graph showing the basal mitochondrial membrane potential. **c** Changes in histamine-induced membrane potential in WT (black) and AnxA5-KO (red) HeLa cells. Data points represent the mean \pm SEM ($n_{\text{WT}} = 98/11$; $n_{\text{AnxA5-KO}} = 96/11$).

2. It is notable that the effects of AnxA5 deletion and VBIT-4 seemed fundamentally independent - dimerization and survival were affected by both changes additively and, hence, independently. Why wouldn't the effect of AnxA5 on VDAC1 dimerization be blocked by VBIT-4? The fact that they occur independent points to unrelated mechanisms.

RESPONSE: This is an excellent point, and we completely agree with the assumption of the referee. We show that AnxA5 does not physically interact with VDAC1; instead, it localizes in the proximity of VDAC1 (Fig 5a,b), perhaps embedding the VDAC1 in a respective environment to stabilize the Ca^{2+} permeable monomeric form (please see our efforts to characterize further this environment above referee #1; point 2;). It is known that AnxA5 binds to lipids/lipid membranes in a Ca^{2+} -dependent manner, self-assembles, and reduces the diffusion of phospholipids in membranes (12), potentially influencing VDAC1's behavior. Indeed, a recent preprint indicated that the lipid composition of the membrane regulates the oligomeric formation of VDAC1 and that VDAC1 can organize lipids around itself, leading to the formation of specific VDAC1 assemblies (13). Therefore, it is tempting to believe that

the effect of AnxA5 on the dimerization of VDAC1 is mediated through its phospholipid-binding/stabilization function in a VDAC1-specific environment. In contrast, VBIT-4 has been developed to bind directly to VDAC1 affecting the activity of the protein directly (11). These assumptions have been discussed in lines 506 to 510.

Minor Comments:

1. In figure 3, Protein K digested Tom20, but not VDAC1. Given the location of VDAC1 in the OMM, please provide an explanation for non-experts as to why there is selective digestion of Tom20.

RESPONSE: While VDAC1, as a channel, is deeply embedded into the membrane, only the N-terminus (i.e. amino acid from 7 to 24) of TOM20 is inserted in the OMM, with the rest protruding into the cytosolic region (amino acid from 25 to 145) accessible to Proteinase K digestion (14). Accordingly, TOM20 is widely used as a marker to confirm effective Proteinase K digestion. Proteinase K digests the cytosolic domain, causing the TOM20 band to disappear (14), while the “deeper” location of VDAC1 in the bilayer might protect the protein. Accordingly, we added the following sentence to lines 191-194: " Both TOM20 and VDAC1 are localized in the OMM. PK digests TOM20's cytosolic protrusion, removing its band, while VDAC1, fully embedded in the OMM remains protected, indicating mitochondrial integrity and effective PK digestion."

References:

1. O. M. Koval, E. K. Nguyen, V. Santhana, T. P. Fidler, S. C. Sebag, T. P. Rasmussen, D. J. Mittauer, S. Strack, P. C. Goswami, E. D. Abel, I. M. Grumbach, Loss of MCU prevents mitochondrial fusion in G1-S phase and blocks cell cycle progression and proliferation. *Sci. Signal.* **12** (2019).
2. M. Waldeck-Weiermair, B. Gottschalk, C. T. Madreiter-Sokolowski, J. Ramadani-Muja, G. Ziomek, C. Klec, S. Burgstaller, H. Bischof, M. R. Depaoli, E. Eroglu, R. Malli, W. F. Graier, Development and Application of Sub-Mitochondrial Targeted Ca²⁺ Biosensors. *Front. Cell. Neurosci.* **13**, 449 (2019).
3. D. Neumann, J. Bückers, L. Kastrup, S. W. Hell, S. Jakobs, Two-color STED microscopy reveals different degrees of colocalization between hexokinase-I and the three human VDAC isoforms. (2010).
4. F. M. Megli, M. Mattiazzi, T. Di Tullio, E. Quagliariello, Annexin V Binding Perturbs the Cardiolipin Fluidity Gradient in Isolated Mitochondria. Can It Affect Mitochondrial Function?. *Biochemistry* **39**, 5534–5542 (2000).
5. F. M. Megli, M. Selvaggi, A. De Lisi, E. Quagliariello, EPR study of annexin V-cardiolipin Ca²⁺-mediated interaction in phospholipid vesicles and isolated mitochondria. *Biochim. Biophys. Acta - Biomembr.* **1236**, 273–278 (1995).
6. P. Hsu, X. Liu, J. Zhang, H.-G. Wang, J.-M. Ye, Y. Shi, Cardiolipin remodeling by TAZ/tafazzin is selectively required for the initiation of mitophagy. doi: 10.1080/15548627.2015.1023984 (2015).

7. Y. A. Su, H. Y. Chiu, Y. C. Chang, C. J. Sung, C. W. Chen, R. Tei, X. R. Huang, S. C. Hsu, S. S. Lin, H. C. Wang, Y. C. Lin, J. C. Hsu, H. Bauer, Y. Feng, J. M. Baskin, Z. F. Chang, Y. W. Liu, NME3 binds to phosphatidic acid and mediates PLD6-induced mitochondrial tethering. *J. Cell Biol.* **222** (2023).
8. S.-Y. Choi, P. Huang, G. M. Jenkins, D. C. Chan, J. Schiller, M. A. Frohman, A common lipid links Mfn-mediated mitochondrial fusion and SNARE-regulated exocytosis. *Nat. Cell Biol.* **8** (2006).
9. B. Gottschalk, C. Klec, G. Leitinger, E. Bernhart, R. Rost, H. Bischof, C. T. Madreiter-Sokolowski, S. Radulović, E. Eroglu, W. Sattler, M. Waldeck-Weiermair, R. Malli, W. F. Graier, MICU1 controls cristae junction and spatially anchors mitochondrial Ca²⁺ uniporter complex. *Nat. Commun.* **10** (2019).
10. M. Patron, V. Granatiero, J. Espino, R. Rizzuto, D. De Stefani, MICU3 is a tissue-specific enhancer of mitochondrial calcium uptake. *Cell Death Differ.* **26**, 179 (2018).
11. D. Ben-Hail, R. Begas-Shvartz, M. Shalev, A. Shteinfer-Kuzmine, A. Gruzman, S. Reina, V. De Pinto, V. Shoshan-Barmatz, Novel Compounds Targeting the Mitochondrial Protein VDAC1 Inhibit Apoptosis and Protect against Mitochondrial Dysfunction. *J. Biol. Chem.* **291**, 24986–25003 (2016).
12. A. Bouter, C. Gounou, R. Bérat, S. Tan, B. Gallois, T. Granier, B. L. D'Estaintot, E. Pöschl, B. Brachvogel, A. R. Brisson, Annexin-A5 assembled into two-dimensional arrays promotes cell membrane repair. *Nat. Commun.* **2**, 1–9 (2011).
13. E. Lafargue, J.-P. Duneau, N. Buzhinsky, P. Ornelas, A. Ortega, V. Ravishankar, J. Sturgis, I. Casuso, L. Bergdoll, Lipid composition of the membrane governs the oligomeric organization of VDAC1. *bioRxiv*, 2024.06.26.597124 (2024).
14. K. Sch, € Afer, C. Engstler, K. Dischinger, C. Carrie, Chapter 13 Assessment of Mitochondrial Protein Topology and Membrane Insertion. doi: 10.1007/978-1-0716-1653-6_13.

Dear Wolfgang,

Thank you for submitting your revised manuscript (EMBOJ-2024-119002R) to The EMBO Journal, as well for your patience with our response. Your amended study was sent back to the referees for their scientific reassessment, and we have received reports from both of them, which I enclose below. As you will see, the experts state that the work has been substantially enhanced by the revisions and they are now broadly in favour of publication.

Thus, we are pleased to inform you that your manuscript has been accepted in principle for publication in The EMBO Journal.

We now need you to take care of a number of issues related to formatting and data presentation as detailed below, which should be addressed at re-submission.

Please contact me at any time if you have additional questions related to below points.

As you might have noted from our webpage, every paper at the EMBO Journal now includes a 'Synopsis', displayed on the html and freely accessible to all readers. The synopsis includes a 'model' figure as well as 2-5 one-short-sentence bullet points that summarize the article. I would appreciate if you could provide this figure and the bullet points.

Thank you for giving us the chance to consider your manuscript for The EMBO Journal. I look forward to your final revision.

Again, please contact me at any time if you need any help or have further questions.

Best regards,

Daniel

>> Please add up to five keywords to your study.

>> Author Contributions: Remove the author contributions information from the manuscript text. Note that CRediT has replaced the traditional author contributions section as of now because it offers a systematic machine-readable author contributions format that allows for more effective research assessment. and use the free text boxes beneath each contributing author's name to add specific details on the author's contribution.

More information is available in our guide to authors.
<https://www.embopress.org/page/journal/14602075/authorguide>

>> Provide a completed Author Checklist. All positive responses need to have the manuscript section specified in the third column (missing for the first item in "Cell materials").

>> Section order should be corrected as follows: Title page - Abstract & Keywords - Introduction - Results - Discussion - Methods - Data Availability - Acknowledgements - Disclosure and Competing Interests Statement - References - Figure Legends - Table(s) - Expanded View Figure Legends.

>> Adjust the title of the 'Competing Interests' section to 'Disclosure and Competing Interests Statement' and move after Acknowledgements.

>>Appendix: please "Appendix for + ms title" on the title page before the ToC with the page numbers for the listed items.

>> Reagents and Tools table: please provide as a separate file using the existing template in the Guide For Authors, listing key reagents, experimental models, software and relevant equipment.

>> Data availability section: Rename the current 'Data and materials availability ' paragraph to 'Data availability section'. Add a statement: 'No large-scale data amenable to data repository deposition were generated in this study.'

>> References: adjust reference format to EMBO Journal format, 10 authors et al, and place References after the Discussion, before figure legends.

>> Please recheck references for the bioRxiv entry Lafargue et al. (2024) and update the citation if in the meantime published as regular article.

>> Figure callouts: Please ensure that the figures and panels are called out in sequential order. Currently, Fig. 5F,G are not called out.

>> Remove the one sentence summary from the manuscript.

>> Funding: please enter the following funding information in "Acknowledgements" section; funding info needs to be congruent - mismatch: in manuscript file: Israel Science Foundation (ISF) (3260/23) / in our online system: Israel Science Foundation (ISF) (974/19); funding info included in the Comments box could not be extracted by Production team, and therefore all funders should be added to "More Funders" list.

>> Figure checks:

>>>> please indicate redisplay of content Figure3B in the Figure legend for Appendix Figure S2H.

>>>> Provide full set of uncropped source data for Appendix Figure S3J.

>>>> Figure 4H: Clear demarcation lines need to added for each cell separating the Basal from the Histamine.

>>>> Figure 5A: enhance resolution of data provided.

>>>> Figure EV1A: provide blots at higher resolution.

>> Consider additional changes and comments from our production team as indicated below:

- Figure Legends (main + EV): 1. Please note that the figure 1E is mislabeled as figure 1A in the legends of the manuscript. This needs to be rectified.

2. Please note that the figure 1L is mislabeled as figure 1I in the legends of the manuscript. This needs to be rectified.

3. Please note that the exact p values are not provided in the legends of figures 1C, L; 2F, G, H; 3D, F, I, M; 4C, E, G, I; 5B, D; 6B, C; 7A, B, D, E, F, G, I; EV1 G, J; EV2 C, F; EV3 C, D; EV5 D, E, G

4. Please note that in figures 5B, D; 6B, C there is a mismatch between the annotated p values in the figure legend and the annotated p values in the figure file that should be corrected.

5. Please note that information related to n is missing in the legends of figures 2K, 5E

6. Please note that the error bars are not defined in the legend of figure 2K

Referee #1:

The authors have added several new experiments that corroborate their hypothesis. Overall, a really nice manuscript that details a novel mechanism for Annexin A5's contribution to calcium homeostasis and cell death.

I have no further questions or concerns.

Referee #2:

I greatly appreciated the careful responses to my comments. No further concerns.

The authors addressed the remaining editorial issues.

Dear Dr. Graier,

Thank you for submitting the revised version of your manuscript. I have now evaluated your amended manuscript and concluded that the remaining minor concerns have been sufficiently addressed.

I am thus pleased to inform you that your manuscript has been accepted for publication in the EMBO Journal.

Related, I would like to hereby ask your consent on keeping the referee response figures included in this file.

On a different note, I would like to alert you that EMBO Press offers a format for a video-synopsis of work published with us, which essentially is a short, author-generated film explaining the core findings in hand drawings, and, as we believe, can be very useful to increase visibility of the work. Please see the following link for representative examples and their integration into the article web page:

<https://www.embopress.org/doi/full/10.15252/emj.2019103932>

Best regards,

Daniel Klimmeck

Daniel Klimmeck, PhD
Senior Editor
The EMBO Journal
EMBO
Postfach 1022-40
Meyerohofstrasse 1
D-69117 Heidelberg
contact@embojournal.org